# SPARC: Concept-Aligned Sparse Autoencoders for Cross-Model and Cross-Modal Interpretability

**Ali Nasiri-Sarvi**                                    *ali.nasirisarvi@mail.concordia.ca*
*Department of Computer Science and Software Engineering (CSSE)*
*Concordia University, Canada*

**Hassan Rivaz**                                    *hrivaz@ece.concordia.ca*
*Department of Electrical and Computer Engineering (ECE)*
*Concordia University, Canada*

**Mahdi S. Hosseini**                                    *mahdi.hosseini@concordia.ca*
*Department of Computer Science and Software Engineering (CSSE)*
*Concordia University, Canada*
*Mila–Quebec AI Institute*

**Reviewed on OpenReview:** *https://openreview.net/forum?id=IJfvoc2GbZ*

## Abstract

Understanding how different AI models encode the same high-level concepts, such as objects or attributes, remains challenging because each model typically produces its own isolated representation. Existing interpretability methods like Sparse Autoencoders (SAEs) produce latent concepts individually for each model, resulting in incompatible concept spaces and limiting cross-model interpretability. To address this, we introduce SPARC (Sparse Autoencoders for Aligned Representation of Concepts), a new framework that learns a single, unified latent space shared across diverse architectures and modalities (e.g., vision models like DINO, and multimodal models like CLIP). SPARC's alignment is enforced through two key innovations: (1) a Global TopK sparsity mechanism, ensuring all input streams activate identical latent dimensions for a given concept; and (2) a Cross-Reconstruction Loss, which explicitly encourages semantic consistency between models. On Open Images, SPARC dramatically improves concept alignment, achieving a Jaccard similarity of 0.80, more than tripling the alignment compared to previous methods. SPARC creates a shared sparse latent space where individual dimensions often correspond to similar high-level concepts across models and modalities, enabling direct comparison of how different architectures represent identical concepts without requiring manual alignment or model-specific analysis. As a consequence of this aligned representation, SPARC also enables practical applications such as text-guided spatial localization in vision-only models and cross-model/cross-modal retrieval. Code and models are available at `https://github.com/AtlasAnalyticsLab/SPARC/`.

## 1 Introduction

As AI models rapidly grow in numbers, a fundamental question emerges: do different architectures, trained with different objectives and modalities, independently converge on similar ways of representing the world? Answering this requires tools that can compare models directly. However, current interpretability methods, including powerful Sparse Autoencoders (SAEs) Bricken et al. (2023); Huben et al. (2024), are designed to analyze models in isolation. While effective, this approach creates isolated concept spaces unique to each model, making direct comparison difficult.

Recent work has begun addressing cross-model interpretability more directly. Universal Sparse Autoencoders (USAE) Thasarathan et al. (2025) introduced the concept of training a single sparse dictionary across

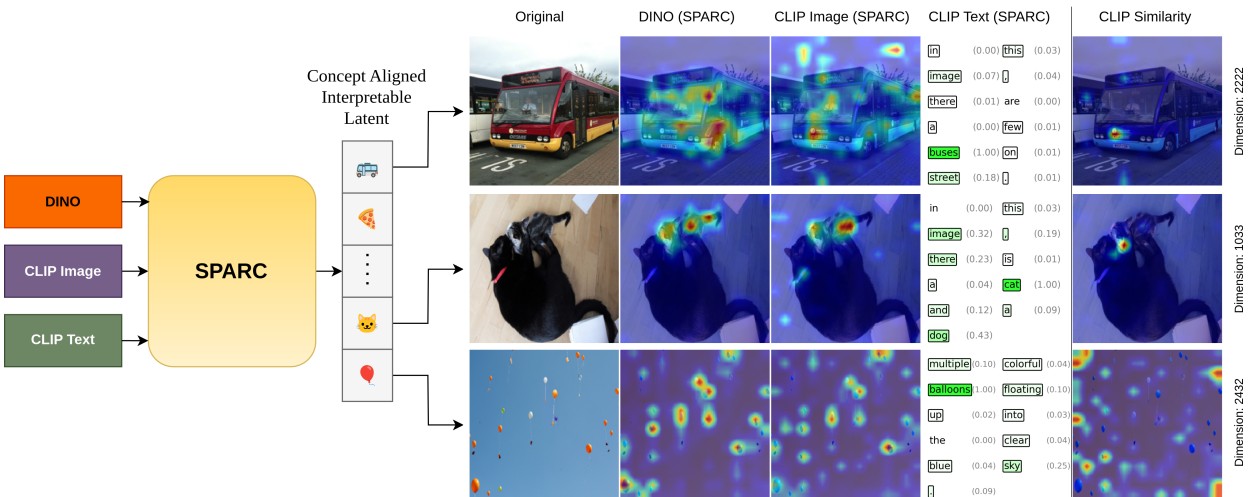

Figure 1: SPARC enables consistent concept visualization across models and modalities using shared latent dimensions. The figure shows how individual concept-specific latents (bus, cat, balloons) produce coherent spatial heatmaps across DINO Oquab et al. (2024) and CLIP Radford et al. (2021) vision encoders, while also generating meaningful text attribution scores in CLIP's text encoder when processing full image captions. For comparison, standard CLIP similarity uses only concept names ("a bus", "a cat", "balloons") rather than full captions, as CLIP produces diffused and scattered heatmaps when given complex captions. SPARC's concept-specific latents can handle full captions while maintaining focused attribution on the target concept.

multiple vision models, using random encoder selection during training to balance computational costs. At each training iteration, USAE randomly selects one model's encoder to produce shared concept activations, then reconstructs all models' features using their respective learned decoders. This approach demonstrated the feasibility of learning shared representations and revealed interesting patterns in how different models encode visual concepts, representing an important step forward in cross-model interpretability.

However, USAE faces several inherent limitations for cross-model interpretability. Methodologically, USAE's training approach randomly selects one model's encoder per iteration while reconstructing all models, which can lead to training instabilities and uneven concept learning across architectures. The method lacks explicit mechanisms to ensure consistent activation patterns across models, potentially allowing latent dimensions to activate differently across architectures without constraint. Additionally, USAE's focus on vision-only models limits its applicability to modern multimodal AI systems. From an evaluation perspective, while USAE demonstrates statistical co-activation and reconstruction fidelity, the fundamental challenge remains in validating whether these statistically aligned activations represent genuinely equivalent semantic concepts across different architectures, rather than merely correlated but distinct representations.

Building on these insights, we introduce SPARC (**Sp**arse Autoencoders for **A**ligned **R**epresentation of **C**oncepts), a method designed to achieve concept alignment across heterogeneous models and modalities while dramatically reducing the manual analysis burden that limits current approaches. By learning a single interpretable latent space that works across multiple models simultaneously, SPARC enables experts to analyze concept representations once rather than repeatedly for each architecture, directly addressing the scalability challenges that motivated this work.

SPARC achieves these objectives through two key methodological innovations: (1) a **Global TopK** sparse activation mechanism that enforces identical activation patterns across all input streams, directly addressing the dead latent problem by ensuring every dimension either activates across all models or remains inactive across all models; and (2) a **Cross-Reconstruction Loss** objective that promotes semantic consistency by training each model's latent representation to reconstruct inputs from other models, creating optimization pressure toward shared semantic understanding rather than mere statistical correlation.

Our evaluation framework validates these aligned representations through multiple complementary approaches: (1) semantic concept alignment measurement; (2) practical utility demonstration through downstream applications including cross-modal attribution, semantic segmentation, and retrieval tasks; and (3) systematic dead neuron quantification across models.

SPARC's design is highly effective, achieving a concept alignment Jaccard similarity of 0.80 on Open Images. In direct comparison using the same evaluation protocol, USAE achieves only 0.22 (Section 4.2). This gap reflects a key architectural difference: USAE relies on soft alignment through reconstruction without an explicit constraint on which latent indices activate, allowing different streams to use different subsets of latents for the same input. SPARC's Global TopK, by contrast, enforces identical active indices across all streams, ensuring each latent dimension either activates consistently across models or remains inactive across all of them. This creates a unified latent space where a single dimension consistently represents the same concept across models like DINO and CLIP, and even across vision and text modalities, as demonstrated in Figure 1.

## 2 Related Work

Understanding what neural networks learn—and how to describe those latent mechanisms in human terms—has become a central challenge for interpretable and trustworthy AI. Early interpretability efforts uncovered neurons that align with human concepts (e.g., textures, object parts) in vision models (Bau et al. (2017; 2019)) and language models (Karpathy et al. (2015); Radford et al. (2017)), but later work revealed that individual units are often polysemantic, entangling many unrelated features within the same activation (Goh et al. (2021); Elhage et al. (2022)).

Representation engineering treats linear directions in a model's activation space as semantic units that can be read or edited Zou et al. (2023). Reading tools include TCAV, which measures a concept vector's influence on predictions (Kim et al. (2018)); ACE, which discovers such vectors by clustering activations Ghorbani et al. (2019); and concept-bottleneck models that expose an explicit concept layer for inspection and override Koh et al. (2020). Steering tools shift activations along learned directions, as in Plug-and-Play Language Models for controlled text generation Dathathri et al. (2020) and InterFaceGAN for semantic image edits Shen et al. (2020). Unsupervised work has also uncovered latent knowledge vectors that reveal and toggle factual beliefs in language models Burns et al. (2023).

While representation engineering works with a model's existing geometry, sparse autoencoders learn a new, disentangled basis first. Dictionary-learning SAEs with an $l_1$ penalty were initially applied to transformer language models, revealing tens of thousands of single-concept directions in GPT-2 activations Bricken et al. (2023); follow-up work showed that the same technique scales to larger language models and yields higher automated interpretability scores across layers Huben et al. (2024). Parallel studies in vision apply sparse autoencoders to convolutional and ViT feature maps, uncovering units aligned with colors, textures, and object parts, and enabling direct feature-level editing in images Lim et al. (2024). A separate line of work abandons the $l_1$ penalty in favor of the hard TopK selection rule introduced by k-Sparse Autoencoders Makhzani & Frey (2013); scaling this constraint to billion-parameter networks further improves feature purity and removes dead units Gao et al. (2025). Building on this, BatchTopK Sparse Autoencoders propose relaxing the hard sparsity constraint from a per-sample to a per-batch basis, which allows for a variable number of active latents per sample and enables the model to adaptively allocate more features to more complex inputs Bussmann et al. (2024). Together, these results establish sparsity as a reliable way to manufacture monosemantic features.

Rosetta Neurons identifies neurons in vision backbones that respond to the same visual patterns Dravid et al. (2023), and Rosetta Concepts clusters semantically aligned features in transformer video models Kowal et al. (2024). Kondapaneni et al. (2025) compares models by aligning interpretable concept masks, while Dominici et al. (2023) builds a manifold of concepts across modalities by embedding direction vectors from each stream into a shared space. Most of these techniques operate post-hoc or depend on modality-specific heuristics.

Universal Sparse Autoencoders (USAE) take a more general approach, learning a single sparse dictionary that can reconstruct multiple vision models at once, revealing a common pool of interpretable features

Thasarathan et al. (2025). However, USAEs rely on random encoder sampling during training and soft alignment through their reconstruction objective, which can lead to inconsistent concept activation across models. SPARC addresses these limitations through two key innovations: (1) a Global TopK mechanism that enforces hard structural alignment by ensuring all models activate identical latent dimensions for the same inputs, and (2) a systematic cross-reconstruction objective that combines self-reconstruction and cross-reconstruction losses rather than random sampling. We demonstrate SPARC's power by learning unified concept representations that operate both cross-model (across different vision models) and cross-modal (bridging vision and language), revealing how the same semantic concepts manifest consistently across diverse model types and modalities.

## 3 Methodology

This section presents our methodology for SPARC: **SP**arse Autoencoder for **A**ligned **R**epresentation of **C**oncepts. We first define the problem setup and core objectives: faithfulness, interpretability, and cross-stream concept alignment (Section 3.1). We then detail the SPARC architecture, highlighting the novel Global TopK activation mechanism that enables concept alignment (Section 3.2). Finally, we formulate the training objective that optimizes the model toward these goals (Section 3.3).

### 3.1 Problem Setup and Objectives

We consider multiple ($M$) distinct streams of information, indexed by the set $\mathcal{S} = \{s_1, s_2, \ldots, s_M\}$. The core assumption is that we process data samples with multiple related representations of the same underlying entity (e.g., features from an image extracted using both CLIP-image Radford et al. (2021) and DINO Oquab et al. (2024) models, or image-caption pairs passed to DINO and CLIP-text ). For each sample, we obtain a set of corresponding feature vectors $\{\mathbf{x}^s\}_{s \in \mathcal{S}}$, where each $\mathbf{x}^s \in \mathbb{R}^{d_s}$ is produced by stream $s$'s feature extraction process with dimensionality $d_s$.

These streams represent heterogeneous processing pathways; they may use different architectures, training objectives (like CLIP vs DINO), or process different modalities associated with the input (like image vs text). Consequently, our framework must accommodate varying input feature dimensionalities ($d_s \neq d_t$ for $s \neq t$ is permissible).

We aim to develop interpretable representations for each input stream while ensuring consistent cross-stream semantic alignment. Given different feature representations $\{\mathbf{x}^s\}_{s \in \mathcal{S}}$ of the same underlying data sample, the SPARC model maps these inputs into concept-aligned sparse latent representations $\{\mathbf{z}^s\}_{s \in \mathcal{S}}$. These latent representations $\mathbf{z}^s$ share a common $L$-dimensional space, $\mathbf{z}^s \in \mathbb{R}^L$, maintain sparsity $\|\mathbf{z}^s\|_0 \ll L$, and crucially, activate the same semantic dimensions across different streams when processing the same underlying content.

The design and learning of the SPARC are guided by three key objectives:

1. **Faithfulness:** The model aims to accurately reconstruct the original features for each stream from its latent representation. Specifically, the reconstruction $\hat{\mathbf{x}}^s$ should closely approximate the original input $\mathbf{x}^s$, ensuring the learned interpretable model captures the information of the upstream features.

2. **Interpretability:** The enforced sparsity ($\|\mathbf{z}^s\|_0 \ll L$) is designed to promote monosemanticity, where individual latent dimensions ideally capture distinct and human-understandable concepts, reducing semantic entanglement (Olah et al., 2020).

3. **Concept Alignment:** Ideally, the interpretation, or the underlying *concept* represented by each latent dimension $j \in \{1, \ldots, L\}$, should remain consistent across all input streams $s \in \mathcal{S}$. When dimension $j$ represents a certain concept when activated via stream $s$, it should represent the same concept when activated via stream $t$. This *concept alignment* leverages the shared origin of the input features $\{\mathbf{x}^s\}_{s \in \mathcal{S}}$, as they derive from the same underlying data sample.

The subsequent sections detail the SPARC architecture (Section 3.2) and training objective (Section 3.3) specifically designed to pursue these three objectives.

## 3.2 SPARC Architecture

The SPARC architecture processes multiple input streams through stream-specific encoders and decoders, coupled by a shared sparse activation mechanism that promotes concept alignment. An overview of SPARC is presented in Figure 2.

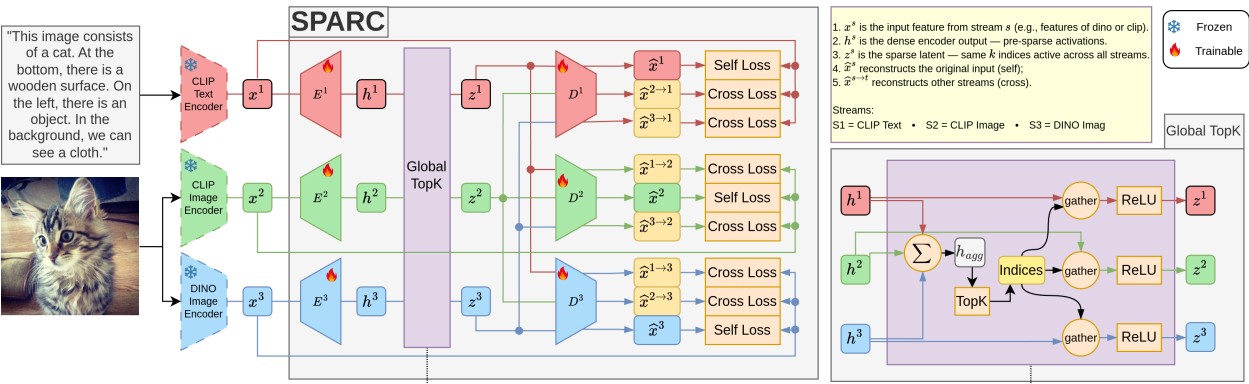

Figure 2: Detailed architecture of the SPARC model as well as the Global TopK mechanism.

**Stream Encoders.** Each input feature vector $\mathbf{x}^s \in \mathbb{R}^{d_s}$ is processed by its corresponding stream-specific encoder $E_s : \mathbb{R}^{d_s} \to \mathbb{R}^L$. The encoder performs an affine transformation to map the input features to the $L$-dimensional latent space, producing pre-activation logits $\mathbf{h}^s$. This involves centering the input using a learnable pre-bias $\mathbf{b}^s_{\mathrm{pre}} \in \mathbb{R}^{d_s}$, applying a linear transformation with weights $\mathbf{W}^s_E \in \mathbb{R}^{L \times d_s}$, and adding a learnable latent bias $\mathbf{b}^s_{\mathrm{lat}} \in \mathbb{R}^L$:

$$\mathbf{h}^s = E_s(\mathbf{x}^s) = \mathbf{W}^s_E(\mathbf{x}^s - \mathbf{b}^s_{\mathrm{pre}}) + \mathbf{b}^s_{\mathrm{lat}} \tag{1}$$

**Global TopK Sparse Activation.** SPARC employs a sparse activation mechanism to jointly satisfy Interpretability and Concept Alignment. Rather than applying sparsity independently to each stream's logits $\mathbf{h}^s$, we use a **Global TopK** approach that enforces shared feature activation across streams.

Logits are first aggregated across all streams:

$$\mathbf{h}_{\mathrm{agg}} = \sum_{s \in \mathcal{S}} \mathbf{h}^s \tag{2}$$

The TopK indices are then selected using this aggregated logit $h_{\mathrm{agg}}$:

$$\mathcal{I}_{\mathrm{global}} = \mathrm{TopK}(\mathbf{h}_{\mathrm{agg}}, k) \tag{3}$$

where $\mathrm{topk}(\cdot, k)$ returns the indices of the top $k$ elements.

This shared index set $\mathcal{I}_{\mathrm{global}}$ is then used to construct sparse latent representations for each stream. For every stream $s$, we select the corresponding values from $\mathbf{h}^s$ at the indices $\mathcal{I}_{\mathrm{global}}$, apply a ReLU activation, and zero out the rest:

$$\mathbf{z}^s = \mathrm{ReLU}(\mathrm{gather}(\mathbf{h}^s, \mathcal{I}_{\mathrm{global}})) \tag{4}$$

where $\mathrm{gather}(\mathbf{h}^s, \mathcal{I}_{\mathrm{global}})$ keeps values from $\mathbf{h}^s$ at positions in $\mathcal{I}_{\mathrm{global}}$ and zeros elsewhere. This ensures both sparsity ($\|\mathbf{z}^s\|_0 \le k$) and concept alignment by forcing identical latent features to activate across all streams

for the same underlying data sample, unlike standard per-stream TopK approaches where each stream selects $\text{topk}(\mathbf{h}^s, k)$ independently.

It is important to note that while Global TopK enforces a shared support (identical active indices) across streams, we maintain separate latent vectors $z^s$ rather than collapsing them into a single shared z. This design reflects that different feature spaces may require different activation magnitudes on the same latent dimension for faithful reconstruction and allows direct comparison of how strongly each model activates a given concept.

**Stream Decoders.** Finally, each stream-specific decoder $D_s : \mathbb{R}^L \to \mathbb{R}^{d_s}$ maps the sparse latent representation $\mathbf{z}^s$ back to the original feature space of stream $s$. The decoder applies a linear transformation using weights $\mathbf{W}_D^s \in \mathbb{R}^{d_s \times L}$ and adds back the stream's pre-bias $\mathbf{b}_{\text{pre}}^s$ (which was subtracted by the encoder) to produce the reconstruction $\hat{\mathbf{x}}^s$:

$$\hat{\mathbf{x}}^s = D_s(\mathbf{z}^s) = \mathbf{W}_D^s \mathbf{z}^s + \mathbf{b}_{\text{pre}}^s \tag{5}$$

We note that all encoder and decoder mappings are affine (linear transformation plus bias). This design follows common practice in sparse autoencoders and linear probes, where late-layer representations of large pretrained models such as CLIP and DINO have been shown to support strong linear readouts. Exploring shallow nonlinear mappings (e.g., two-layer MLPs) is a natural extension that we leave for future work.

### 3.3 Training Objective

The objective combines the two components with a weighting factor $\lambda$:

$$\mathcal{L}_{\text{total}} = \mathcal{L}_{\text{self}} + \lambda \mathcal{L}_{\text{cross}} = \sum_{s \in \mathcal{S}} \mathcal{L}_{\text{NMSE}}(\mathbf{x}^s, D_s(\mathbf{z}^s)) + \lambda \sum_{\substack{s,t \in \mathcal{S} \\ s \neq t}} \mathcal{L}_{\text{NMSE}}(\mathbf{x}^t, D_t(\mathbf{z}^s)) \tag{6}$$

Here, $D_s(\mathbf{z}^s)$ is the self-reconstruction of stream $s$, while $D_t(\mathbf{z}^s)$ is the cross-reconstruction of stream $t$'s input using the latent code from stream $s$.

While Global TopK activation structurally enforces shared activation patterns, the cross-reconstruction loss provides complementary semantic alignment that ensures the meaning of those activations is transferable between streams. This dual approach to concept alignment combines a hard structural constraint (which set of neurons to activate) with a soft semantic constraint (what information those neurons encode).

## 4 Evaluation

The goal of SPARC is interpretable neuron alignment where individual latent dimensions represent consistent and monosemantic concepts across different input streams. We measure SPARC quality through concept alignment analysis, reconstruction fidelity assessment, and probe loss evaluation. Throughout this section, we compare SPARC (Global TopK with cross-reconstruction) against two baselines: (1) Local TopK, which applies the TopK operation independently to each stream's logits $\mathbf{h}^s$ rather than to the aggregated logits, and (2) USAE (Thasarathan et al., 2025), a prior cross-model SAE method that uses random encoder selection during training. We also ablate the effect of the cross-reconstruction loss by varying $\lambda \in \{0, 1\}$.

### 4.1 Latent Activation Alignment

Figure 3 shows neuron 6463's top-activating images across four training configurations. In Local TopK with $\lambda = 1$ (top right), CLIP-text stream shows no activations (dead neuron) while DINO and CLIP-image streams activate on different image types. In contrast, Global TopK with $\lambda = 1$ (bottom right) shows consistent activations across all three streams for the same object type.

We find this pattern to be common where without Global TopK, one or two streams have dead latents while others remain active. We quantified this pattern across all 8,192 latent dimensions (Table 1). Local TopK with $\lambda = 1$ produces 48.8% mixed activation patterns where only 2/3 streams are active, creating alignment failures. Global TopK with $\lambda = 1$ achieves 84.4% all-alive neurons with consistent cross-stream

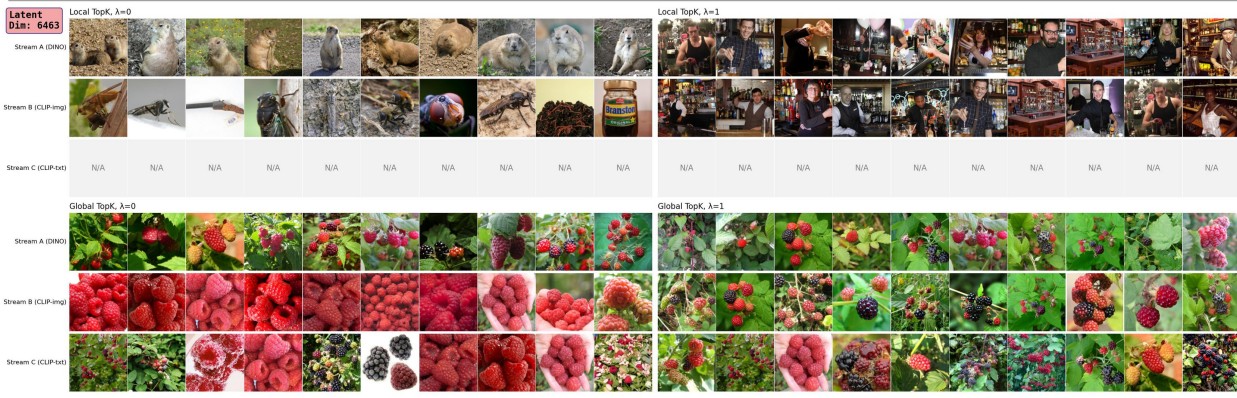

Figure 3: Top-activating samples for the latent dimension 6463 across three streams (DINO, CLIP-img, CLIP-txt) under four SPARC configurations. Each row shows top-10 images that activate the latent. The CLIP-text stream shows no activations under Local TopK with $\lambda = 1$ (top right) due to a dead neuron, demonstrating that cross-reconstruction loss alone is insufficient to ensure consistent activation patterns across streams. In contrast, Global TopK with $\lambda = 1$ (bottom right) shows consistent activations across all three streams for the same object type.

activation and 0% cases of partial activation. This means with Global TopK with $\lambda = 1$, all neurons are either active across all streams or dead across all streams. We also report per-stream percentages of dead neurons, showing that with Local TopK with $\lambda = 1$, almost half of the CLIP-text neurons are dead. With Global TopK with $\lambda = 1$, dead neurons are distributed equally across all streams.

As an external baseline, USAE shows only 45.3% all-alive neurons, with the majority of latents in mixed 1/3 or 2/3 patterns (17.7% and 33.6%) and markedly uneven dead-neuron rates across streams (31.0% for CI, 39.1% for CT, and 9.3% for D), indicating substantially lower cross-stream activation consistency and highly imbalanced dead-neuron distribution.

Table 1: Neuron activation patterns and stream-specific dead neuron rates across 8192 latent dimensions. Mixed patterns indicate partial cross-stream alignment where only 1/3 or 2/3 of the streams are active for the same latent. CI = CLIP-image, CT = CLIP-text, D = DINO.

| Method | TopK | $\lambda$ | Activation Patterns | | | | Dead Neurons | | |
|---|---|---|---|---|---|---|---|---|---|
| | | | All Dead | 1/3 | 2/3 | All Alive | CI | CT | D |
| **SPARC (ours)** | Global | 1.0 | 15.6% | 0.0% | 0.0% | 84.4% | 15.6% | 15.6% | 15.6% |
| Ablation | Global | 0.0 | 1.6% | 0.0% | 0.2% | 98.2% | 1.6% | 1.8% | 1.6% |
| Ablation | Local | 1.0 | 0.0% | 7.6% | 48.8% | 43.6% | 16.5% | 45.0% | 2.4% |
| Ablation | Local | 0.0 | 0.0% | 0.0% | 14.3% | 85.7% | 0.3% | 14.1% | 0.0% |
| USAE | – | – | 3.5% | 17.7% | 33.6% | 45.3% | 31.0% | 39.1% | 9.3% |

Additional latent visualizations are provided in Appendix D.

## 4.2 Concept Alignment

To evaluate concept consistency, we test if a single latent dimension represents the same high-level concept across all input streams. Although assigning discrete labels to neurons has inherent limitations—as monosemantic features may not align perfectly with dataset labels—this analysis provides a systematic framework for measuring concept alignment.

Our procedure first constructs a quantitative proxy for each latent's meaning. For every latent dimension $\ell$ and stream $s$, we create a "concept profile" by aggregating the ground-truth labels from the 50 test images

that yield the highest activations ($z_\ell^s$). This produces a label frequency vector for each (latent, stream) pair. We then measure the alignment between two streams ($s_i, s_j$) for the same latent by computing the generalized Jaccard similarity of their concept profiles, **a** and **b**:

$$J(\ell; s_i, s_j) = \frac{\sum_c \min(a_c, b_c)}{\sum_c \max(a_c, b_c)}$$

In this formula, $a_c$ and $b_c$ are the frequency counts for a concept $c$ within each profile. The resulting score quantifies the conceptual overlap, ranging from 0 (no overlap) to 1 (identical concept profiles).

Table 2: Mean Jaccard similarity of concept profiles using fine-grained labels at hierarchy depth 5 of Open Images. We report mean $\mu$ and standard error (SE) over 8,192 latents.
The full SPARC model (Global TopK, $\lambda = 1$) shows substantially higher concept alignment within this evaluation framework.

| Method | TopK | $\lambda$ | Jaccard ($\mu \pm$ SE) |
|---|---|---|---|
| **SPARC (ours)** | Global | 1.0 | **0.8018 $\pm$ 0.0039** |
| Ablation | Global | 0.0 | 0.7344 $\pm$ 0.0015 |
| Ablation | Local | 1.0 | 0.2599 $\pm$ 0.0026 |
| Ablation | Local | 0.0 | 0.1651 $\pm$ 0.0010 |
| USAE | – | – | 0.2166 $\pm$ 0.0024 |

Table 2 reports the mean Jaccard similarity averaged over all latents and stream pairs. The results show a stark contrast: the full SPARC model (Global TopK, $\lambda = 1$) achieves an alignment score of **0.80**, whereas ablations perform poorly. Using Local TopK drops the score to 0.26, and removing the cross-reconstruction loss ($\lambda = 0$) degrades it further. For comparison, USAE attains a Jaccard score of 0.22, closely matching our Local TopK regimes and remaining far below SPARC with Global TopK and cross-reconstruction. This provides strong quantitative evidence that both Global TopK and cross-reconstruction are critical for learning a robustly aligned concept space.

### 4.3 Label Purity: Interpretability Comparison with USAE

Beyond cross-stream alignment, we also ask whether individual latent dimensions concentrate their strongest activations on semantically coherent image sets. Following a simple label-purity protocol, we measure, for each latent and stream, how often the most frequent Open Images label appears among the top-$N$ most-activating examples for that latent.

Concretely, for each latent $\ell$ and stream $s$, we: (i) rank test images by activation magnitude $z_\ell^s$; (ii) take the top-$N$ examples (we vary $N \in \{10, 25, 50, 100\}$ and also consider using *all* samples); (iii) compute the fraction of these images that contain the most frequent Open Images class label in that subset. Because Open Images is multi-label, we normalize by the number of images, not the number of label instances. We then average this fraction over all active latents to obtain a mean label-purity score and report mean $\pm$ standard error (SE).

Table 3 compares SPARC (Global TopK, $\lambda = 1$) against USAE, using the official USAE implementation adapted to our Open Images feature pipeline. Across all choices of $N$ and all three streams, SPARC consistently achieves higher label purity, indicating that its latents not only align across models but also respond more selectively to coherent semantic concepts within each model.

While label purity is a useful sanity check, it is also limited by the quality and granularity of dataset labels. Open Images annotations are multi-label, incomplete, and not guaranteed to align cleanly with the monosemantic features the SAE discovers, so high purity does not imply a perfect concept neuron, and low purity does not prove the opposite. We therefore treat Table 3 primarily as a *relative* comparison between SPARC and USAE under the same labeling noise, rather than as an absolute measure of interpretability.

Table 3: Label purity on Open Images for SPARC (Global TopK, $\lambda = 1$) vs. USAE. Entries are mean $\pm$ standard error over active latents.

| # Top Images $N$ | Stream | SPARC (Global+Cross) | USAE |
|---|---|---|---|
| 10 | CLIP-Image | $0.7950 \pm 0.0025$ | $0.6014 \pm 0.0032$ |
| | CLIP-Text | $0.7256 \pm 0.0026$ | $0.5479 \pm 0.0035$ |
| | DINO | $0.7960 \pm 0.0024$ | $0.6426 \pm 0.0029$ |
| 25 | CLIP-Image | $0.7416 \pm 0.0026$ | $0.5525 \pm 0.0032$ |
| | CLIP-Text | $0.6825 \pm 0.0025$ | $0.4941 \pm 0.0035$ |
| | DINO | $0.7403 \pm 0.0025$ | $0.5862 \pm 0.0029$ |
| 50 | CLIP-Image | $0.6950 \pm 0.0026$ | $0.5235 \pm 0.0033$ |
| | CLIP-Text | $0.6495 \pm 0.0025$ | $0.4681 \pm 0.0035$ |
| | DINO | $0.6892 \pm 0.0026$ | $0.5427 \pm 0.0029$ |
| 100 | CLIP-Image | $0.6395 \pm 0.0026$ | $0.4947 \pm 0.0032$ |
| | CLIP-Text | $0.6121 \pm 0.0025$ | $0.4468 \pm 0.0034$ |
| | DINO | $0.6298 \pm 0.0025$ | $0.4960 \pm 0.0029$ |
| All | CLIP-Image | $0.5300 \pm 0.0024$ | $0.4203 \pm 0.0032$ |
| | CLIP-Text | $0.5256 \pm 0.0024$ | $0.4018 \pm 0.0032$ |
| | DINO | $0.5275 \pm 0.0024$ | $0.4201 \pm 0.0028$ |

## 4.4  $R^2$ Score for Cross-Model and Cross-Modal reconstruction

We quantify reconstruction quality with the coefficient of determination $R^2$ Wright (1921) for each ordered pair of streams $(s \to t)$. Given SAE codes $z_i^s$ from stream $s$ and target decoder $D_t$, the reconstruction of sample $i$ is $\hat{\mathbf{x}}_i^{s \to t} = D_t(z_i^s)$. Let $\bar{\mathbf{x}}^t$ be the mean of the target features $\{\mathbf{x}_i^t\}_{i \in \mathcal{E}}$ over the evaluation set $\mathcal{E}$, i.e., $\bar{\mathbf{x}}^t = \frac{1}{|\mathcal{E}|} \sum_{i \in \mathcal{E}} \mathbf{x}_i^t$. Then $R_{s \to t}^2$ is defined as:

$$R_{s \to t}^2 = 1 - \frac{\sum_{i \in \mathcal{E}} \left\| \mathbf{x}_i^t - \hat{\mathbf{x}}_i^{s \to t} \right\|_2^2}{\sum_{i \in \mathcal{E}} \left\| \mathbf{x}_i^t - \bar{\mathbf{x}}^t \right\|_2^2}.$$

where $R_{s \to t}^2$ measures the fraction of the target stream's variance explained by the reconstruction. A value of $R_{s \to t}^2 = 1$ indicates perfect reconstruction, $R_{s \to t}^2 = 0$ matches the mean predictor, and $R_{s \to t}^2 < 0$ is worse than predicting the mean.

Table 4 reports $R_{s \to t}^2$ for self-reconstruction (diagonal terms) and cross-reconstruction (off-diagonal terms), with rows as targets and columns as sources. Across the three streams (CLIP-image, CLIP-text, DINO), **Global TopK** yields consistently positive cross-stream $R_{s \to t}^2$ values (0.407–0.559) and balanced self-reconstruction scores (0.663/0.725/0.690). In contrast, **Local TopK** increases CLIP self $R_{s \to t}^2$ (0.732/0.874) but fails on cross-stream transfer, especially with DINO as target, where the values are negative ($-0.391$ from CLIP-image, $-4.485$ from CLIP-text) and the DINO self term is near the mean baseline (0.069). For comparison, **USAE** achieves moderate CLIP self- and cross-reconstruction (self $R_{s \to t}^2$ of 0.506/0.616 and cross terms in the 0.389–0.474 range between CLIP-image and CLIP-text), but performs poorly on DINO: DINO self-reconstruction is weak (0.111), and cross-stream $R_{s \to t}^2$ into DINO is near-zero or slightly negative (0.017 from CLIP-image, $-0.005$ from CLIP-text).

Notably, when DINO is the source and CLIP is the target, $R_{s \to t}^2$ remains positive (0.456/0.304), indicating that cross-reconstruction into CLIP retains usable signal, whereas the reverse direction is weak. This asymmetry could reflect differences in encoder/decoder capacity, feature compatibility, or training dynamics that preferentially aid the DINO$\to$CLIP pathway; we do not attempt to disentangle these factors here. Empirically, Global TopK improves both directions and yields stable cross-stream and self-reconstruction, outperforming both Local TopK and USAE on this three-stream setup.

Table 4: $R^2$ (variance-weighted; higher is better) for three streams. Rows are targets ($t$), columns are sources ($s$). Diagonals (self) in bold; negative values indicate worse than predicting the mean.

| Original Target | **Global TopK** (source) | | | **Local TopK** (source) | | | **USAE** (source) | | |
|---|---|---|---|---|---|---|---|---|---|
| | clip_img | clip_txt | dino | clip_img | clip_txt | dino | clip_img | clip_txt | dino |
| clip_img | **0.663** | 0.513 | 0.556 | **0.732** | 0.234 | 0.456 | **0.506** | 0.389 | 0.421 |
| clip_txt | 0.526 | **0.725** | 0.519 | 0.340 | **0.874** | 0.304 | 0.474 | **0.616** | 0.454 |
| dino | 0.559 | 0.407 | **0.690** | -0.391 | -4.485 | **0.069** | 0.017 | -0.005 | **0.111** |
| *means* | self = 0.693, cross = 0.513 | | | self = 0.558, cross = −0.590 | | | self = 0.411, cross = 0.292 | | |

The three-stream configuration used throughout this section enables detailed analysis of concept alignment and interpretability across vision and language modalities. Appendix C extends this evaluation to ten encoders spanning diverse architectures (including SigLIP, ViT, Swin, E5, GTE, and Qwen), demonstrating that Global TopK maintains stable cross-stream reconstruction across a broader range of models.

### 4.5 How well can 1d probes recover known concepts?

Following prior work Gao et al. (2025); Gurnee et al. (2023), we assess whether individual latent dimensions in SPARC encode recognizable semantic features. Specifically, we use the Open Images test set, which includes 601 binary classification labels. We filter to retain only those labels with at least 50 positive samples, resulting in 432 binary tasks.

For each task, we apply a 1D logistic probe over individual latent dimensions using the sparse values. The probe parameters are optimized to minimize cross-entropy loss:

$$\min_{i,w,b} \mathbb{E}\left[y \log \sigma(wz_i + b) + (1 - y) \log(1 - \sigma(wz_i + b))\right],$$

where $z_i$ is the $i$-th latent dimension and $\sigma$ is the sigmoid function. We report the best-performing dimension (i.e., the one with the lowest loss) per task and average this across all tasks for each configuration and stream, following the method used in Gao et al. (2025).

Table 5: Probe loss (lower is better) across 432 Open Images binary classification tasks. Values are mean ± standard error over tasks.

| Method | TopK Type | $\lambda$ | CLIP-Image | CLIP-Text | DINO |
|---|---|---|---|---|---|
| SPARC | Global | 1.0 | $0.5354 \pm 0.0049$ | $0.5646 \pm 0.0042$ | $0.5409 \pm 0.0049$ |
| Ablation | Global | 0.0 | $0.5336 \pm 0.0049$ | $0.4942 \pm 0.0057$ | $0.5194 \pm 0.0058$ |
| Ablation | Local | 1.0 | $0.4990 \pm 0.0056$ | $0.5363 \pm 0.0053$ | $0.5170 \pm 0.0053$ |
| Ablation | Local | 0.0 | $0.5238 \pm 0.0052$ | $0.4904 \pm 0.0054$ | $0.5265 \pm 0.0062$ |
| USAE | – | – | $0.6372 \pm 0.0028$ | $0.6470 \pm 0.0025$ | $0.6494 \pm 0.0023$ |

All rows in Table 5 report average probe losses below 0.69, which is the expected loss for a random binary classifier on balanced labels (i.e., when predictions are always 0.5 and labels are equally likely to be 0 or 1, the expected binary cross-entropy is $-\log(0.5) = \log(2) \approx 0.693$). For both Global and Local TopK variants, probe losses tend to be lower when the cross-loss weight $\lambda$ is set to zero. Compared under the same protocol, USAE yields substantially higher probe losses than any SPARC configuration across all three streams, indicating that its individual latents are less informative for recovering these labeled concepts.

This probing approach offers a simple and efficient way to assess whether certain labeled concepts are linearly recoverable from the latent space. However, it relies on several assumptions, including that labeled attributes correspond to isolated latent directions. Probe loss differences do not necessarily reflect alignment quality, interpretability, or general usefulness of the representations, and we therefore use these results primarily as a relative comparison between SPARC, its ablations, and USAE under identical conditions.

Full implementation details, including sampling strategy, candidate filtering, and classifier configuration, are provided in Appendix A.3.

## 4.6 Semantic Segmentation via SPARC's Aligned Latents

This section tests whether SPARC's concept-aligned latents can be used for semantic segmentation by serving as scalar targets in gradient-based attribution methods. Gradient-based attribution methods like relevancy maps (Chefer et al., 2021) and GradCAM (Selvaraju et al., 2017) require scalar targets to compute $\nabla A = \frac{\partial \text{target}}{\partial A}$ for attribution computation.

We find that SPARC's aligned latents can effectively serve as these scalar targets. While standard SAEs could also provide individual latent activations for attribution, SPARC's key advantage is alignment: the same latent indices (e.g., latent 279) represent the same concepts across different streams, enabling systematic cross-modal analysis. Additionally, SPARC enables cross-modal dot products like $\mathbf{z}^{\text{dino}} \cdot \mathbf{z}^{\text{clip\_txt}}$ that would be meaningless with separate SAEs due to incompatible latent spaces.

We explore two SPARC approaches and compare them against established baselines. First, we use concept-specific latent selections $\sum_{j \in \mathcal{J}} z_j^s$ as scalar targets, where $\mathcal{J}$ represents the set of latent indices relevant to the target concept. Second, we compute cross-modal similarities $\mathbf{z}^s \cdot \mathbf{z}^t$ in the aligned latent space as scalar targets, enabling text-guided attribution in vision-only models.

**Single Latent Attribution.** We examine cases where $\mathcal{J}$ contains a single latent dimension, using $z_j^s$ as scalar targets for attribution computation. Figure 4 demonstrates this using $z_{279}^s$ (where $\mathcal{J} = \{279\}$) as the scalar target. The saliency maps show this same latent dimension responding to cat-related features across all streams: in the image (concentrated regions around the cat) and in the text (highest relevance for "cat" token). We compare against CLIP similarity baseline, but use simplified prompts (e.g., "a cat") for CLIP since its cross-modal similarity naturally responds to all concepts mentioned in complex captions, while SPARC's concept-specific latent focuses solely on the target concept even when processing full captions.

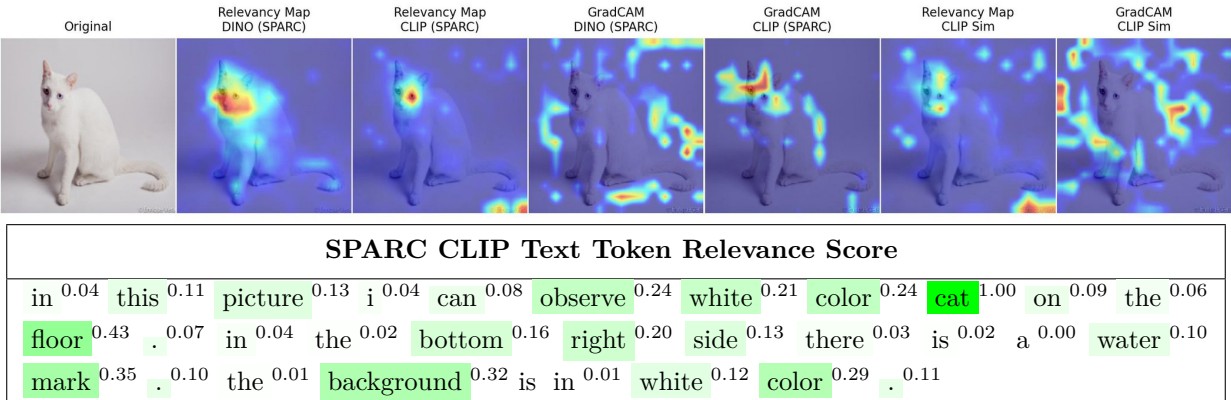

Figure 4: Individual latent attribution using SPARC dimension 279 vs. CLIP similarity baseline. (Above) Saliency maps show the same latent responding to cat-related features across image and text modalities. (Below) Text token relevance scores using SPARC and CLIP text.

Additional qualitative results for single latent attribution can be found in Appendix E.

**Cross-Modal Similarity Attribution.** We examine cross-modal dot products $\mathbf{z}^s \cdot \mathbf{z}^t$ computed in SPARC's aligned latent space as scalar targets. Figure 5 demonstrates this approach with $\mathbf{z}^{\text{dino}} \cdot \mathbf{z}^{\text{clip\_txt}}$ for DINO visualizations and $\mathbf{z}^{\text{clip\_img}} \cdot \mathbf{z}^{\text{clip\_txt}}$ for SPARC's CLIP implementation, compared against the baseline CLIP similarity $\mathbf{x}^{\text{clip\_img}} \cdot \mathbf{x}^{\text{clip\_txt}}$. This enables text-guided spatial attention in vision-only models.

Additional visualizations for cross-modal similarity attribution can be found in Appendix F.

**Quantitative Semantic Segmentation.** To quantify the heatmaps generated using SPARC's aligned latents, we conduct weakly supervised semantic segmentation on the MS COCO validation set (note that

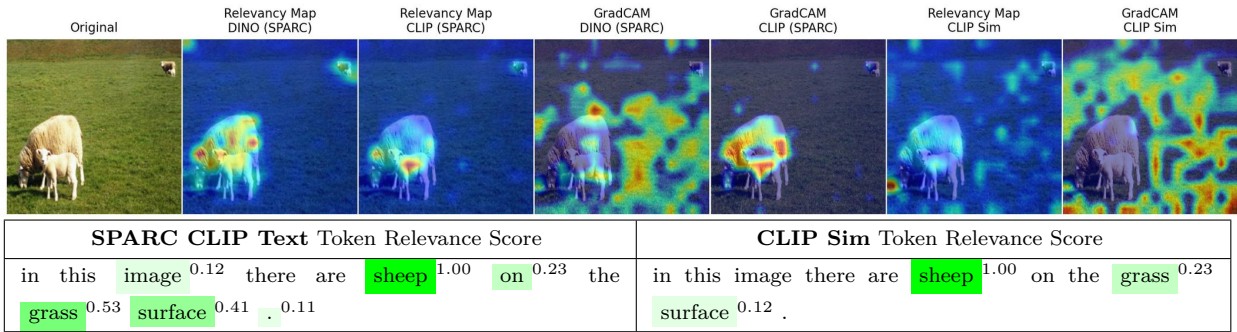

| SPARC CLIP Text Token Relevance Score | CLIP Sim Token Relevance Score |
|---|---|
| in this image $^{0.12}$ there are sheep $^{1.00}$ on $^{0.23}$ the grass $^{0.53}$ surface $^{0.41}$ . $^{0.11}$ | in this image there are sheep $^{1.00}$ on the grass $^{0.23}$ surface $^{0.12}$ . |

Figure 5: Cross-modal similarity attribution comparing SPARC's aligned latent space against CLIP similarity baseline. Both methods process the same image and caption, showing different attribution patterns.

the SPARC model was trained on the Open Images training set, not COCO). Since our text-based approach cannot distinguish between multiple instances of the same object class, we convert instance segmentation annotations to semantic segmentation by merging all instances of the same class into unified binary masks.

We follow the evaluation protocol of Chefer et al. (2021), using IoU threshold of 0.2 and excluding small objects. All methods use Chefer et al. (2021) for relevance map generation but differ in their similarity computation approaches. DETR uses attention maps from its encoder-decoder architecture, representing an upper bound of what Chefer et al. (2021) can achieve with a specialized object detection model. The CLIP baseline computes dot product similarity $\mathbf{x}^{\text{clip\_img}} \cdot \mathbf{x}^{\text{clip\_txt}}$ directly in the original feature space.

Table 6: Weakly supervised segmentation results on MS COCO comparing cross-modal similarity approaches. All methods use Chefer et al. (2021) for relevance map generation. SPARC methods compute similarities in the concept-aligned sparse latent space, while baselines and USAE operate in their respective feature/latent spaces . Evaluation follows the protocol from Table 1 of Chefer et al. (2021) with IoU threshold of 0.2.

| Method | Variant | TopK | AP | AP (M) | AP (L) | AR | AR (M) | AR (L) | mIoU |
|---|---|---|---|---|---|---|---|---|---|
| *Baselines* | | | | | | | | | |
| DETR | - | - | **0.467** | **0.526** | **0.539** | **0.644** | **0.762** | **0.682** | **0.305** |
| CLIP Similarity | - | - | 0.248 | 0.370 | 0.282 | 0.420 | 0.422 | 0.625 | 0.157 |
| *DINO Variants* | | | | | | | | | |
| Cross-Modal-Sim | USAE | - | 0.126 | 0.118 | 0.166 | 0.253 | 0.139 | 0.415 | 0.100 |
| Cross-Modal-Sim | SPARC | Local | 0.194 | 0.221 | 0.242 | 0.343 | 0.265 | 0.546 | 0.129 |
| Cross-Modal-Sim | SPARC | Global | **0.222** | 0.253 | **0.274** | **0.373** | 0.297 | **0.581** | **0.143** |
| Concept-Latent-Sum | USAE | - | 0.108 | 0.089 | 0.140 | 0.224 | 0.105 | 0.364 | 0.096 |
| Concept-Latent-Sum | SPARC | Local | 0.212 | 0.243 | 0.251 | 0.349 | 0.283 | 0.518 | 0.136 |
| Concept-Latent-Sum | SPARC | Global | **0.222** | 0.244 | 0.264 | 0.352 | 0.284 | 0.516 | 0.137 |
| *CLIP Variants* | | | | | | | | | |
| Cross-Modal-Sim | USAE | - | 0.004 | 0.019 | 0.004 | 0.022 | 0.021 | 0.029 | 0.023 |
| Cross-Modal-Sim | SPARC | Local | 0.176 | 0.242 | 0.204 | 0.304 | 0.266 | 0.472 | 0.102 |
| Cross-Modal-Sim | SPARC | Global | 0.223 | **0.283** | 0.262 | 0.369 | **0.319** | 0.569 | 0.138 |
| Concept-Latent-Sum | USAE | - | 0.006 | 0.027 | 0.007 | 0.027 | 0.028 | 0.036 | 0.035 |
| Concept-Latent-Sum | SPARC | Local | 0.203 | 0.238 | 0.234 | 0.325 | 0.258 | 0.474 | 0.123 |
| Concept-Latent-Sum | SPARC | Global | 0.222 | 0.276 | 0.254 | 0.347 | 0.310 | 0.493 | 0.131 |

Our SPARC methods implement the two approaches described earlier. **Cross-Modal-Sim** computes cross-modal similarities in the aligned sparse latent space: $\mathbf{z}^{\text{clip\_txt}} \cdot \mathbf{z}^{\text{dino}}$ for SPARC DINO and $\mathbf{z}^{\text{clip\_txt}} \cdot \mathbf{z}^{\text{clip\_img}}$ for SPARC CLIP. **Concept-Latent-Sum** uses concept-specific latent selections $\sum_{j \in \mathcal{J}} z_j^s$ where $\mathcal{J}$ contains latents that activate frequently ($\geq 50$ times) for the target class. For each image and target class, we use the class name as text input to generate the text representations. Table 6 presents the quantitative results.

The results reveal two key observations about SPARC's concept alignment. First, Global TopK consistently outperforms Local TopK across both backbones (DINO: 0.143 vs 0.129 mIoU; CLIP: 0.138 vs 0.102 mIoU), indicating that shared activation patterns produce more coherent cross-modal representations. Second, SPARC DINO Global achieves 0.143 mIoU compared to the CLIP baseline's 0.157 mIoU, demonstrating that text-based spatial localization through a vision-only backbone can approach the performance of natively cross-modal similarity computation when operating in SPARC's aligned latent space. Compared under the same evaluation, USAE variants obtain substantially lower AP and mIoU than all SPARC configurations (e.g., DINO-based USAE reaches only 0.096 mIoU and CLIP-based USAE 0.023–0.035 mIoU), indicating that their aligned latents are less effective as targets for weakly supervised segmentation in this setting.

## 5 Ablation Study

Unless noted otherwise, we train for 50 epochs with $L$=4096, $k$=64, $\lambda$=1, and $\eta$=$10^{-4}$. All numbers are mean NMSE over the three streams; dashed lines are Global TopK, solid lines Local TopK.

### 5.1 Number of latents $L$

Figure 6 shows the effect of scaling latent dimension $L \in \{2048, 4096, 8192, 16384\}$ while keeping sparsity level $k$ and TopK type fixed. For self-reconstruction, Local TopK improves in most cases with additional latents, while Global TopK exhibits higher reconstruction loss at each specific $k$. However, for cross-reconstruction, Global TopK demonstrates dramatic improvements when increasing the number of latents, whereas Local TopK shows much higher loss and actually degrades with additional latents. These patterns reveal that Global TopK's structural constraint comes at the cost of self-reconstruction but delivers substantial gains in cross-reconstruction.

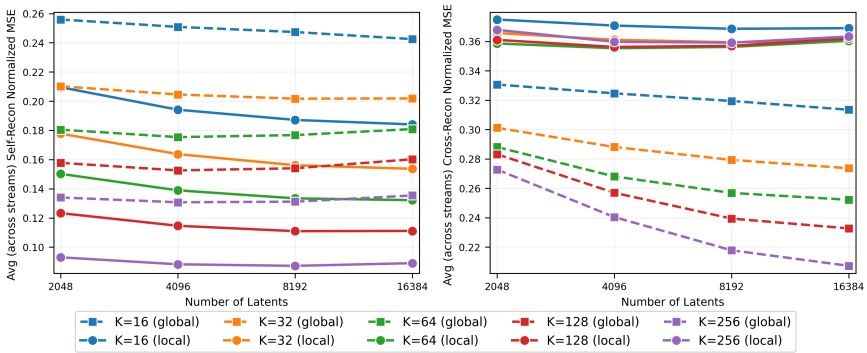

Figure 6: Self- and cross-reconstruction NMSE vs. number of latents.

This tradeoff is better quantified in Figure 7, which shows that Global TopK incurs a consistent self-reconstruction cost compared to Local TopK. The left panel reveals penalties of 0.030-0.060 NMSE across different configurations. However, the right panel demonstrates that these costs are more than compensated by substantial cross-reconstruction gains of 0.044-0.156 NMSE. For Global TopK, the cross-reconstruction benefits are consistently 2-3× larger than the self-reconstruction penalties, indicating that the Global constraint represents a favorable trade-off. This benefit ratio becomes even more pronounced when increasing the number of latents, particularly at higher sparsity levels where k=256 shows gains reaching 0.156 while costs remain below 0.047.

### 5.2 Sparsity $k$

Figure 8 demonstrates the expected trade-off between sparsity and reconstruction quality as $k$ varies from 16 to 256. Note that our goal is higher sparsity (lower $k$), making this analysis particularly relevant for understanding performance under our target operating conditions.

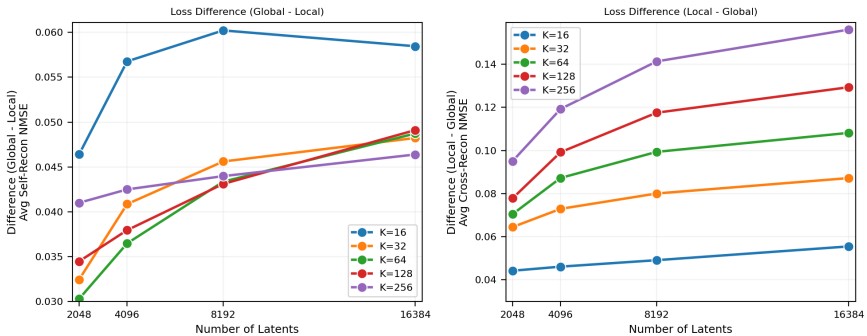

Figure 7: Global-vs-Local loss gap.

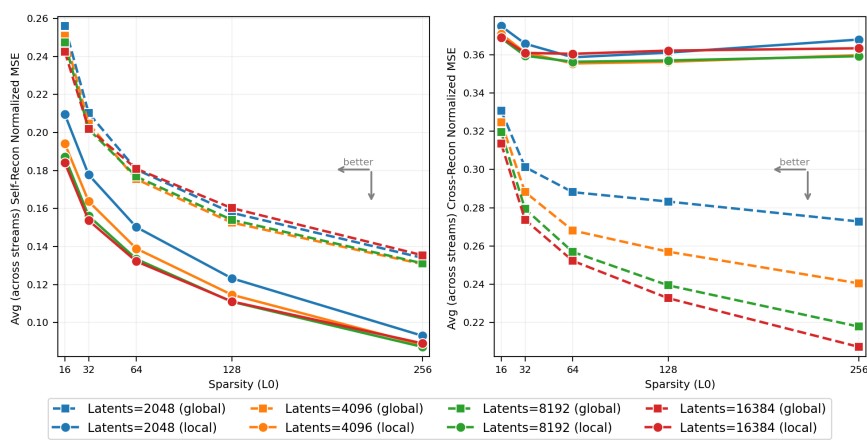

Figure 8: Effect of TopK sparsity.

For self-reconstruction, both Global and Local TopK show consistent improvement with higher $k$, as expected when more features are available. However, cross-reconstruction reveals a more complex pattern: Local TopK initially improves with higher $k$ but then degrades, while Global TopK shows consistent improvement across the entire range. This suggests that without the structured constraint of global activation selection, Local TopK struggles to maintain cross-stream alignment when more features are active, potentially due to increased difficulty in coordinating which features to activate across different streams.

We provide further ablation on the cross-loss weight $\lambda$ and learning rate $\eta$ in Appendix B.

## 6    Conclusion

We introduced **SPARC**, a sparse autoencoder that learns a single, shared latent space across heterogeneous representation streams. Its two core mechanisms, Global TopK sparsity and a cross-reconstruction loss, enforce consistent index selection and semantic alignment across modalities.

Without either mechanism, alignment is minimal: Local TopK without cross-loss achieves just **0.16** general Jaccard similarity on Open Images. In contrast, SPARC with both components achieves **0.80** general Jaccard. This shared space eliminates dead and mixed neurons across both models and modalities, and enables capabilities such as text-driven localization using vision-only models. Overall, this unified approach allows identifying shared concept in cross-model and cross-modal settings.

## Broader Impact

SPARC enables direct comparison of how different models represent semantic concepts, which has both beneficial and potentially harmful applications.

On the positive side, a unified concept space across models can support cross-model auditing, helping researchers identify what concepts different architectures have learned and whether they encode problematic biases. This capability may also aid model debugging and safety analysis by revealing shared failure modes or unintended representations across model families.

However, the same capabilities raise dual-use concerns. A shared concept space could be used to transfer harmful concepts between models or to localize sensitive content (e.g., identifying individuals or private information) across vision and language modalities. The text-guided localization demonstrated in this work, while useful for interpretability, could potentially be repurposed for surveillance or content targeting.

To mitigate these risks, we recommend avoiding training SPARC on sensitive or harmful target concepts, restricting deployment in high-risk domains without appropriate oversight, and coupling use of cross-model interpretability tools with governance frameworks and auditing practices. We hope that the interpretability benefits of SPARC outweigh these risks when deployed responsibly.

### Acknowledgments

This project was funded by Fonds de recherche du Québec – Nature et technologies (FRQNT) PBEEE scholarship [A.N]. This work was also partially supported by the Natural Sciences and Engineering Research Council of Canada (NSERC) RGPIN-2022-05378 [M.S.H], RGPIN-2025-06770 [H. R.], AWS AI Amazon Research Awards (ARA) [M.S.H], and FRQNT 361263 [H.R.].

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

# Appendix

## A Experimental Details

### A.1 Datasets

We use the dense-annotated subset of Open Images V7, containing 1.9M images (1.7M train, 41k val, 125k test), for all training and evaluation of SPARC. This subset includes bounding boxes, segmentation masks, and image-level labels. MS-COCO 2017 (118k train, 5k val) is used only for downstream segmentation and retrieval experiments.

### A.2 Hyperparameters and Training Configuration

We provide complete hyperparameter settings for all SPARC experiments to ensure reproducibility.

**Model Architecture.** All experiments use a latent dimension of $L = 8{,}192$ with sparsity level $k = 64$. Input feature dimensions vary by dataset: for Open Images, DINOv2-ViT-L/14 features have dimension 1024 and CLIP-ViT-L-14 features have dimension 768; for MS-COCO, DINOv2-ViT-B/14 features have dimension 768 and CLIP-ViT-B/16 features have dimension 512.

**Training Hyperparameters.** We train all models for 50 epochs using a batch size of 256. The optimizer is Adam with learning rate $\eta = 10^{-4}$, $\beta_1 = 0.9$, $\beta_2 = 0.999$, and $\epsilon = 10^{-8}$. All experiments use a fixed random seed of 42 for reproducibility. Training data comprises 80% of the available samples (`train_ratio = 0.8`), with the remaining 20% reserved for validation.

**Loss Function Configuration.** The total loss combines self-reconstruction, cross-reconstruction, and auxiliary reconstruction terms. For cross-loss experiments, we set the cross-reconstruction coefficient $\lambda = 1.0$, while no-cross experiments use $\lambda = 0.0$.

We implement an auxiliary loss (AuxK) identical to Gao et al. (2025), which targets dead neurons to prevent dormant latent dimensions. Given the main reconstruction error $\mathbf{e} = \mathbf{x} - \hat{\mathbf{x}}$, the auxiliary loss reconstructs this residual using the top-$k'$ dead latents: $\mathcal{L}_{\text{aux}} = \|\mathbf{e} - \hat{\mathbf{e}}\|_2^2$, where $\hat{\mathbf{e}} = \mathbf{W}_D^s \mathbf{z}_{\text{aux}}$. The key difference between Local and Global TopK variants is the source of the residual: Local TopK computes residuals from stream-specific reconstructions, while Global TopK computes residuals from reconstructions using globally shared indices. We use auxiliary sparsity level $k' = 64$ and coefficient $\gamma = 0.03125$ (1/32), with neurons considered dead after 1000 inactive steps.

**Weight Initialization.** Encoder and decoder weights are initialized with tied weights: $\mathbf{W}_D^s = (\mathbf{W}_E^s)^T$. Decoder weights are unit-normalized column-wise and maintained through gradient adjustments that project out components parallel to unit vectors. All bias parameters are initialized to zero. Dead neurons are reinitialized using Gaussian noise with a standard deviation of 0.01.

**Dead Neuron Management.** Latent dimensions with activation frequency below $10^{-3}$ (`auxk_threshold`) for more than 1000 consecutive training steps (`dead_steps_threshold`) are considered dead and reinitialized. This prevents the emergence of inactive dimensions during training while the auxiliary loss encourages their reactivation.

**Data Loading Optimization.** To address HDF5 loading efficiency, we implement a custom `ContiguousRandomBatchSampler` that shuffles at the batch level rather than sample level, reducing training time from hours to minutes per epoch while maintaining training dynamics. DataLoaders use 4 workers with memory pinning enabled.

**Feature Model Specifications.** For Open Images experiments, we extract features using DINOv2-ViT-L/14 (with registers variant) and CLIP-ViT-L-14 trained on DataComp-1B. For MS-COCO experiments, we use DINOv2-ViT-B/14 (with registers variant) and CLIP-ViT-B/16 trained on DataComp-XL dataset (`datacomp_xl_s13b_b90k`).

**Experimental Configurations.** We evaluate four training configurations across two datasets, resulting in eight total experimental conditions: Global/Local TopK activation $\times$ Cross-loss/No Cross-loss $\times$ Open

Images/MS-COCO datasets. The Global TopK variant aggregates logits across streams before index selection, while Local TopK applies independent TopK selection per stream.

## A.3 Probe Implementation Details

We evaluate concept recoverability using 1D logistic probes following the methodology of Gao et al. (2025). This section details the complete experimental procedure for the Open Images binary classification evaluation reported in Section 4.5.

**Dataset and Task Selection.** We use the Open Images test set containing 112,699 samples with 601 available binary classification labels. To ensure statistical reliability, we filter tasks to retain only those with at least 50 positive examples, resulting in 432 binary classification tasks used in our evaluation.

**Data Preprocessing and Balancing.** For each binary classification task, we address class imbalance by randomly sampling negative examples to match the number of positive examples, creating balanced datasets. Latent activations are binarized using a threshold of zero: sparse representations $\mathbf{z}^s$ are converted to binary indicators $(z_i^s > 0)$ for probe training.

**Data Splitting Strategy.** We employ a stratified three-way split with 70% training, 15% validation, and 15% test samples from the balanced dataset. Stratification ensures both positive and negative classes are represented proportionally across all splits. We use task-specific random seeds $(1000 + \texttt{task\_id}, 2000 + \texttt{task\_id}, 3000 + \texttt{task\_id})$ for reproducible splits across experiments.

**Candidate Latent Selection.** Since our goal is to test whether individual latent dimensions can recover semantic concepts, we focus the evaluation on the most promising candidates rather than all 8,192 latent dimensions. We first exclude dimensions that show zero activation during training, then rank remaining candidates by counting how many positive training examples activate each dimension. For each task, we select the 20 most frequently activated dimensions as candidates, reducing computational cost while focusing on latents most likely to encode the target concept.

**Probe Training Configuration.** For each candidate latent dimension, we train a 1D logistic regression probe with the following hyperparameters: L-BFGS solver, maximum 200 iterations, L2 regularization with default strength $(C = 1.0)$, single latent activation value $z_i^s$ as input, and binary class labels as targets.

**Model Selection and Evaluation.** We select the best-performing latent dimension for each task based on validation set performance. For each of the 20 candidate dimensions, we train individual probes, evaluate on the validation set using binary cross-entropy loss, select the dimension achieving the lowest validation loss, and report final performance on the held-out test set.

**Evaluation Metrics.** We report mean binary cross-entropy loss across all 432 tasks for each stream and training configuration. As a baseline reference, random binary classification on balanced data yields an expected loss of $-\log(0.5) \approx 0.693$.

**Experimental Scope.** We evaluate four SPARC training configurations (Global/Local TopK $\times$ Cross-loss/No Cross-loss) across three feature streams (DINO, CLIP-image, CLIP-text), resulting in 12 total experimental conditions. Each condition processes all 432 binary classification tasks using the identical probe methodology described above.

## A.4 Computational Cost

This section summarizes the computational profile of SPARC and compares it to Universal Sparse Autoencoders (USAE) (Thasarathan et al., 2025).

**Asymptotic cost of the SPARC objective.** For $M$ streams, SPARC computes stream-specific logits $\{\mathbf{h}^s\}_{s \in \mathcal{S}}$, aggregates them into $\mathbf{h}_{\text{agg}}$, applies Global TopK once to obtain the shared index set, and then

evaluates both self- and cross-reconstruction terms:

$$\mathcal{L}_{\text{self}} = \sum_{s \in \mathcal{S}} \mathcal{L}_{\text{NMSE}}\big(\mathbf{x}^s, D_s(\mathbf{z}^s)\big), \qquad \mathcal{L}_{\text{cross}} = \sum_{\substack{s,t \in \mathcal{S} \\ s \neq t}} \mathcal{L}_{\text{NMSE}}\big(\mathbf{x}^t, D_t(\mathbf{z}^s)\big).$$

Each encoder or decoder is a dense linear layer between $\mathbb{R}^{d_s}$ and $\mathbb{R}^L$, so for batch size $B$, a single forward pass costs $O(BLd_s)$. Assuming comparable feature dimensions $d_s \approx d$ and treating $B$ as fixed, a single minibatch therefore performs: - $M$ encoder passes, costing $O(MLd)$, and - $M^2$ decoder passes across all ordered pairs $(s,t)$, costing $O(M^2Ld)$.

Thus, the dominant term for SPARC training scales as $O(M^2Ld)$ per batch.

By contrast, USAE (Thasarathan et al., 2025) encodes activations from a single randomly selected model $i$ at each step and decodes into all $M$ models: - one encoder pass, $O(Ld)$, and - $M$ decoder passes, $O(MLd)$, for an overall cost of $O(MLd)$ per batch. In other words, SPARC pays an additional factor of roughly $M$ in decoder work compared to USAE's random-encoder scheme. For the regimes we consider ($M \leq 10$), this is a modest constant factor rather than a prohibitive blow-up.

**Cached backbone features.** In all experiments, we follow standard SAE practice and treat upstream encoders (e.g., CLIP, DINO, SigLIP, ViT) as frozen feature extractors. Features from each stream are computed once on the training split and stored; SPARC training then operates entirely on these cached activations, optimizing only the linear encoder/decoder layers in the shared latent space. USAE is also trained on pretrained activations rather than updating the backbone models (Thasarathan et al., 2025), so both approaches incur a comparable one-off cost to run the dataset through the backbone encoders. We therefore focus our quantitative reporting on the SAE training itself and do not attempt to micro-benchmark the feature-extraction pass, which is dominated by the backbone architectures and hardware configuration.

**Wall-clock runtimes.** With cached features, optimizing SPARC is cheap in absolute terms. On a single H100 GPU, our main three-stream configuration on Open Images (CLIP-image/CLIP-text/DINO, $L$=8192, $k$=64, batch size 256, 50 epochs) takes under 25 minutes end-to-end (i.e., less than 30 seconds per epoch). The ten-stream configuration used in Appendix C (CLIP-image, CLIP-text, SigLIP-image, SigLIP-text, DINOv2, ViT, Swin, E5, GTE, Qwen) takes about 2 hours for 50 epochs, corresponding to roughly 2.5 minutes per epoch.

For context, USAE (Thasarathan et al., 2025) reports that training their model on ImageNet on a single Nvidia RTX 6000 GPU takes approximately three days (Appendix A.1). The datasets, architectures, and hardware are not strictly comparable, so this should be viewed only as a coarse reference point rather than a head-to-head benchmark. Nonetheless, it illustrates that the additional $O(M)$ factor in SPARC's objective does not translate into a large practical overhead once backbone features are cached, and that SPARC's wall-clock training cost is small relative to both feature extraction and existing universal SAE setups.

## A.5 USAE Baseline Configuration

We provide complete details for the Universal Sparse Autoencoder (USAE) baseline (Thasarathan et al., 2025) to ensure reproducibility.

**Implementation.** We used the authors' official implementation, specifically importing `TopKSAE` and `top_k_auxiliary_loss` from the `overcomplete` package in their GitHub repository. We adapted only the data loading pipeline to process our pre-extracted Open Images features in the three-stream configuration (CLIP-image, CLIP-text, DINO).

**Matched Hyperparameters.** To ensure a fair comparison, we matched key hyperparameters to our SPARC setup: dictionary size $L = 8{,}192$, sparsity level $k = 64$, batch size 256, and 50 training epochs. The optimizer is Adam with learning rate $\eta = 10^{-4}$, $\beta_1 = 0.9$, $\beta_2 = 0.999$. USAE was trained from scratch on the same pre-extracted frozen features used for SPARC (see Section A.2 for feature extraction details).

**USAE-Specific Parameters.** For parameters specific to the USAE architecture, we used the authors' defaults: auxiliary loss coefficient 0.1, gradient clipping at 1.0, linear encoder module, and L2 dictionary normalization.

**Training Procedure.** Following the original USAE training scheme, at each iteration we randomly select one stream $i$ as the encoder, compute latent codes $\mathbf{z}^i$, and reconstruct all streams using their respective decoders. This contrasts with SPARC, which encodes all streams simultaneously and applies Global TopK to the aggregated logits.

**Evaluation Protocol.** We evaluated USAE using identical metrics and procedures as SPARC: Jaccard-based concept alignment (Section 4.2), $R^2$ reconstruction (Section 4.4), activation consistency (Section 4.2), and label purity (Section 4.3). '

# B    Extra Ablation

This section details the ablation experiments used to select the optimal cross-loss weight $\lambda$ and learning rate $\eta$.

## B.1    Cross-loss weight $\lambda$

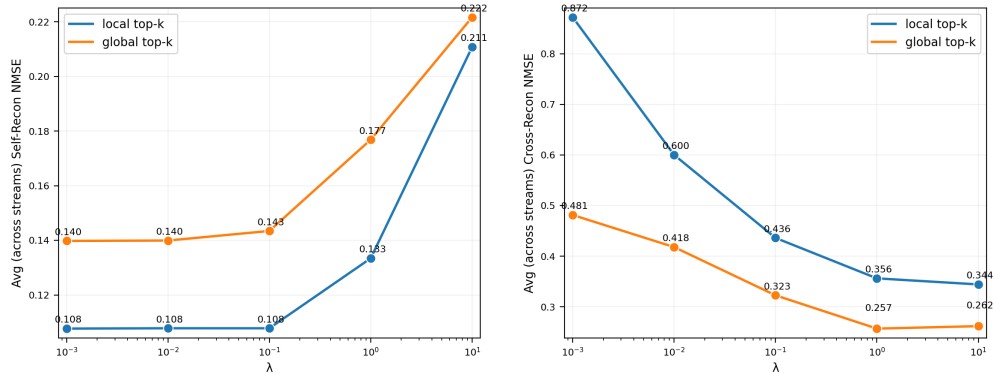

Figure 9: NMSE as a function of $\lambda$ (log scale).

Figure 9 shows the impact of $\lambda$ parameter. Cross-reconstruction drops quickly up to $\lambda \approx 1$. Beyond that point, the alignment gain saturates while self-reconstruction keeps rising, so we keep $\lambda=1$.

## B.2    Learning rate $\eta$

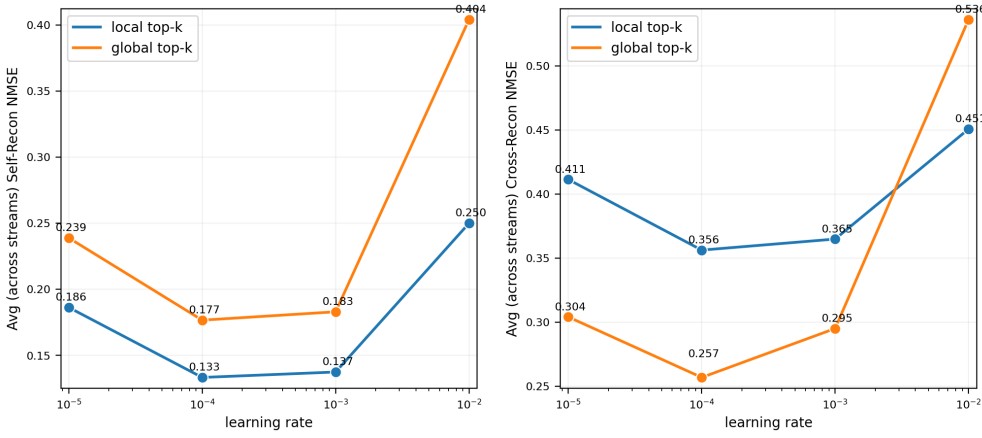

Figure 10: Learning-rate sweep.

The impact of learning rate is shown in the Figure 10. The learning rate $\eta=10^{-4}$ minimizes both losses. A smaller rate under-fits; $\eta \geq 10^{-3}$ destabilizes training and erases the Global advantage.

# C   Full $R^2_{s \to t}$ Reconstruction Evaluation

We report $R^2_{s \to t}$ matrices for more model combinations and training setups. Rows are targets ($t$), columns are sources ($s$). Negative values indicate worse than predicting the mean.

Table 7 summarizes the ten encoders used across the configurations below. Features are extracted as follows: for ViT we take the `[CLS]` token from the last hidden state; for Swin we use the model's pooled output; for DINOv2 we use the backbone's global representation returned by the hub model; for CLIP (image/text) we use OpenCLIP's `encode_image` / `encode_text`; for SigLIP2 (image/text) we use the HF `AutoProcessor` and `get_image_features` / `get_text_features` (captions lowercased, `padding=max_length`, `max_length=64`, `truncation=true`); for E5 we encode captions with a `"query:  "` prefix and normalized embeddings; for GTE and Qwen we use the default SentenceTransformers embeddings. Unless a model's own processor is used (SigLIP2), images are resized to 256, center-cropped to 224, converted to tensors, and normalized with ImageNet mean/std.

Table 7: Encoders used across configurations.

| Stream | Modality | Backbone / Model (brief) |
|---|---|---|
| `clip_img` Radford et al. (2021) | image | OpenCLIP CLIP-ViT-L/14 (DataComp) |
| `clip_txt` Radford et al. (2021) | text | OpenCLIP CLIP-ViT-L/14 (DataComp) |
| `siglip_img` Tschannen et al. (2025) | image | google/siglip2-so400m-patch14-384 (image) |
| `siglip_txt` Tschannen et al. (2025) | text | google/siglip2-so400m-patch14-384 (text) |
| `dino` Oquab et al. (2024) | image | DINOv2 ViT-L/14 (registers) |
| `vit` Dosovitskiy et al. (2021) | image | google/vit-large-patch16-224 (CLS) |
| `swin` Liu et al. (2021) | image | microsoft/swin-large-patch4-window12-384 (pooler) |
| `e5` Wang et al. (2022) | text | intfloat/e5-large-v2 (SentenceTransformers) |
| `gte` Li et al. (2023) | text | thenlper/gte-large (SentenceTransformers) |
| `qwen` Zhang et al. (2025) | text | Qwen/Qwen3-Embedding-0.6B (SentenceTransformers) |

- **CLIP:** This case focuses on CLIP, whose image and text encoders are natively aligned by training. Table 8 reports the results.

Table 8: $R^2$ (higher is better), CLIP image/text. Rows are targets ($t$), columns are sources ($s$).

| Original Target | Global TopK (source) | | Local TopK (source) | | USAE (source) | |
|---|---|---|---|---|---|---|
| | clip_img | clip_txt | clip_img | clip_txt | clip_img | clip_txt |
| clip_img | **0.657** | 0.574 | **0.737** | 0.239 | **0.577** | 0.398 |
| clip_txt | 0.676 | **0.789** | 0.333 | **0.882** | 0.495 | **0.708** |

- **SigLIP:** This case focuses on SigLIP, whose image and text encoders are also natively aligned by training. Table 9 reports the results.

Table 9: $R^2$ (higher is better), SigLIP image/text. Rows are targets ($t$), columns are sources ($s$).

| Original Target | Global TopK (source) | | Local TopK (source) | | USAE (source) | |
|---|---|---|---|---|---|---|
| | siglip_img | siglip_txt | siglip_img | siglip_txt | siglip_img | siglip_txt |
| siglip_img | **0.490** | 0.387 | **0.702** | 0.195 | **0.594** | 0.497 |
| siglip_txt | 0.547 | **0.704** | 0.261 | **0.831** | 0.516 | **0.657** |

- **Vision-only:** This case considers vision encoders trained without multimodal objectives. Table 10 reports the results.

Table 10: $R^2$ (higher is better), Vision-only. Rows are targets ($t$), columns are sources ($s$).

| Original Target | Global TopK (source) | | | Local TopK (source) | | | USAE (source) | | |
| --- | --- | --- | --- | --- | --- | --- | --- | --- | --- |
| | dino | swin | vit | dino | swin | vit | dino | swin | vit |
| dino | **0.716** | 0.579 | 0.533 | **0.482** | 0.335 | 0.255 | **0.177** | 0.061 | 0.040 |
| swin | -1.630 | **-1.574** | -1.854 | -13.691 | **-9.542** | -11.171 | 0.073 | **0.153** | 0.086 |
| vit | 0.562 | 0.589 | **0.671** | 0.442 | 0.519 | **0.669** | 0.137 | 0.171 | **0.259** |

- **Text-only:** This case considers text encoders trained without multimodal objectives (E5, GTE, Qwen). Table 11 reports the results.

Table 11: $R^2$ (higher is better), Text-only. Rows are targets ($t$), columns are sources ($s$).

| Original Target | Global TopK (source) | | | Local TopK (source) | | | USAE (source) | | |
| --- | --- | --- | --- | --- | --- | --- | --- | --- | --- |
| | e5 | gte | qwen | e5 | gte | qwen | e5 | gte | qwen |
| e5 | **0.805** | 0.752 | 0.725 | **0.854** | 0.778 | 0.775 | **0.834** | 0.828 | 0.826 |
| gte | 0.771 | **0.823** | 0.747 | 0.811 | **0.868** | 0.800 | 0.853 | **0.858** | 0.852 |
| qwen | 0.757 | 0.761 | **0.832** | 0.788 | 0.781 | **0.898** | 0.656 | 0.657 | **0.670** |

- **Image:** This case combines image encoders, some trained multimodally (CLIP-img, SigLIP-img) and others unimodally (DINO, Swin, ViT). Table 12 reports the results.

Table 12: $R^2$ (higher is better), Image. Rows are targets ($t$), columns are sources ($s$).

| Global TopK (source) | | | | |
| --- | --- | --- | --- | --- |
| Original Target | clip_img | dino | siglip_img | swin | vit |
| clip_img | **0.698** | 0.563 | 0.634 | 0.568 | 0.539 |
| dino | 0.525 | **0.678** | 0.544 | 0.536 | 0.497 |
| siglip_img | 0.569 | 0.495 | **0.653** | 0.501 | 0.467 |
| swin | 0.477 | 0.476 | 0.490 | **0.580** | 0.490 |
| vit | 0.523 | 0.525 | 0.532 | 0.568 | **0.651** |

| Local TopK (source) | | | | |
| --- | --- | --- | --- | --- |
| Original Target | clip_img | dino | siglip_img | swin | vit |
| clip_img | **0.746** | 0.474 | 0.591 | 0.494 | 0.429 |
| dino | 0.265 | **0.648** | 0.263 | 0.400 | 0.347 |
| siglip_img | 0.539 | 0.403 | **0.693** | 0.424 | 0.356 |
| swin | -8.475 | -5.250 | -5.427 | **-2.565** | -3.824 |
| vit | 0.437 | 0.444 | 0.429 | 0.522 | **0.688** |

| USAE (source) | | | | |
| --- | --- | --- | --- | --- |
| Original Target | clip_img | dino | siglip_img | swin | vit |
| clip_img | **0.427** | 0.372 | 0.397 | 0.380 | 0.371 |
| dino | 0.002 | **0.064** | 0.006 | 0.012 | 0.007 |
| siglip_img | 0.492 | 0.480 | **0.510** | 0.484 | 0.480 |
| swin | 0.065 | 0.066 | 0.068 | **0.097** | 0.070 |
| vit | 0.066 | 0.067 | 0.069 | 0.081 | **0.122** |

- **Text:** This case combines text encoders, some trained multimodally (CLIP-txt, SigLIP-txt) and others unimodally (E5, GTE, Qwen). Table 13 reports the results.

Table 13: $R^2$ (higher is better), Text. Rows are targets ($t$), columns are sources ($s$).

| **Global TopK (source)** | | | | | |
|---|---|---|---|---|---|
| **Original Target** | clip_txt | e5 | gte | qwen | siglip_txt |
| clip_txt | **0.813** | 0.721 | 0.720 | 0.717 | 0.705 |
| e5 | 0.672 | **0.784** | 0.735 | 0.710 | 0.641 |
| gte | 0.689 | 0.754 | **0.800** | 0.731 | 0.659 |
| qwen | 0.717 | 0.750 | 0.753 | **0.819** | 0.686 |
| siglip_txt | 0.598 | 0.583 | 0.585 | 0.572 | **0.743** |

| **Local TopK (source)** | | | | | |
|---|---|---|---|---|---|
| **Original Target** | clip_txt | e5 | gte | qwen | siglip_txt |
| clip_txt | **0.903** | 0.722 | 0.704 | 0.763 | 0.758 |
| e5 | 0.775 | **0.830** | 0.741 | 0.773 | 0.714 |
| gte | 0.792 | 0.777 | **0.837** | 0.796 | 0.737 |
| qwen | 0.815 | 0.763 | 0.752 | **0.898** | 0.754 |
| siglip_txt | 0.668 | 0.583 | 0.569 | 0.612 | **0.841** |

| **USAE (source)** | | | | | |
|---|---|---|---|---|---|
| **Original Target** | clip_txt | e5 | gte | qwen | siglip_txt |
| clip_txt | **0.497** | 0.476 | 0.473 | 0.475 | 0.474 |
| e5 | 0.809 | **0.816** | 0.813 | 0.811 | 0.809 |
| gte | 0.833 | 0.836 | **0.840** | 0.835 | 0.833 |
| qwen | 0.608 | 0.612 | 0.614 | **0.620** | 0.606 |
| siglip_txt | 0.517 | 0.516 | 0.513 | 0.513 | **0.535** |

- **All:** This case spans all ten streams, covering both unimodal and multimodally trained encoders across image and text. Table 14 reports the results. We can see that Global TopK is required to avoid the large negative numbers.

Table 14: $R^2$ (higher is better) for all streams. Rows are targets ($t$), columns are sources ($s$).

| Global TopK (source) | | | | | | | | | |
|---|---|---|---|---|---|---|---|---|---|
| **Original Target** | clip_img | clip_txt | dino | e5 | gte | qwen | siglip_img | siglip_txt | swin | vit |
| clip_img | **0.563** | 0.286 | 0.407 | 0.315 | 0.320 | 0.302 | 0.498 | 0.293 | 0.433 | 0.401 |
| clip_txt | 0.447 | **0.786** | 0.429 | 0.700 | 0.696 | 0.708 | 0.471 | 0.701 | 0.447 | 0.431 |
| dino | 0.241 | 0.123 | **0.376** | 0.135 | 0.139 | 0.133 | 0.246 | 0.126 | 0.259 | 0.239 |
| e5 | 0.356 | 0.643 | 0.322 | **0.715** | 0.680 | 0.674 | 0.395 | 0.612 | 0.356 | 0.327 |
| gte | 0.381 | 0.662 | 0.343 | 0.701 | **0.734** | 0.699 | 0.424 | 0.630 | 0.381 | 0.348 |
| qwen | 0.462 | 0.716 | 0.441 | 0.725 | 0.726 | **0.788** | 0.487 | 0.684 | 0.461 | 0.445 |
| siglip_img | 0.437 | 0.239 | 0.335 | 0.271 | 0.276 | 0.256 | **0.511** | 0.248 | 0.368 | 0.332 |
| siglip_txt | 0.330 | 0.590 | 0.308 | 0.568 | 0.566 | 0.566 | 0.357 | **0.704** | 0.329 | 0.315 |
| swin | 0.333 | 0.204 | 0.329 | 0.220 | 0.223 | 0.216 | 0.340 | 0.208 | **0.445** | 0.366 |
| vit | 0.357 | 0.245 | 0.360 | 0.260 | 0.263 | 0.256 | 0.363 | 0.248 | 0.413 | **0.492** |

| Local TopK (source) | | | | | | | | | |
|---|---|---|---|---|---|---|---|---|---|
| **Original Target** | clip_img | clip_txt | dino | e5 | gte | qwen | siglip_img | siglip_txt | swin | vit |
| clip_img | **0.750** | 0.212 | 0.397 | 0.186 | 0.186 | 0.207 | 0.550 | 0.204 | 0.441 | 0.382 |
| clip_txt | 0.344 | **0.880** | 0.301 | 0.667 | 0.658 | 0.730 | 0.367 | 0.719 | 0.339 | 0.306 |
| dino | -2.246 | -11.318 | **-0.408** | -7.649 | -7.961 | -8.587 | -1.393 | -13.086 | -1.524 | -1.237 |
| e5 | 0.243 | 0.706 | 0.190 | **0.808** | 0.706 | 0.737 | 0.287 | 0.654 | 0.227 | 0.196 |
| gte | 0.277 | 0.729 | 0.220 | 0.741 | **0.825** | 0.763 | 0.327 | 0.679 | 0.258 | 0.225 |
| qwen | 0.240 | 0.726 | 0.180 | 0.715 | 0.711 | **0.875** | 0.309 | 0.676 | 0.211 | 0.184 |
| siglip_img | 0.497 | 0.175 | 0.345 | 0.153 | 0.154 | 0.172 | **0.684** | 0.166 | 0.379 | 0.320 |
| siglip_txt | 0.238 | 0.621 | 0.198 | 0.531 | 0.527 | 0.572 | 0.263 | **0.817** | 0.233 | 0.204 |
| swin | -20.136 | -145.388 | -27.648 | -64.633 | -69.942 | -91.360 | -16.134 | -222.031 | **-9.140** | -17.424 |
| vit | 0.370 | -0.015 | 0.351 | 0.049 | 0.040 | 0.051 | 0.343 | -0.094 | 0.459 | **0.684** |

| USAE (source) | | | | | | | | | |
|---|---|---|---|---|---|---|---|---|---|
| **Original Target** | clip_img | clip_txt | dino | e5 | gte | qwen | siglip_img | siglip_txt | swin | vit |
| clip_img | **0.335** | 0.318 | 0.322 | 0.319 | 0.319 | 0.319 | 0.328 | 0.319 | 0.324 | 0.322 |
| clip_txt | 0.339 | **0.359** | 0.334 | 0.353 | 0.352 | 0.353 | 0.341 | 0.353 | 0.339 | 0.336 |
| dino | -0.002 | -0.005 | **0.011** | -0.004 | -0.004 | -0.004 | -0.001 | -0.004 | 0.000 | 0.000 |
| e5 | 0.790 | 0.792 | 0.790 | **0.794** | 0.793 | 0.792 | 0.790 | 0.792 | 0.790 | 0.790 |
| gte | 0.820 | 0.822 | 0.821 | 0.823 | **0.824** | 0.823 | 0.820 | 0.822 | 0.821 | 0.821 |
| qwen | 0.548 | 0.554 | 0.547 | 0.556 | 0.556 | **0.558** | 0.549 | 0.554 | 0.548 | 0.548 |
| siglip_img | 0.466 | 0.461 | 0.464 | 0.462 | 0.462 | 0.461 | **0.469** | 0.461 | 0.464 | 0.464 |
| siglip_txt | 0.444 | 0.451 | 0.443 | 0.451 | 0.451 | 0.450 | 0.446 | **0.455** | 0.445 | 0.444 |
| swin | 0.055 | 0.053 | 0.056 | 0.053 | 0.053 | 0.053 | 0.056 | 0.053 | **0.061** | 0.056 |
| vit | 0.028 | 0.025 | 0.029 | 0.025 | 0.025 | 0.025 | 0.029 | 0.025 | 0.031 | **0.039** |

# D   Latent Dimension Visualizations

This section provides examples of latent activation patterns across training configurations (Local/Global TopK $\times$ $\lambda$=0/1). Each figure shows the 2$\times$2 grid, with each configuration displaying top-10 activating images across three streams (DINO, CLIP-image, CLIP-text) for latent dimensions.

Figures 11, 12, and 13 demonstrate concept alignment under Global TopK with $\lambda = 1$, where latent dimensions exhibit consistent activation behavior across all streams, either fully active or completely dead across all three modalities. In contrast, Local TopK with $\lambda = 1$ shows mixed activation patterns, where latents may be active in some streams while remaining inactive in others.

For more results, check `https://github.com/AtlasAnalyticsLab/SPARC/blob/main/VISUALIZATIONS.md`.

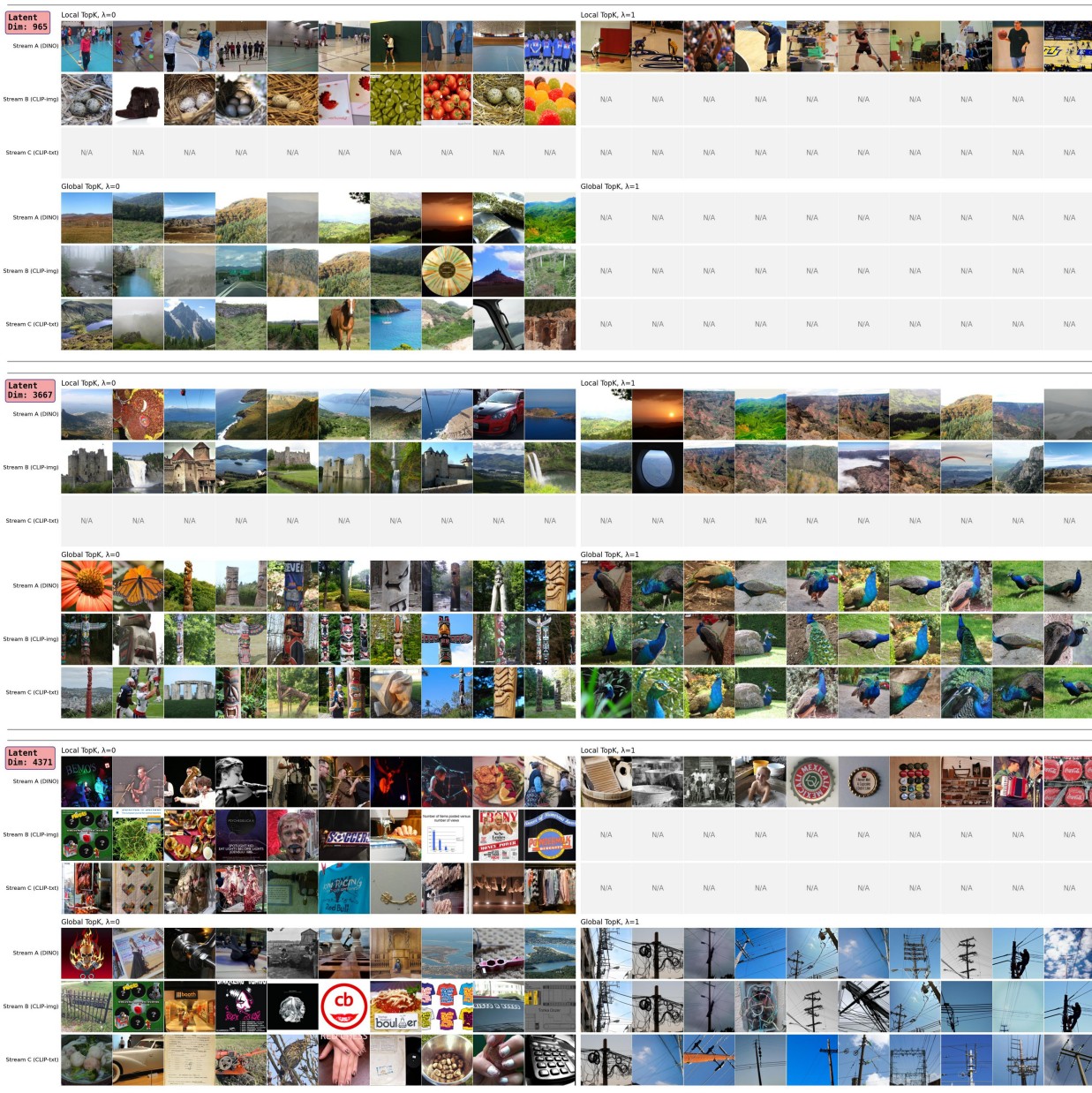

Figure 11: Latent activation examples for dimensions 965, 3667, and 4371 showing top-10 activating images across different configurations.

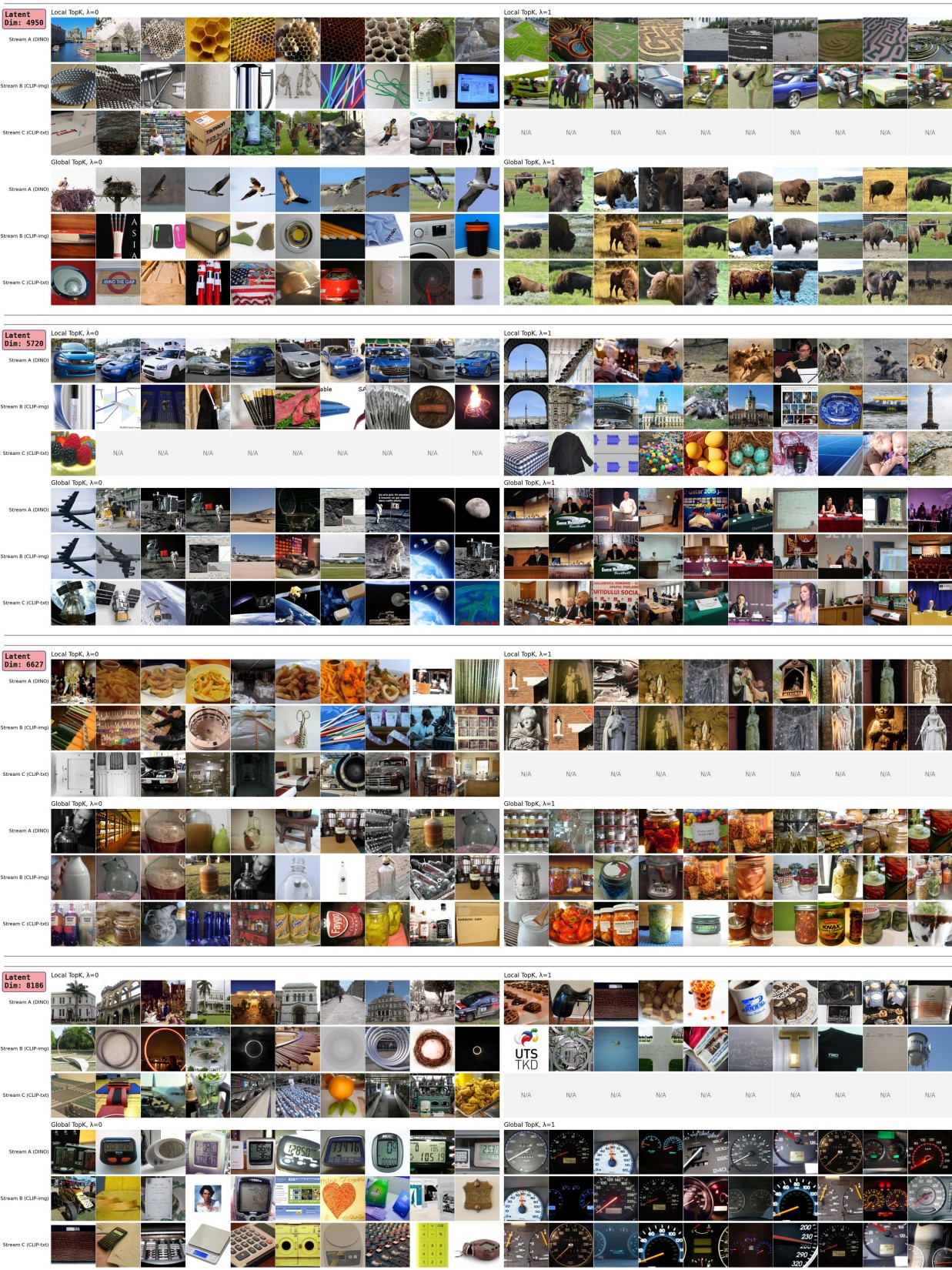

Figure 12: Additional latent activation examples for dimensions 4950, 5720, 6627, and 8186 showing top-10 activating images across different configurations.

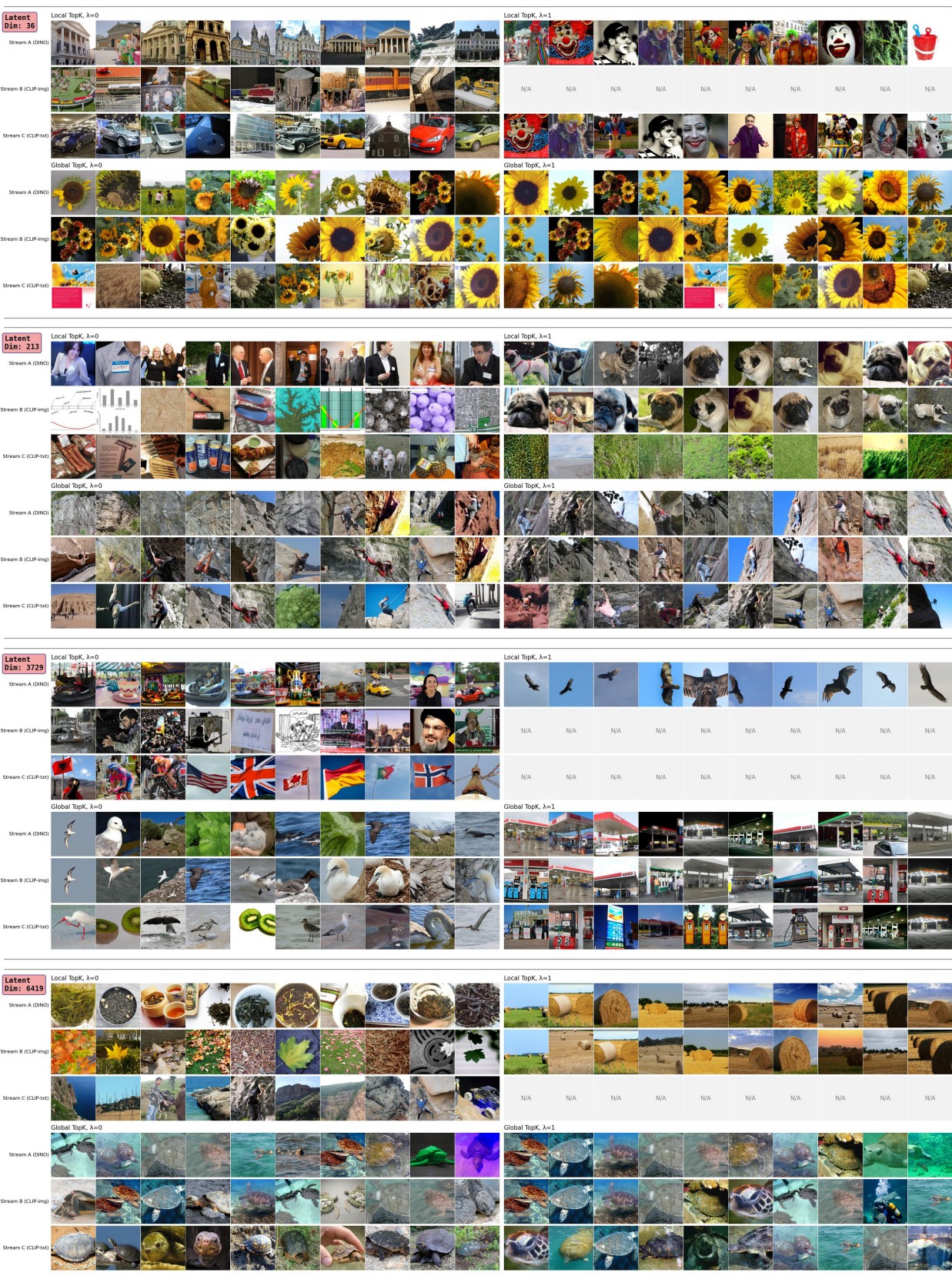

Figure 13: Additional latent activation examples for dimensions 36, 213, 3729, and 6419 showing top-10 activating images across different configurations.

# E   Latent Attribution based on concepts

This section demonstrates attribution analysis using SPARC's concept-aligned latents as scalar targets for gradient-based methods. We identify concept-relevant latents by analyzing their activation patterns on labeled data, then use their combined activations as attribution targets for spatial and textual attribution.

To identify which concepts each latent represents, we examine what types of images most strongly activate each latent dimension. For each latent, we analyze its top-50 activating samples and determine the most frequent concept category among them. We measure semantic consistency by computing the purity score, the fraction of top-activating samples that belong to the dominant category. When this purity exceeds 0.3, indicating sufficient semantic consistency, we assign the corresponding concept name to that latent. Latents without clear semantic patterns remain unlabeled. This process enables identification of multiple latents representing the same concept across different streams.

For attribution, we collect all latents assigned to a target concept across streams, forming the set $\mathcal{J}$ of concept-relevant indices. We then compute spatial attribution using both relevancy maps (Chefer et al., 2021) and GradCAM (Selvaraju et al., 2017) with the summed activations $\sum_{j \in \mathcal{J}} z_j^s$ as scalar targets. For text attribution, we compute token relevancy scores (Chefer et al., 2021). Each figure displays spatial attribution heatmaps alongside text token relevance scores, with scores below 0.1 omitted for clarity.

For more results, check `https://github.com/AtlasAnalyticsLab/SPARC/blob/main/VISUALIZATIONS.md`.

## E.1   Concept Specific Latents

Figures 14, 15, 16, and 17 show attribution results using concept-specific latent selections across various object categories.

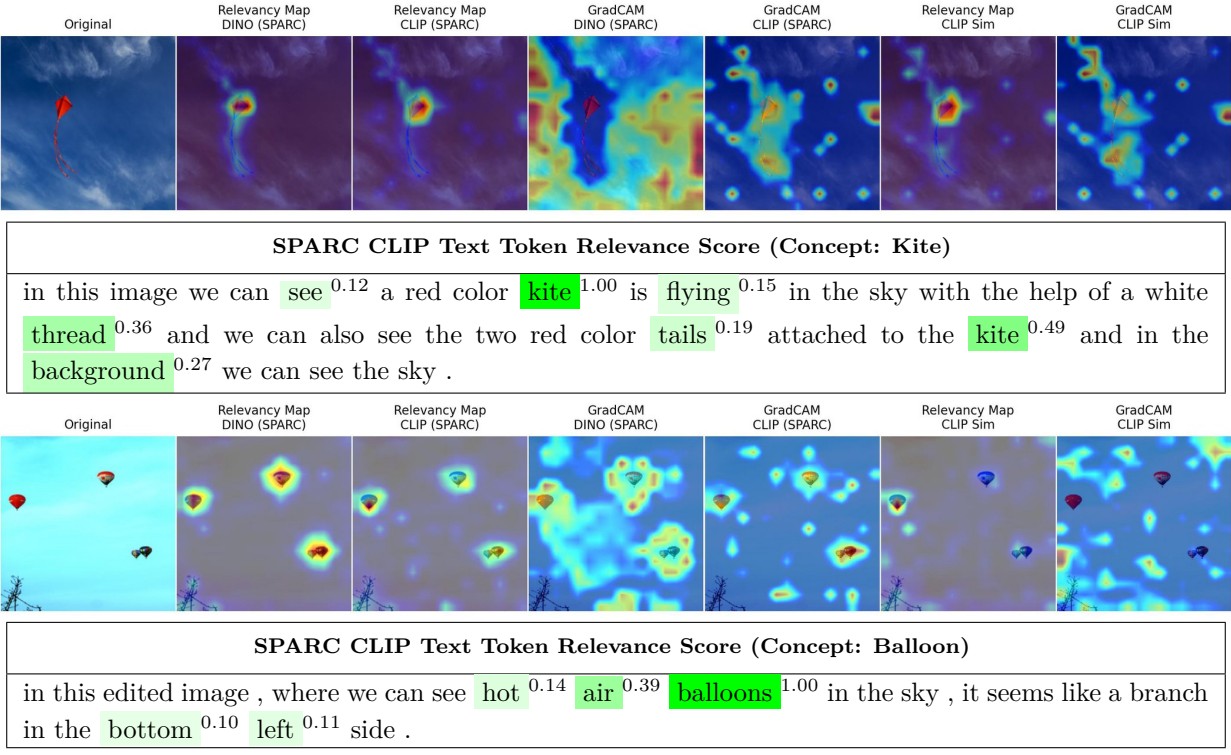

Figure 14: SPARC CLIP text token relevance for kite and balloon concepts. CLIP similarity baseline uses concept names "a kite" and "a balloon" rather than full captions.

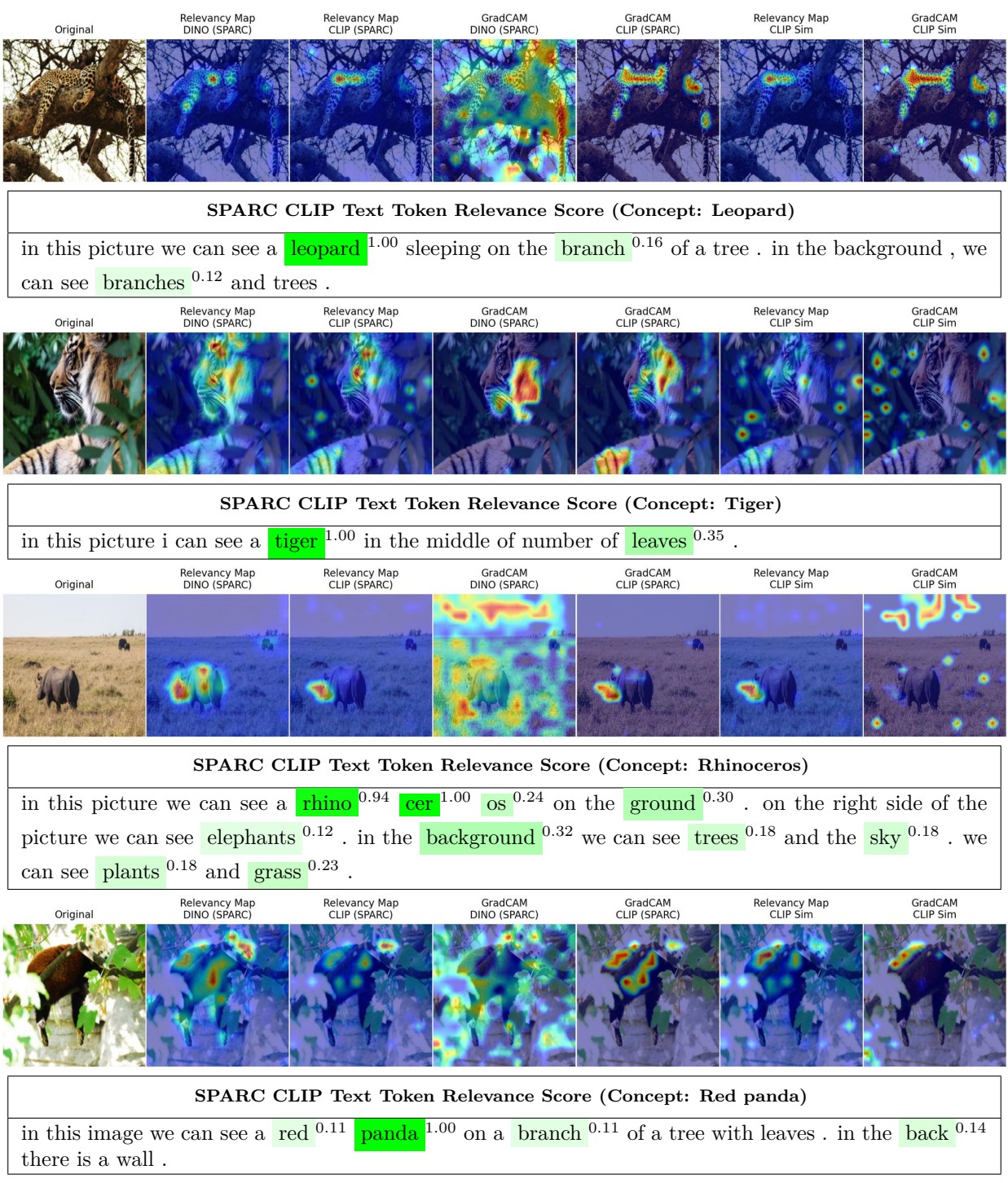

Figure 15: SPARC CLIP text token relevance for leopard, tiger, rhinoceros, and red panda concepts. CLIP similarity baseline uses concept names "a leopard", "a tiger", "a rhinoceros", and "a red panda" rather than full captions.

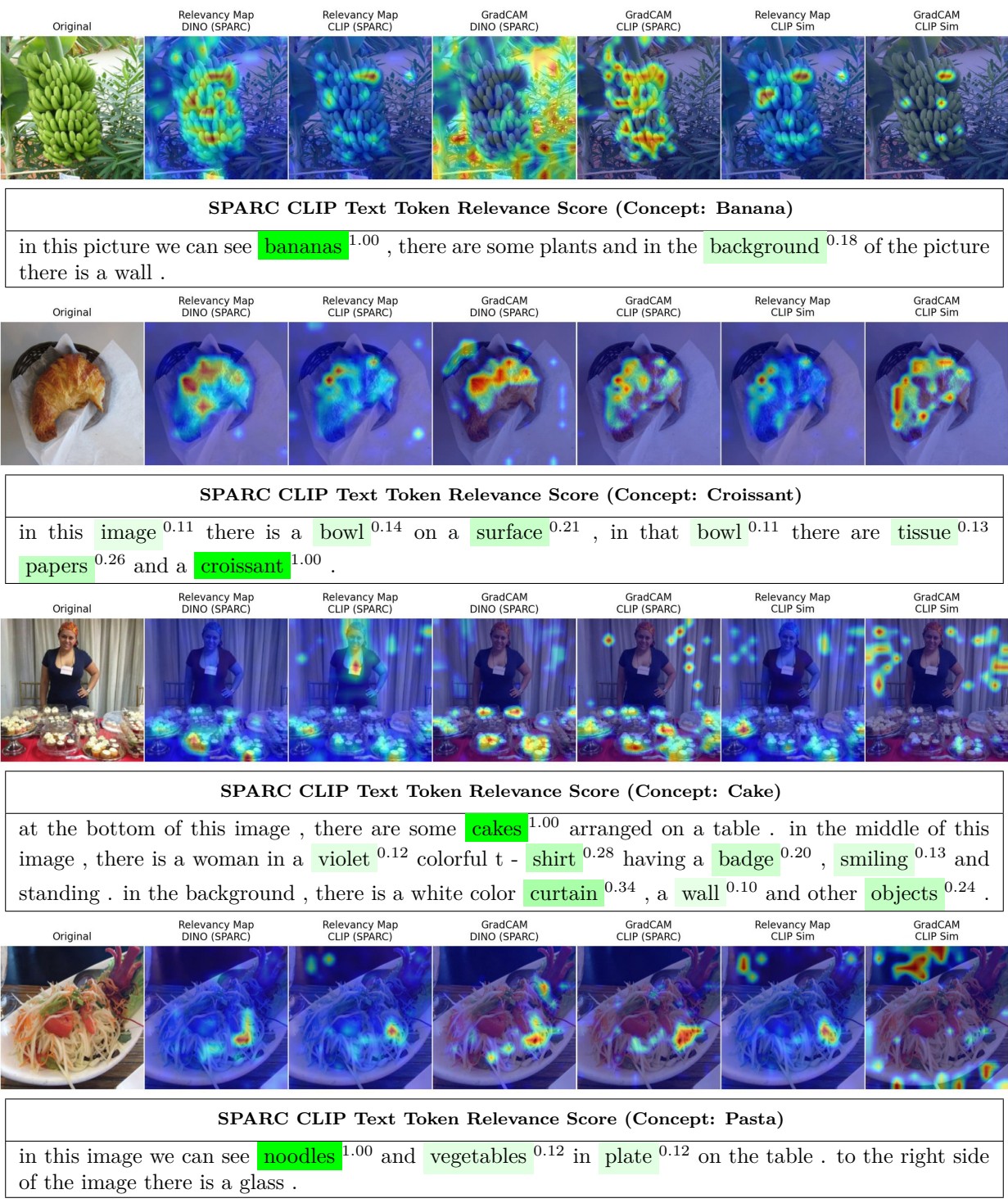

Figure 16: SPARC CLIP text token relevance for banana, croissant, cake, and pasta concepts. CLIP similarity baseline uses concept names "a banana", "a croissant", "a cake", and "pasta" rather than full captions.

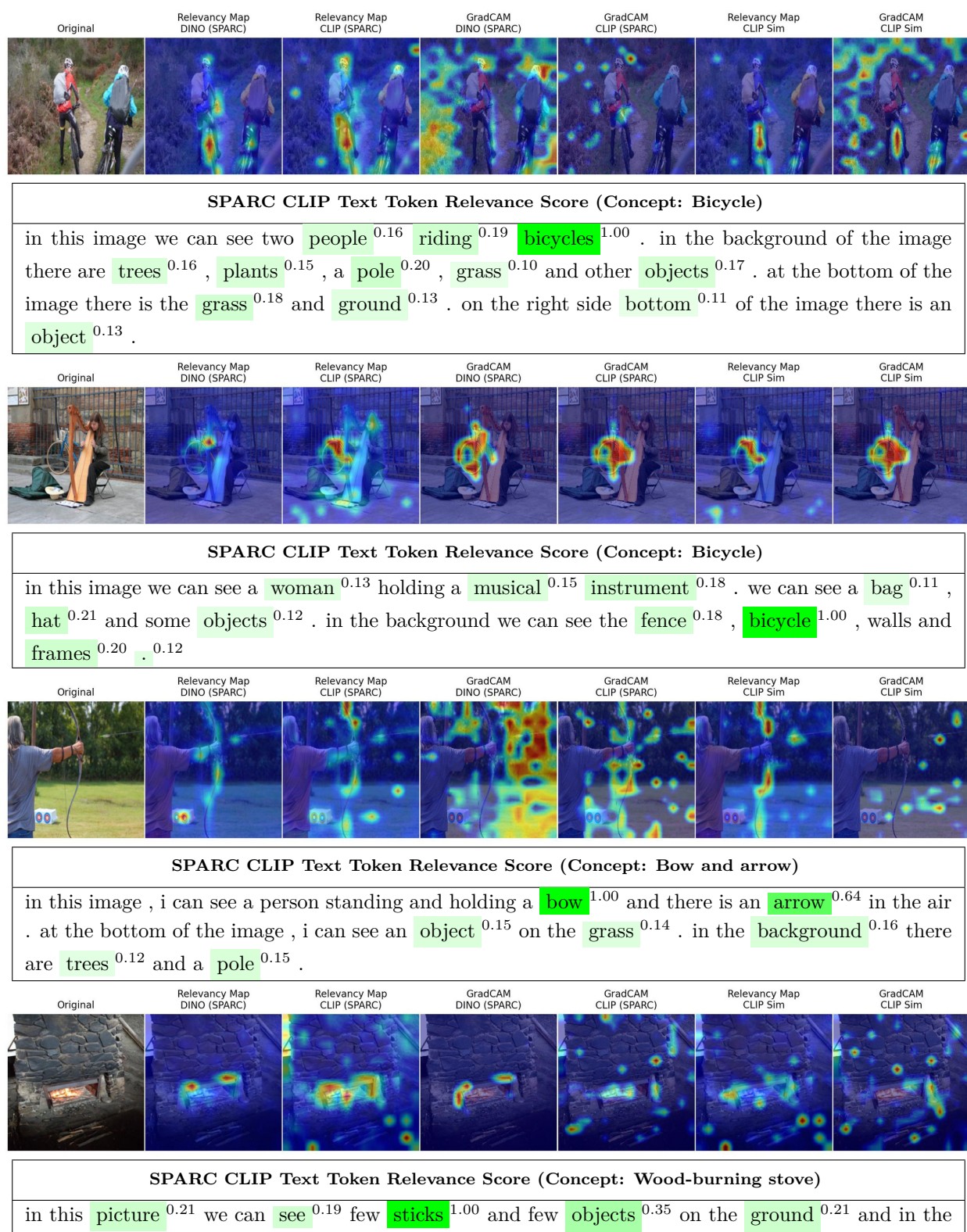

Figure 17: SPARC CLIP text token relevance for bicycle, bow and arrow, and wood-burning stove concepts. CLIP similarity baseline uses concept names "a bicycle", "bow and arrow", and "wood-burning stove" rather than full captions.

## E.2 Same image/caption, different latents

Figure 18 demonstrates how different concept selections produce attribution patterns for identical inputs. By changing the set $\mathcal{J}$ while keeping same image and caption, SPARC generates different attribution patterns.

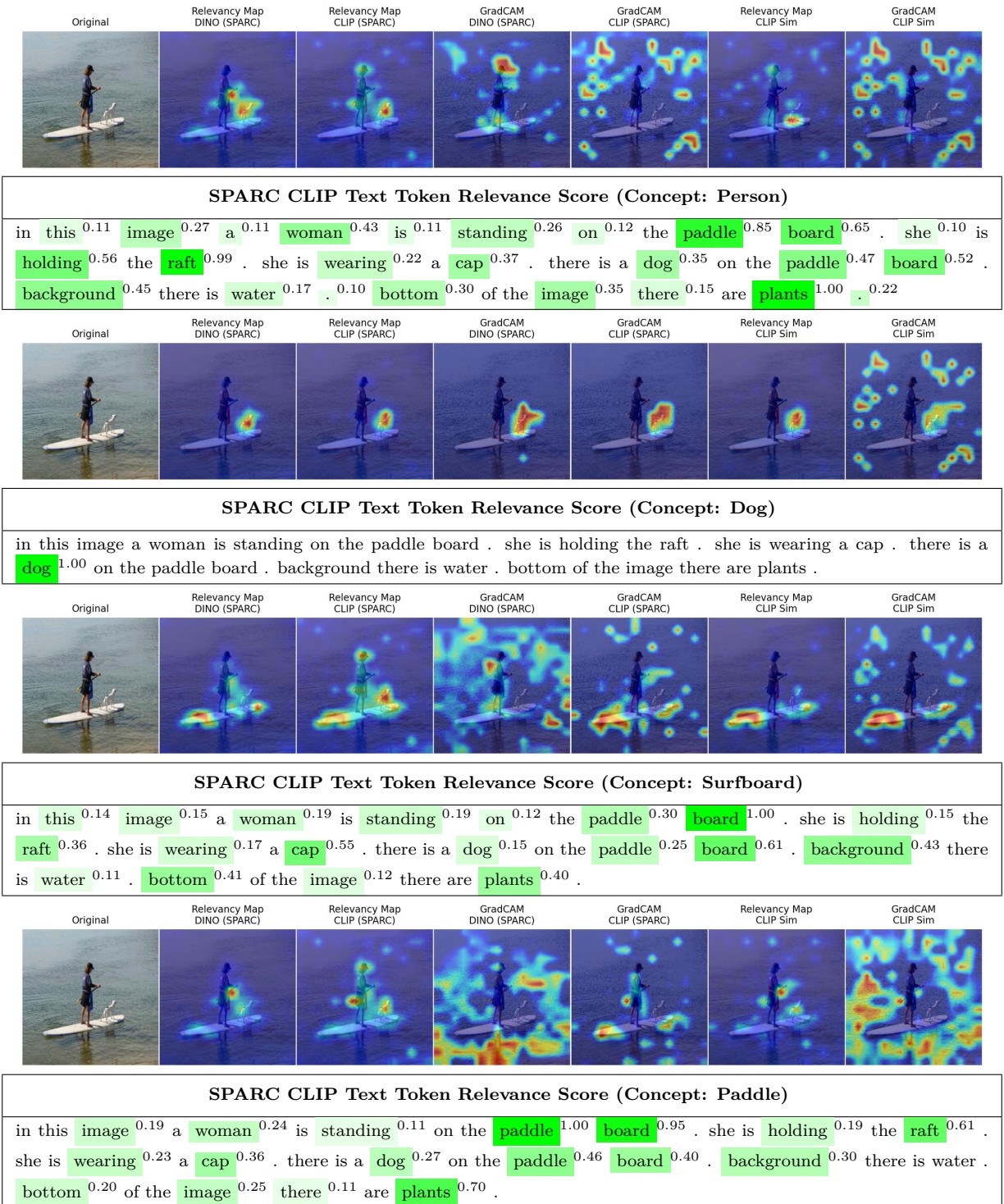

Figure 18: SPARC CLIP text token relevance for the same image/caption using different concept-specific latent sets. Each panel shows attribution for a different target concept as indicated in the table headers.

### E.3   Using latents of concepts that are not present in the image/caption

Figure 19 examines SPARC's behavior when applying concept latents to samples lacking those concepts. For these samples, when using irrelevant concept latent sets $\mathcal{J}$, SPARC latents will be zero, meaning no gradients are produced and all attribution scores remain zero.

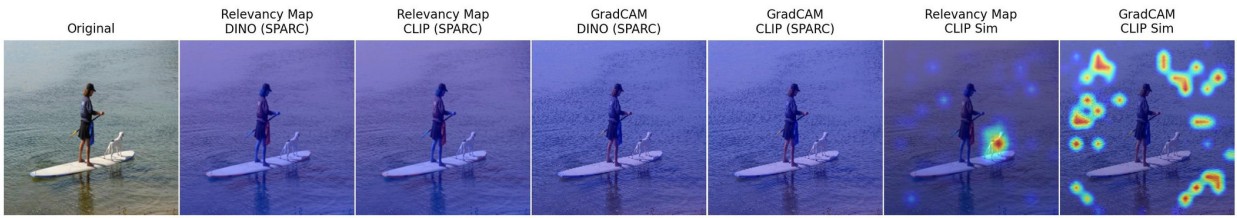

**SPARC CLIP Text Token Relevance Score (Concept: Cat)**

in this image a woman is standing on the paddle board . she is holding the raft . she is wearing a cap . there is a dog on the paddle board . background there is water . bottom of the image there are plants .

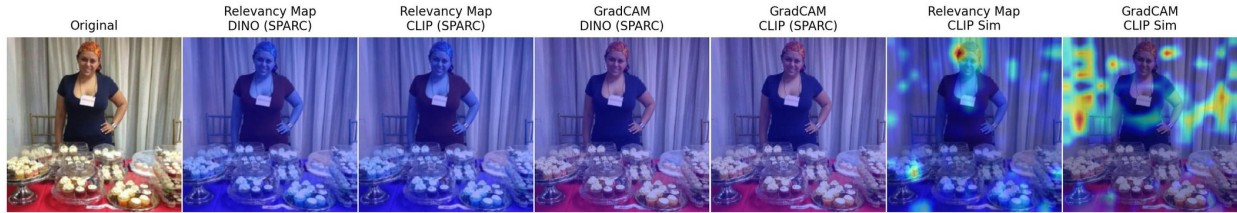

**SPARC CLIP Text Token Relevance Score (Concept: Apple)**

at the bottom of this image , there are food items arranged on a table . in the middle of this image , there is a woman in a violet colorful t - shirt having a badge , smiling and standing . in the background , there is a white color curtain , a wall and other objects .

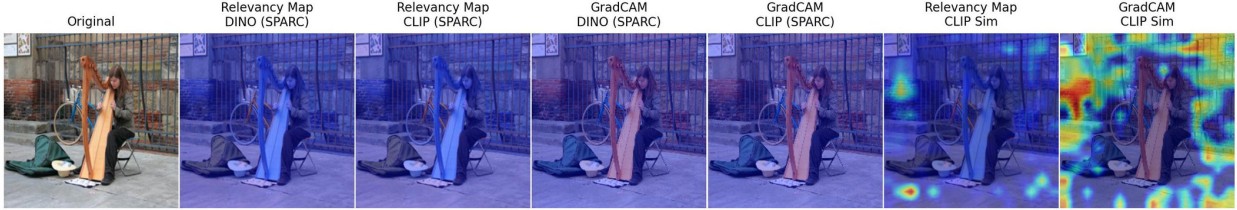

**SPARC CLIP Text Token Relevance Score (Concept: Flag)**

in this image we can see a woman holding a musical instrument . we can see a bag , hat and some objects . in the background we can see the fence , bicycle , walls and frames .

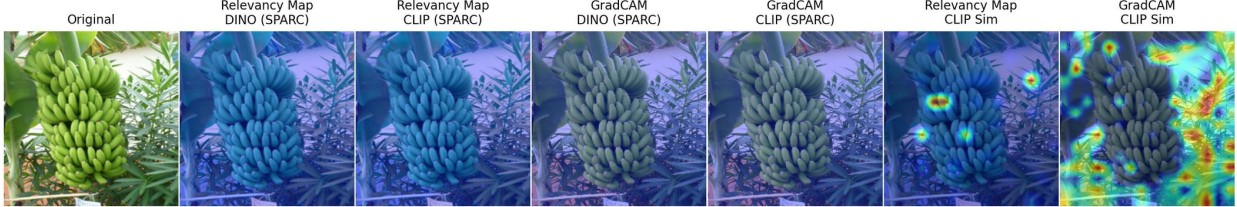

**SPARC CLIP Text Token Relevance Score (Concept: Tiger)**

in this picture we can see bananas , there are some plants and in the background of the picture there is a wall .

Figure 19: SPARC text token relevance for image/captions with a concept that's not present in the sample. Using irrelevant latent dimensions in SPARC causes no gradients. For text scores, all scores are 0.

### E.4 Limitations of concept-based latent attribution

For certain common classes such as person and car, many latents receive concept assignments, over 500 latents for "person" and nearly 300 for "car". This is mainly a limitation of assigning concepts to latents as discussed in Section 4.2.

This breaks the selective attribution behavior of SPARC demonstrated in Appendix E.3. While concept-specific latents for most classes produce no gradients when the concept is absent, common concepts with numerous assigned latents produce spurious activations even when not present. Figure 20 demonstrates this limitation, where "person" and "car" latents generate non-zero attributions on images containing neither concept.

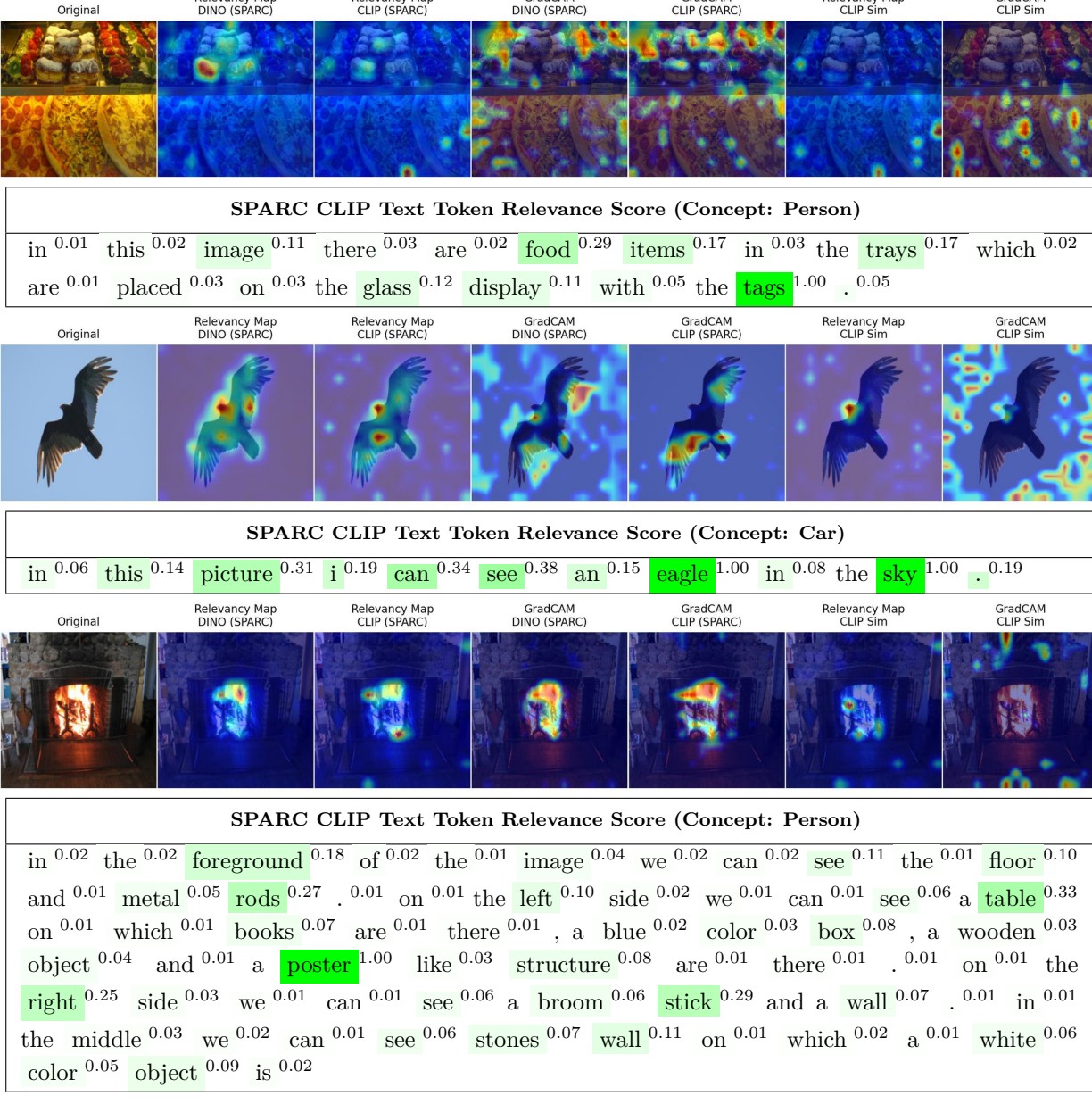

Figure 20: SPARC CLIP text token relevance for image/captions with a concept that's not present in the sample. SPARC produces non-zero gradients for some of common concepts even in the absence of the concept.

# F  Cross-Modal Similarity Attribution

This section demonstrates attribution using cross-modal similarities $\mathbf{z}^s \cdot \mathbf{z}^t$ in SPARC's aligned latent space as scalar targets. We compute spatial attribution using both relevancy maps (Chefer et al., 2021) and GradCAM (Selvaraju et al., 2017), while text attribution uses token relevancy scores (Chefer et al., 2021).

## F.1  Cross-modal heatmaps with full captions

Figures 21, 22, and 23 demonstrate cross-modal attribution using full captions as text input. Each figure shows spatial heatmaps alongside text token relevance scores, comparing SPARC's aligned latent similarities against CLIP similarity baselines across various object categories. For more results, check `https://github.com/AtlasAnalyticsLab/SPARC/blob/main/VISUALIZATIONS.md`.

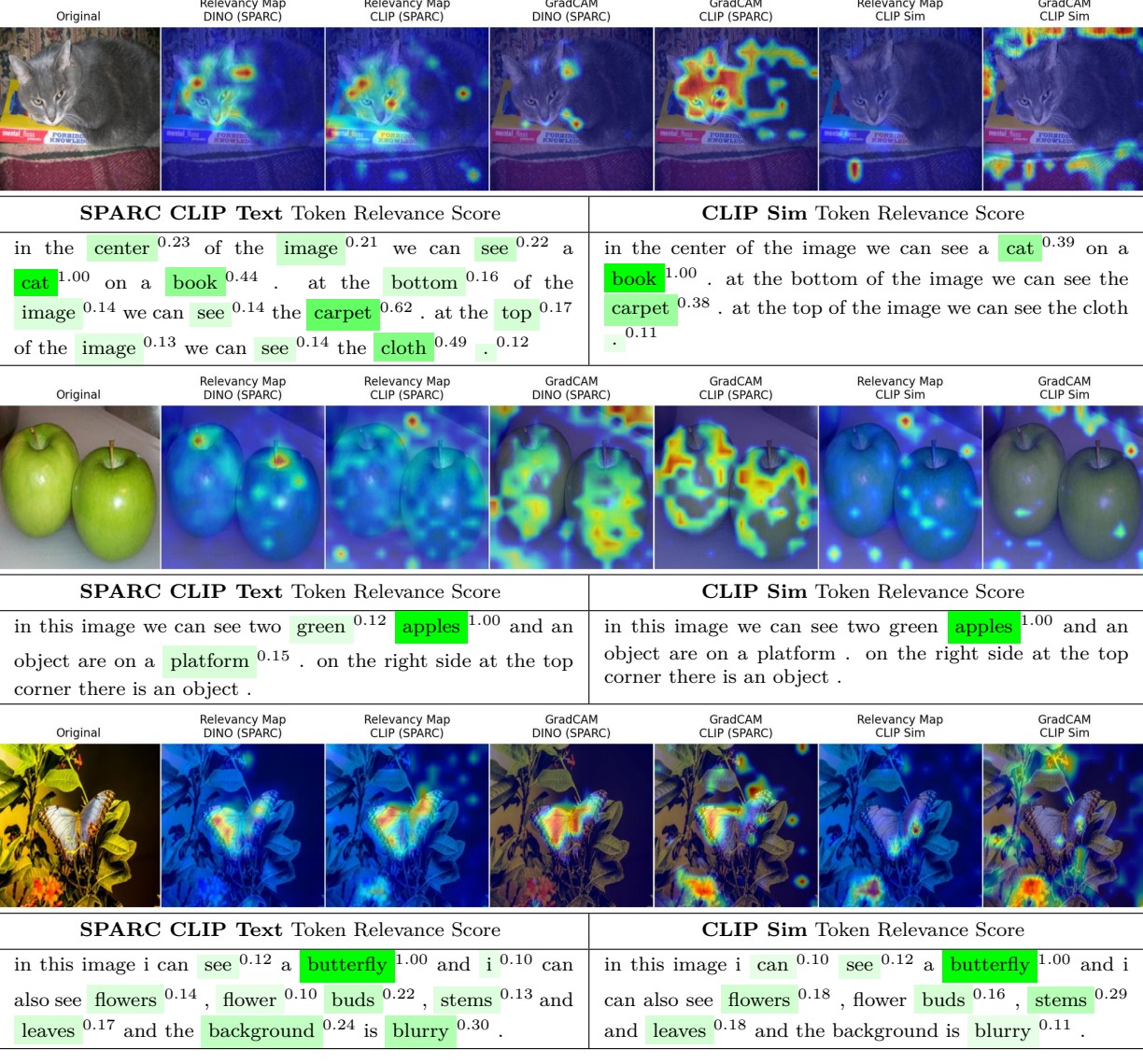

Figure 21: Cross-modal similarity attribution for mixed concepts (cat, apple, butterfly) comparing SPARC's aligned latent space against CLIP similarity baseline.

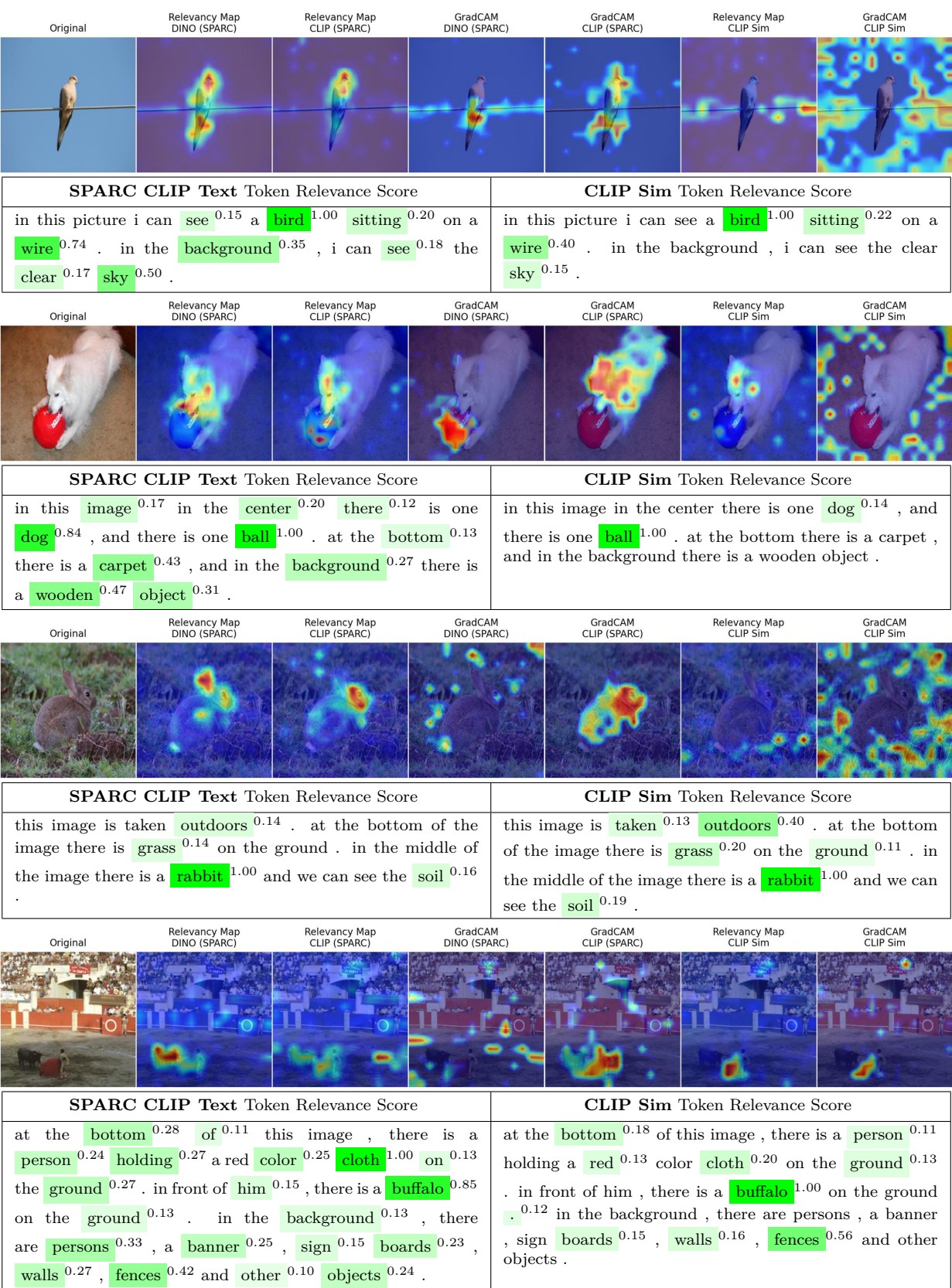

Figure 22: Cross-modal similarity attribution for animal concepts comparing SPARC's aligned latent space against CLIP similarity baseline.

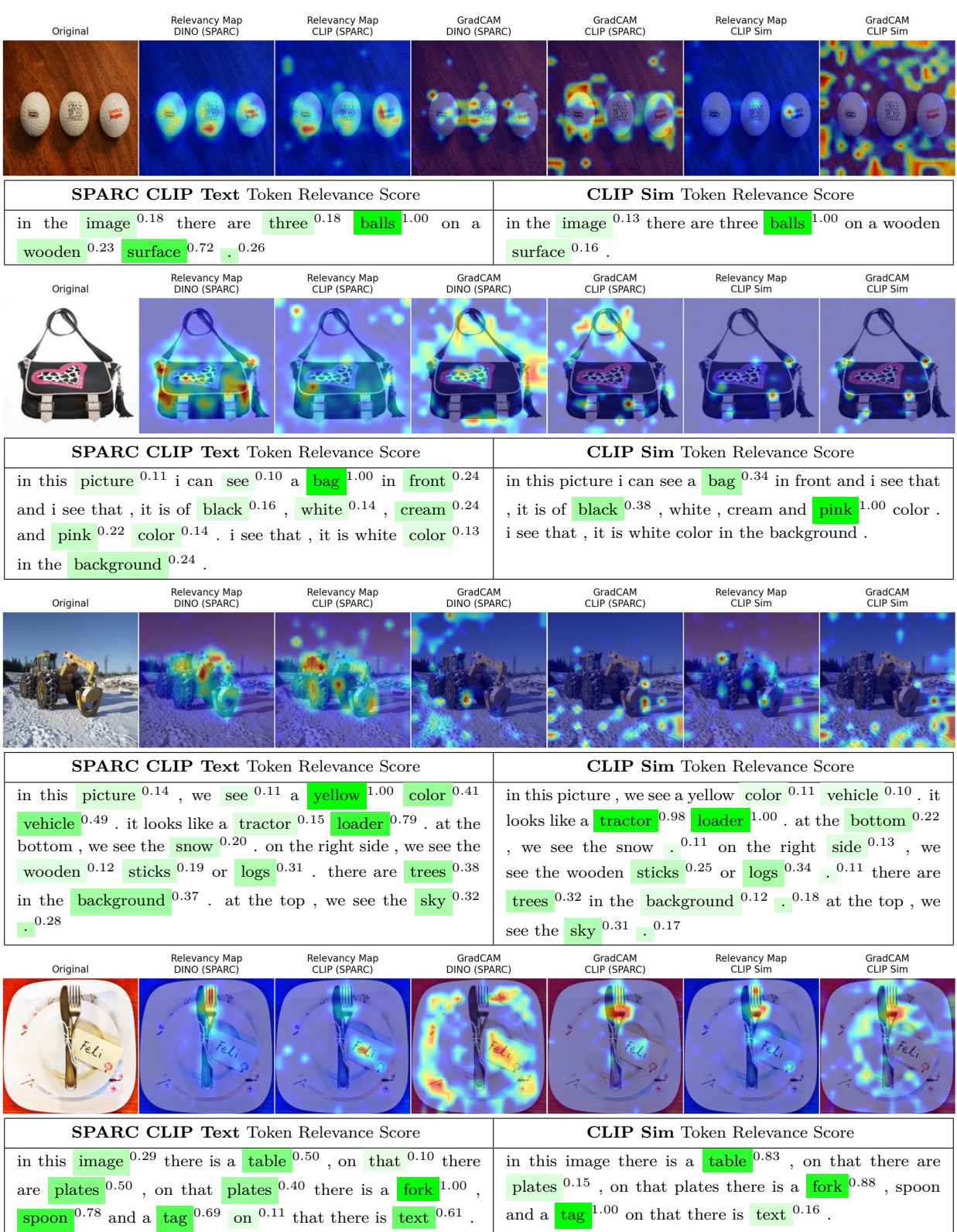

Figure 23: Cross-modal similarity attribution for object concepts comparing SPARC's aligned latent space against CLIP similarity baseline.

### F.2  Same image, different captions

Figure 24 demonstrates how different text queries produce distinct spatial attribution patterns when applied to the same image. Using simple concept names ("Banana", "Apple", "Kiwi", "Cat") as text inputs, we examine how cross-modal similarities $\mathbf{z}^s \cdot \mathbf{z}^t$ generate different heatmaps based on text guidance, including cases where the queried concept is absent from the image.

For this sample image, CLIP similarity shows focused attribution on target objects when present in the image. SPARC exhibits more varied attribution patterns, localizing to different image regions with less precision than CLIP similarity. We make no claims about performance or whether these heatmaps are meaningful or relevant to what the actual encoders are looking at. We present these examples to demonstrate that SPARC's spatial attribution varies with different text inputs, indicating text-guided behavior rather than text-independent object highlighting.

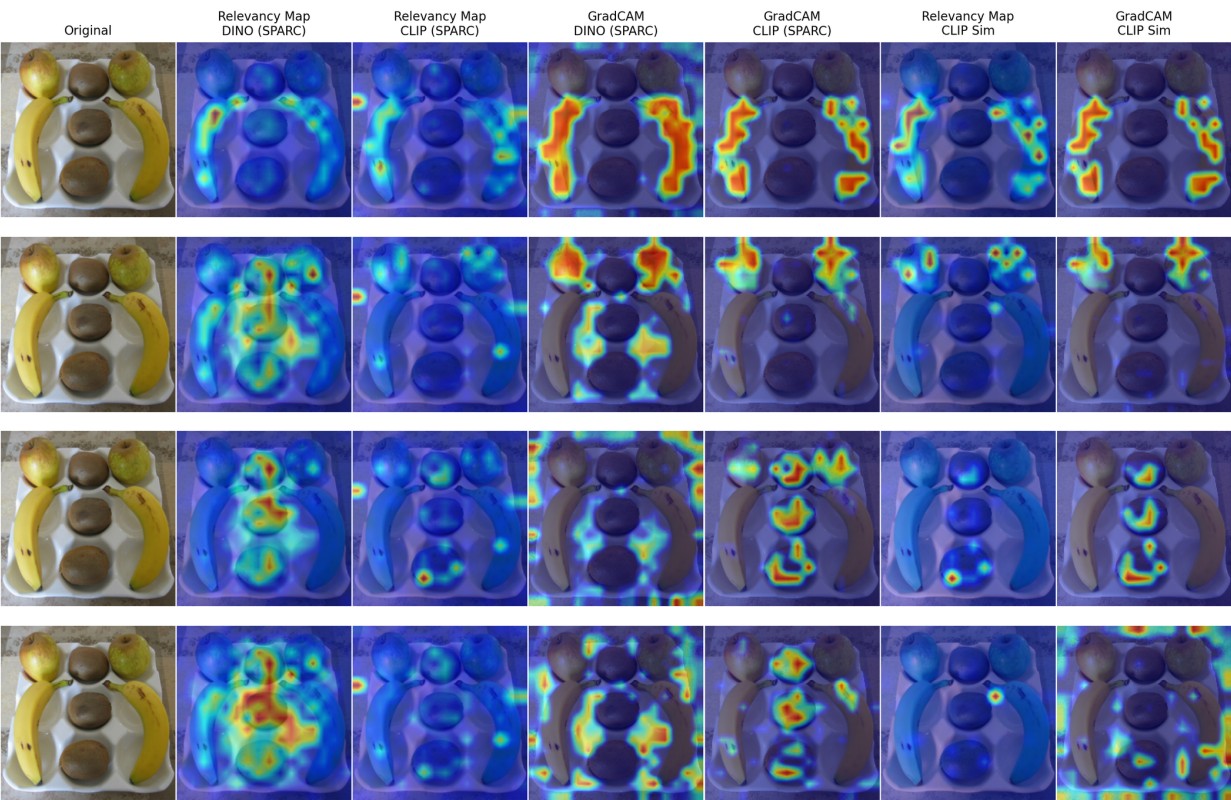

Figure 24: Captions used are "Banana", "Apple", "Kiwi", and "Cat" (non-existent concept).

### F.3 Cross-modal attribution limitations

Figure 25 shows a case where SPARC's spatial attribution produces less meaningful localization compared to Figure 24. Using detailed spatial queries ("Cat's Ears", "Cat's Eyes", "Cat's Nose", "A Cat"), SPARC generates similar, poorly localized attribution patterns with minimal variation across different text inputs. Although CLIP doesn't match the captions perfectly, it shows more responsiveness to the text input.

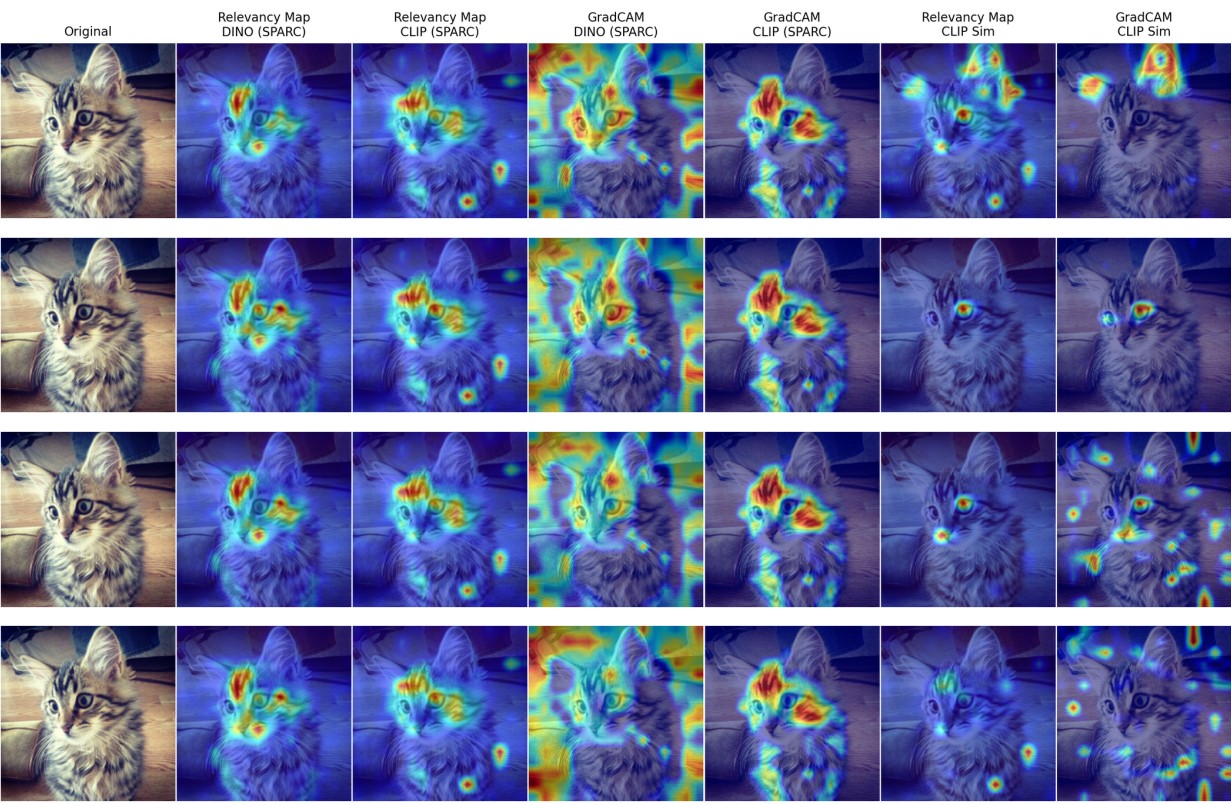

Figure 25: Captions used are "Cat's Ears", "Cat's Eyes", "Cat's Nose", and "A Cat". We find SPARC fails in the case of detailed heatmaps.

Similarly, Figure 26 further illustrates this limitation using distinct objects within a scene. When querying for specific components of the bedroom scene ("Wall", "Lamps", "Pillows"), SPARC's attribution maps exhibit minimal variation, consistently focusing on the dominant central object (the bed and pillows) regardless of the text input. In contrast, the CLIP similarity baseline demonstrates effective responsiveness, shifting its focus to the background for "Wall", the peripheral regions for "Lamps", and the center for "Pillows".

A similar failure mode is observed in Figure 27 when distinguishing fine-grained facial features. Using specific queries ("Mouth", "A tie", "Eyes"), SPARC generates a static attribution pattern focused primarily on the lower face, failing to localize the eyes or the tie. Conversely, CLIP similarity accurately localizes the tie and eyes when prompted, showing superior sensitivity to fine-grained part-level text queries.

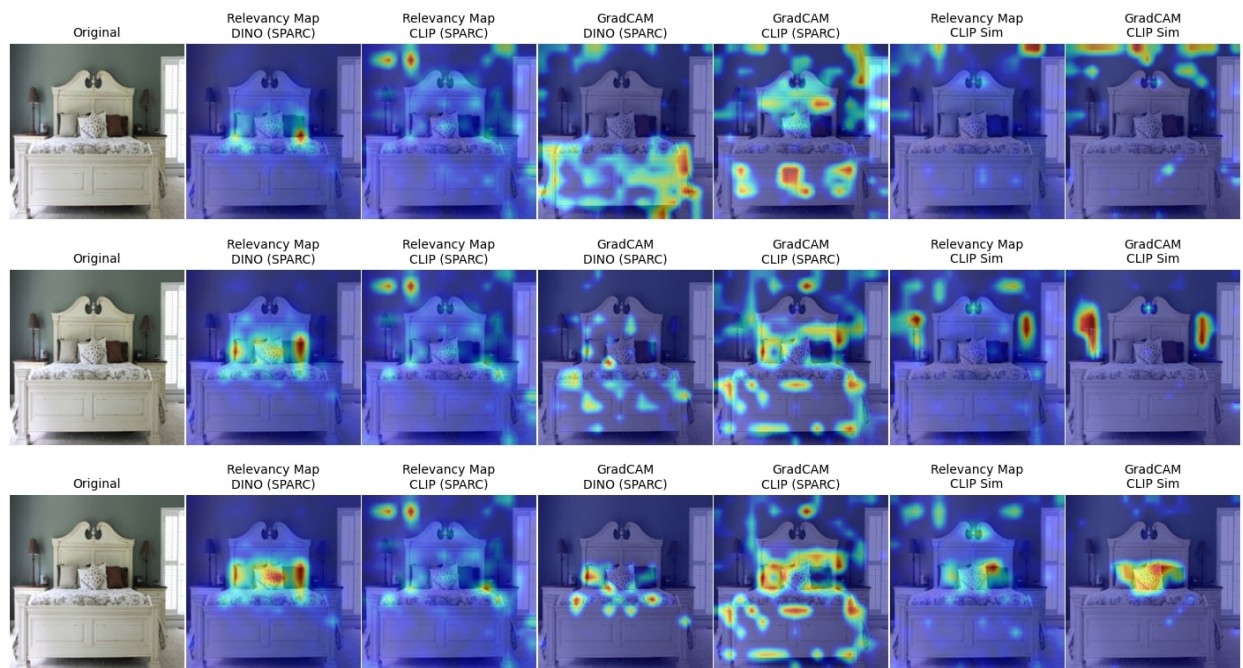

Figure 26: Captions used are "Wall", "Lamps", and "Pillows". SPARC produces nearly identical heatmaps for all queries, defaulting to the central object (bed), whereas CLIP correctly shifts focus to the queried objects.

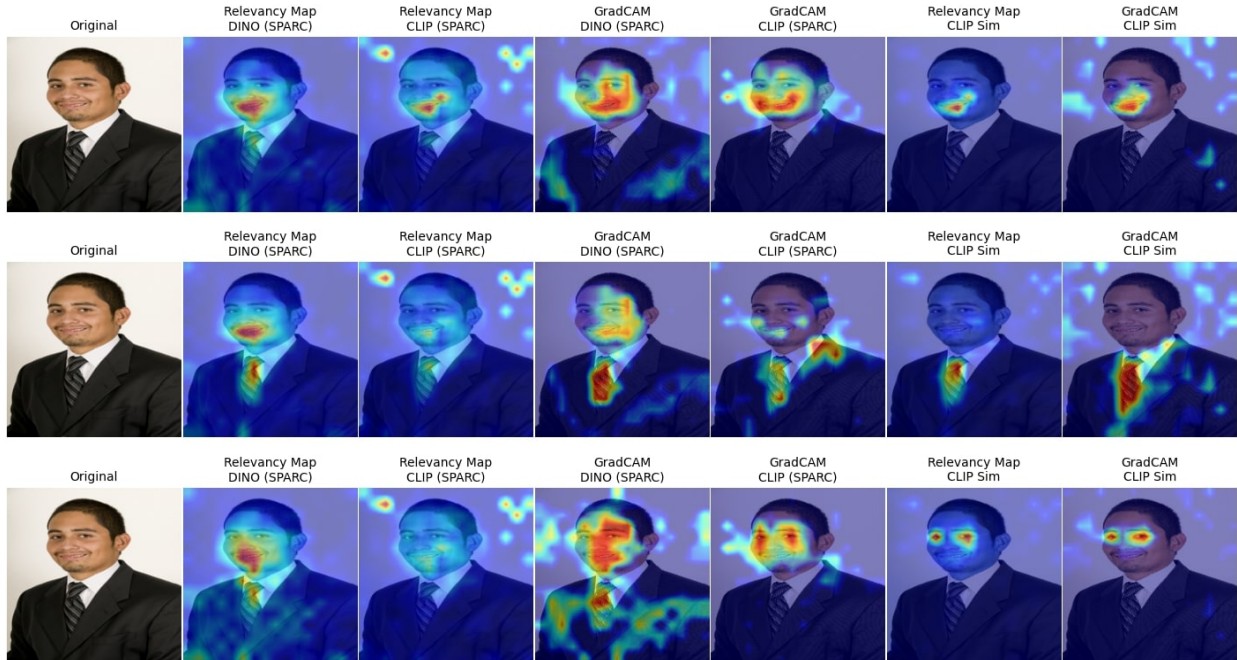

Figure 27: Captions used are "Mouth", "A tie", and "Eyes". SPARC fails to disentangle the specific features, while CLIP similarity successfully localizes the tie and eyes.

Additional failure cases are shown in Figures 28, 29, and 30, where we use single-word queries that refer primarily to background scene elements ("Trees", "Grass", "Rocks"). Across all three examples, SPARC's cross-modal attribution remains biased toward salient foreground objects and produces diffuse or weakly localized heatmaps that only partially overlap with the queried regions. In contrast, the CLIP similarity baseline shifts attribution more clearly toward the corresponding background structures.

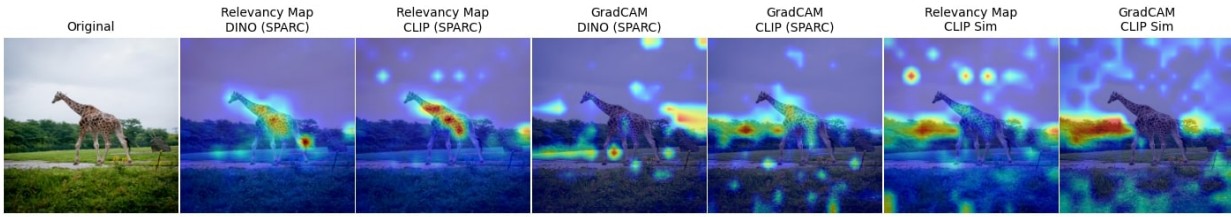

Figure 28: Caption used is "Trees". SPARC produces diffuse, weakly localized attribution, while CLIP similarity better aligns with the tree regions in the scene.

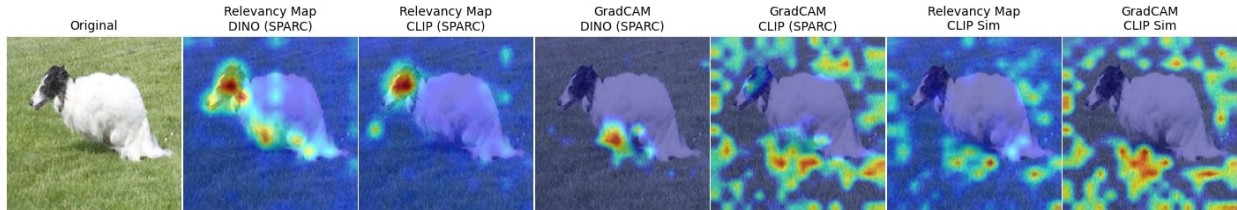

Figure 29: Caption used is "Grass". SPARC's attribution primarily attends to the foreground object (dog) rather than the ground, whereas CLIP similarity allocates more emphasis to the grassy regions, better matching the queried concept.

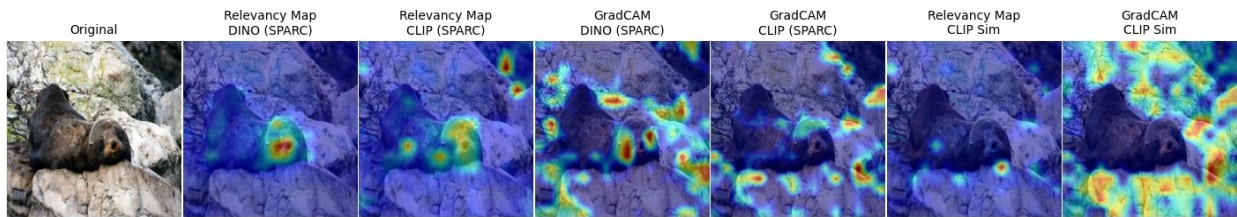

Figure 30: Caption used is "Rocks". SPARC's attribution remains focused on the foreground animals, while CLIP similarity better aligns with the rocky background queried by the text.

## G  Downstream Retrieval

To assess the holistic alignment of the latent space, beyond individual neuron analysis, we measure performance on a cross-stream vector retrieval task ($\mathbf{z}^s \to \mathbf{z}^t$). Table 15 summarizes the Recall@1 (R@1) scores across datasets and training configurations. A clear pattern emerges: the combination of Global TopK and a cross-reconstruction loss ($\lambda > 0$) yields a substantial improvement in alignment, especially for vision-to-vision retrieval.

Table 15: Latent alignment R@1 scores across datasets and training regimes. CI = CLIP_IMG, CT = CLIP_TXT, D = DINO.

| TopK | $\lambda$ | CI→CT | CI→D | CT→CI | CT→D | D→CI | D→CT |
|------|-----------|-------|------|-------|------|------|------|
| | | *Open Images* | | | | | |
| Global | 1 | **0.034** | 0.352 | **0.031** | **0.018** | 0.284 | **0.021** |
| | 0 | 0.010 | 0.078 | 0.007 | 0.002 | 0.173 | 0.010 |
| Local | 1 | 0.011 | **0.366** | 0.010 | 0.008 | **0.302** | 0.009 |
| | 0 | 0.000 | 0.000 | 0.000 | 0.000 | 0.000 | 0.000 |
| | | *MS-COCO* | | | | | |
| Global | 1 | **0.421** | **0.762** | **0.386** | **0.325** | **0.716** | **0.347** |
| | 0 | 0.181 | 0.259 | 0.204 | 0.102 | 0.369 | 0.188 |
| Local | 1 | 0.365 | 0.694 | 0.329 | 0.261 | 0.714 | 0.301 |
| | 0 | 0.000 | 0.000 | 0.000 | 0.000 | 0.000 | 0.000 |

Cross-modal retrieval on Open Images shows weak performance compared to MS-COCO. SPARC's best cross-modal results (Global TopK, $\lambda = 1.0$) achieve 3.4% R@1 for CI→CT and 3.1% R@1 for CT→CI. To contextualize these numbers, raw CLIP features achieve 9.8% and 7.7% R@1, respectively on the same tasks, indicating that Open Images presents retrieval challenges even for CLIP's well-aligned cross-modal representations.

Qualitative examples in Appendices H.1–H.2 show semantically appropriate retrievals despite low R@1 scores. Cross-model image retrieval for image-to-image tasks is provided in Appendix H.3. Out-of-distribution tests (Appendices H.4–H.5) use external images and free-form captions without corresponding pairs in the dataset.

## H  Retrieval Qualitative Results

This section presents qualitative retrieval examples using SPARC's aligned latent representations. The following subsections show retrieval samples across in-distribution tasks using test data queries, and out-of-distribution tasks using external queries not present in the dataset. In both cases, the reference database consists of test set samples.

### H.1  Image → Caption Retrieval (In-Distribution)

We evaluate cross-modal alignment through image-to-caption retrieval using SPARC latent representations. Tables 16–25 show retrieval results comparing Global vs Local TopK training configurations.

The four model configurations represent:

- **Global DINO**: Query image's DINO features → Reference database of CLIP-text features (both from Global SPARC)

- **Local DINO**: Query image's DINO features → Reference database of CLIP-text features (both from Local SPARC)

- **Global CLIP**: Query image's CLIP-image features → Reference database of CLIP-text features (both from Global SPARC)

- **Local CLIP**: Query image's CLIP-image features → Reference database of CLIP-text features (both from Local SPARC)

Each model shows top-5 retrieved captions ranked by cosine similarity in the SPARC latent space.

**Open Images dataset's Image -> Caption Retrieval**

Tables 16, 17, 18, 19, and 20 show examples on the Open Images test set across diverse scene types.

| Model | Rank | Caption |
|---|---|---|
| Global DINO | 1 | In this image I can see bed with bed sheet and pillows and I can also see table, lamps, something looking like glass and in the background I can see curtains and wall. |
| | 2 | This is a picture of a room, in this image in the center there is a bed, on the bed there are pillows and there are lamps, photo frame, tables. On the tables there telephones and some papers, and on the ri... |
| | 3 | This picture is clicked inside the room. In this picture, we see the beds and the pillows. We see the blankets in grey color. In between the beds, we see a table on which a telephone and a remote are place... |
| | 4 | In this image there are two wooden beds with mattresses and pillows on them. In between the beds on the table there is a telephone and some other objects. Behind the bed there are switches, lamps and curta... |
| | 5 | In this image there are pillows, book, paper on the bed. Beside the bed there is another bed. On top of the bed there is a pillow. There are lamps, landline phone and some other objects on the table. There... |
| Local DINO | 1 | In this image I can see a bed. I can see few pillows, towels and blankets on the bed. I can see a lamp on the stool. On the left side I can see few curtains. |
| | 2 | In the center of the image we can see beds. On the beds we can see clothes, curtains and some objects. At the bottom we can see shoes on the floor. In the background there is wall. |
| | 3 | In this image there is a bed in the room, behind the bed there is a wall, beside the bed there are doors. |
| | 4 | This image is taken in the room. In this image there are beds and we can see clothes placed on the beds. There is a television placed on the stand. We can see a table and there is a laptop, lamp and some o... |
| | 5 | This is an inside view of a room. In this picture we can see a bed. On this bed, we can see blankets and pillows. We can see walls, a window with window doors and a window shelf. We can see an object on th... |
| Global CLIP | 1 | This is a picture of a room, in this image in the center there is a bed, on the bed there are pillows and there are lamps, photo frame, tables. On the tables there telephones and some papers, and on the ri... |
| | 2 | This picture is clicked inside the room. In this picture, we see two beds and the pillows. In between the beds, we see a table on which a remote is placed. On the left side, we see a white color object. In... |
| | 3 | In this picture I can see two beds with blankets and pillows on the beds. I can see a lamp on the table and I can see few items on the table. Looks like another lamp on the table on the right side, curtain... |
| | 4 | In this picture we can see two beds with pillows and bed sheets on it and these beds and a table on the floor and on this table we can see two lights table lamp, papers, pen and in the background we can se... |
| | 5 | In this picture we can see two beds, there are pillows and bed sheets placed on the beds, in the background we can see a wall, a curtain, a photo frame and a light, there is some text at the right bottom. |
| Local CLIP | 1 | In this image I can see two beds with pillows and blankets and I can also see wooden table with drawers, lamp, telephone, some other items and in the background I can see picture frame, floor, door and wal... |
| | 2 | In this picture I can see two beds with blankets and pillows on the beds. I can see a lamp on the table and I can see few items on the table. Looks like another lamp on the table on the right side, curtain... |
| | 3 | In this picture we can see two beds, there are pillows and bed sheets placed on the beds, in the background we can see a wall, a curtain, a photo frame and a light, there is some text at the right bottom. |
| | 4 | This image is taken indoors. In the middle of the image there are two beds with mattress, bed sheets, blankets and pillows on them. On the right side of the image there is a wall. At the left bottom of the... |
| | 5 | This image is taken from inside. In this image there are two beds with pillows on it, in the middle of them, there is a table. On the table there is a book, paper, telephone, lamp and other object. In the ... |
| Original | – | This is an inside view of a room, we can see two beds and there are some pillows on the beds. On the left side, there is a table and we can see the chairs. There is an air conditioner on the wall, we can see the window and curtains. |

Table 16: The query image 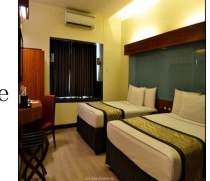 is used to retrieve the captions. None of the retrieved text is the exact caption of the query image, but still highly relevant captions.

| Model | Rank | Caption |
|---|---|---|
| Global DINO | 1 | In this image there are people sitting on wheelchairs and playing basketball, in the background it is blurred. |
| | 2 | In this image there are a few people in wheelchairs are playing basketball. Behind them there are a few people sitting in chairs, behind them there is a person performing gymnasium on the ropes. In the bac... |
| | 3 | In this image, I can see a group of people sitting on the wheelchairs, which are on the floor. Among them one person is holding a basketball. In the background there is a wall and a railing. In the top lef... |
| | 4 | In this image I can see few people playing wheelchair basketball. |
| | 5 | In this image, I see a man sitting in a wheel chair while holding a basket ball in his hands and behind him I see group of people sitting in wheel chairs. In the background I see a group of people standing... |
| Local DINO | 1 | In the picture I can see the men wearing the sports jersey and they are playing the basketball. I can see the spectators sitting on the chairs. |
| | 2 | In this image I can see few people playing basketball. I can see a person sitting on the metal stand, few persons sitting in the stadium and few stairs. |
| | 3 | In this image we can see men are playing basketball. In the background, we can see people are sitting and watching the game. |
| | 4 | In this image we can see people playing a game. The person in the center is holding a ball. In the background there are people standing. |
| | 5 | In this picture I can see a few people wearing jerseys. I can see a few people sitting. I can see the ball. I can see basketball rings. |
| Global CLIP | 1 | In this image, I can see a group of people sitting on the wheelchairs, which are on the floor. Among them one person is holding a basketball. In the background there is a wall and a railing. In the top lef... |
| | 2 | In this image there are people sitting on wheelchairs and playing basketball, in the background it is blurred. |
| | 3 | In this image I can see few people playing wheelchair basketball. |
| | 4 | In this image there are a few people in wheelchairs are playing basketball. Behind them there are a few people sitting in chairs, behind them there is a person performing gymnasium on the ropes. In the bac... |
| | 5 | In this image, I see a man sitting in a wheel chair while holding a basket ball in his hands and behind him I see group of people sitting in wheel chairs. In the background I see a group of people standing... |
| Local CLIP | 1 | In this picture I can see two people wearing medals and sitting on the wheelchairs. I can see a few people standing. I can see a person sitting on the wheelchair on the left side. I can see the roof at the... |
| | 2 | In the image there are few people sitting in the wheelchairs. And those wheelchairs are on the road. In the background there are doors which are looking blur. |
| | 3 | In this picture we can see a person sitting in wheel chair, holding a bag and talking in phone, back side there is another person pushing the wheel chair. |
| | 4 | In this image we can see a person sitting on the racing wheelchair. In the background there are people. At the bottom there is a road. |
| | 5 | In this image we can see a wheelchair. On the wheelchair there is a dog. Behind the wheelchair there are two people. In the background of the image there are trees, a shelter, glasses, table, chairs and ot... |
| Original | – | This image is taken indoors. In the middle of the image many people are sitting in the wheelchairs. We can see the floor. On the right side of the image a person is standing on the floor. In the background we can see the boards with text. We can see the chairs. We can see the railings. Many people are sitting on the chairs. We can see the stairs. There is a banner. |

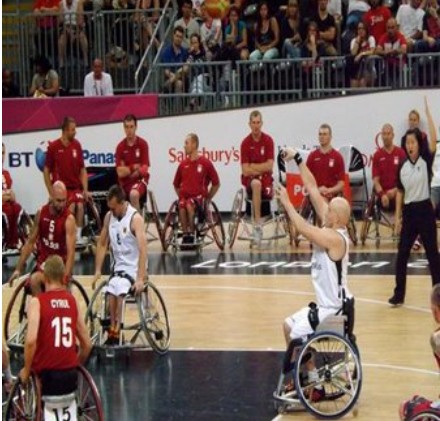

Table 17: The query image is used to retrieve the captions.

| Model | Rank | Caption |
|---|---|---|
| Global DINO | 1 | In this image, we can see a person wearing astronaut suit and he is wearing a hat and the background is dark. |
| | 2 | In this image, we can see an astronaut suit in front of the wall. |
| | 3 | In front of the image there is a space suit on the display, behind the suit there's a wall. |
| | 4 | In the center of the image we can see astronaut suit. In the background we can see wall, curtain and some drawing on the wall. At the top there is light. |
| | 5 | In the picture I can see a man wearing a spacesuit. |
| Local DINO | 1 | In this image there is a person standing and looking at the right side of the image, behind him there are few people. |
| | 2 | In this image there is a person standing. |
| | 3 | In this image we can see a person standing on the floor and a stand. |
| | 4 | In this image I can see there are few persons walking. |
| | 5 | In this image there are people standing, in the background there are clothes. |
| Global CLIP | 1 | In this image, we can see a person wearing astronaut suit and he is wearing a hat and the background is dark. |
| | 2 | In the picture I can see a man wearing a spacesuit. |
| | 3 | In the center of the image we can see astronaut suit. In the background we can see wall, curtain and some drawing on the wall. At the top there is light. |
| | 4 | In the image I can see space suits. |
| | 5 | In this image there is an astronaut suit which is visible. |
| Local CLIP | 1 | In this picture, we see two girls and the boys are standing. They might be exercising. On the right side, we see the legs of two people. At the bottom, we see the floor. |
| | 2 | In this image we can see a man and a woman standing. |
| | 3 | In this image we can see two persons holding each other and behind them, we can see a woman standing. |
| | 4 | In front of the image there is a woman, behind the woman there is a person standing. |
| | 5 | In this image we can see there are two women standing with smile, behind them there are few people. The background is dark. |
| Original | – | In this picture it looks like the cutouts of space suits holding a flag pole with 2 girls standing behind them. In the background, we can see other toys, trees, lights, games etc., |

Table 18: The query image 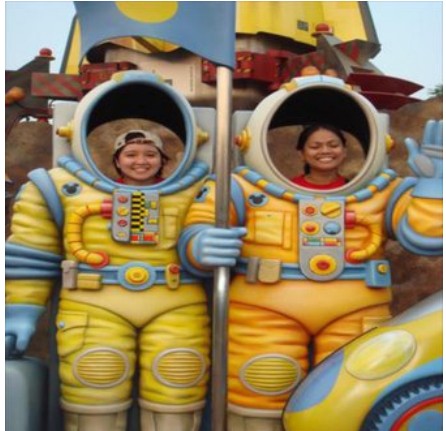 is used to retrieve the captions.

| Model | Rank | Caption |
|---|---|---|
| Global DINO | 1 | In this image in the center it looks like a toy train and there is a toy railway track, at the bottom it might be a floor. |
| | 2 | In this image we can see a scale model, we can see toy trains on the track, which is on the bridge, beneath that there is a tunnel, trees, a building made up of cardboard and a snow. In the background there is a wall and at the top of the image there is a ceiling with lights. |
| | 3 | In this image, we can see a train on the track and there are people inside the train. In the background, there are trees, railings, stairs, plants, flowers and there is a rock wall and a shed. On the left, we can see a pole. |
| | 4 | In this image, I can see a toy train on a toy railway track and there are few other toy railway tracks. In the top left side of the image, I can see an object on the surface. |
| | 5 | In this picture I can see a miniature set, where I can see the railway tracks, a tree and a overhead tank. I see that, it is blurred in the background. |
| Local DINO | 1 | In this image there is a person standing. |
| | 2 | In this image we can see a person standing on the floor and a stand. |
| | 3 | In this image there is a person standing and looking at the right side of the image, behind him there are few people. |
| | 4 | This image consists of a person. At the bottom, there are clothes to the legs. The person is standing on the floor. |
| | 5 | In the middle of the image a person is standing and watching. Behind him we can see a locomotive. |
| Global CLIP | 1 | This image consists of miniatures. In this image the we can see the colored background. In the middle of the image we can see the toy houses and buildings. We can see the toy trees and plants. We can see the railings. We can see the toys. |
| | 2 | In this image we can see miniature model. There are railway tracks. Also there is a train. And there is a building with windows. Also there is sky. |
| | 3 | There is a model of a train in the foreground area of the image and the background is white. |
| | 4 | In this image we can see a scale model, we can see toy trains on the track, which is on the bridge, beneath that there is a tunnel, trees, a building made up of cardboard and a snow. In the background there is a wall and at the top of the image there is a ceiling with lights. |
| | 5 | In this image in the center it looks like a toy train and there is a toy railway track, at the bottom it might be a floor. |
| Local CLIP | 1 | In this image there is a person standing and looking at the right side of the image, behind him there are few people. |
| | 2 | In the middle of the image a person is standing and watching. Behind him we can see a locomotive. |
| | 3 | In this image there is a person standing. |
| | 4 | In this image we can see rocks and a person on the land. |
| | 5 | In this image we can see a man standing and holding an object, before him there is an animal and we can see grass. On the left there are rocks. |
| Original | – | In this picture we can see a machine on a wooden platform. We can see poles, metal chains, a ladder and few objects. We can see rocks and there are stones on the ground. In the background we can see rock hills and the sky. |

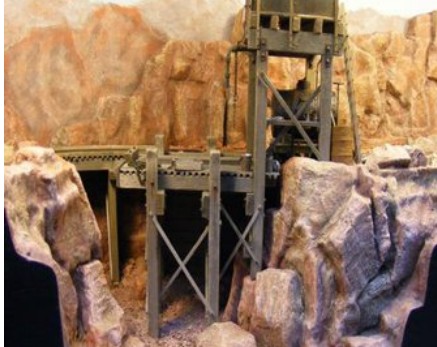

Table 19: The query image is used to retrieve the captions.

| Model | Rank | Caption |
|---|---|---|
| Global DINO | 1 | Here in this picture we can see a group of people standing over a place and we can see all of them are wearing ice skates, gloves and helmet and we can see they are standing on an ice floor and playing ice hockey, as we can see they are holding hockey sticks in their hands. |
| | 2 | In the center of the image we can see three people are ice skating and they are in different costumes. And we can see they are holding sticks and they are wearing helmets. In the background there is a wall, transparent glass, banners with some text and barriers. On one of the barriers, we can see an object. |
| | 3 | Here in this picture we can see a group of people skating on the ice floor with ice skate under their legs and we can see they are wearing gloves, helmet and holding hockey sticks in their hands and we can see they are playing ice hockey over there and on the right top side we can see a goal post with net present and we can see a person is standing near it with helmet, gloves, knee pads and ice skates and in the bottom we can see the glass walls present. |
| | 4 | Here in this picture we can see some people skating on the ice floor present over a place and we can see all of them are wearing ice skates, gloves, helmet and holding a ice hockey stick in their hand and behind them we can see hoarding present and we can also see fencing present and we can see number of people standing and sitting in the stands over there and watching the game. |
| | 5 | In this picture I can see a group of people wearing ice skating shoes and standing on the ice. There are few people holding hockey sticks. I can see a sports net with poles. |
| Local DINO | 1 | In this picture I can see a group of people wearing ice skating shoes and standing on the ice. There are few people holding hockey sticks. I can see a sports net with poles. |
| | 2 | In this image I can see two people with ice-skates and one person holding the stick. These people are on the ice. |
| | 3 | In this image we can see the people wearing the helmet and holding the hockey sticks and playing the ice hockey. In the background we can see some person's legs. |
| | 4 | In this image, there is a man wearing helmet and seems like he is playing ice hockey. At the top, there are legs of a person. |
| | 5 | In this image, I can see a person standing on ice with a helmet, gloves, ice skates and holding an ice hockey stick. In the background, I can see a group of people standing and there are walls. |
| Global CLIP | 1 | Here in this picture we can see some people skating on the ice floor present over a place and we can see all of them are wearing ice skates, gloves, helmet and holding a ice hockey stick in their hand and behind them we can see hoarding present and we can also see fencing present and we can see number of people standing and sitting in the stands over there and watching the game. |
| | 2 | In this picture we can see a man and a woman wearing ice skates visible on the floor. We can see the lights, stairs and other things. We can see a few people and the dark view in the background. |
| | 3 | Here in this picture we can see a group of people standing over a place and we can see all of them are wearing ice skates, gloves and helmet and we can see they are standing on an ice floor and playing ice hockey, as we can see they are holding hockey sticks in their hands. |
| | 4 | In this image we can see children doing ice skating on ice. In the back there is a wall. |
| | 5 | In this image we can see two persons ice skating. We can see a person holding a hockey stick. In the background, we can see people, boards with text, wall, poles and some objects. |
| Local CLIP | 1 | In this image there is a person standing and looking at the right side of the image, behind him there are few people. |
| | 2 | In this image we can see a person standing on the floor and a stand. |
| | 3 | In this picture, we see two girls and the boys are standing. They might be exercising. On the right side, we see the legs of two people. At the bottom, we see the floor. |
| | 4 | This image consists of a person. At the bottom, there are clothes to the legs. The person is standing on the floor. |
| | 5 | In this image there are people standing, in the background there are clothes. |
| Original | – | In the picture I can see the design floor and there are objects on the floor. I can see the lights at the top of the picture. I can see the logos on the wall and there are glass windows on the top left side of the picture. |

Table 20: The query image 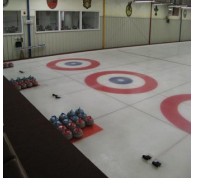 is used to retrieve the captions.

**MS COCO dataset's Image -> Caption Retrieval**

Tables 21, 22, 23, 24, and 25 show results on the MS COCO validation set.

| Model | Rank | Caption |
|---|---|---|
| Global DINO | 1 | A kite being flown in the middle of a beach. |
| | 2 | People flying kites on a sandy beach while a bucket sits in the sand. |
| | 3 | Kites being used by people on a beach. |
| | 4 | A group of people flying kites at the beach |
| | 5 | Two people on a beach flying a kite in the air. |
| Local DINO | 1 | A kite being flown in the middle of a beach. |
| | 2 | People flying kites on a sandy beach while a bucket sits in the sand. |
| | 3 | A person standing on top of a beach flying a kite. |
| | 4 | Kites being used by people on a beach. |
| | 5 | A shot of the blue water with people flying a kite. |
| Global CLIP | 1 | A kite being flown in the middle of a beach. |
| | 2 | A person standing on top of a beach flying a kite. |
| | 3 | A man is flying a kite at the beach. |
| | 4 | a man is flying a kite at on the shore at the beach |
| | 5 | Kites being used by people on a beach. |
| Local CLIP | 1 | A person standing on top of a beach flying a kite. |
| | 2 | A kite being flown in the middle of a beach. |
| | 3 | People flying kites on a sandy beach while a bucket sits in the sand. |
| | 4 | A man is flying a kite at the beach. |
| | 5 | A man flying a kite on a beach with people standing around. |
| Original | – | A man is flying a kite at the beach. |

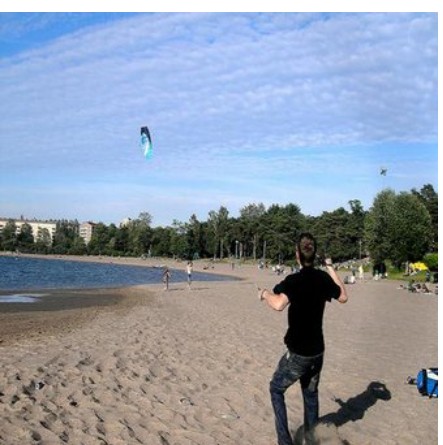

Table 21: The query image is used to retrieve the captions. Green color is for the original caption from the dataset.

| Model | Rank | Caption |
|---|---|---|
| Global DINO | 1 | A giraffe is walking in some tall grass |
| | 2 | A giraffe standing on a grass covered field. |
| | 3 | A single giraffe looks over the green brush. |
| | 4 | there is a very tall giraffe standing in the wild |
| | 5 | a giraffe in a field with trees in the background |
| Local DINO | 1 | A giraffe standing on a grass covered field. |
| | 2 | A tall giraffe standing on top of a grass covered field. |
| | 3 | there is a very tall giraffe standing in the wild |
| | 4 | A giraffe is walking in some tall grass |
| | 5 | A giraffe standing by a pair of skinny trees. |
| Global CLIP | 1 | A giraffe stands near a tree in the wilderness. |
| | 2 | A giraffe is walking in some tall grass |
| | 3 | A giraffe standing on a grass covered field. |
| | 4 | A group of giraffes that are standing in the grass. |
| | 5 | there is a very tall giraffe standing in the wild |
| Local CLIP | 1 | A tall giraffe standing on top of a grass covered field. |
| | 2 | A giraffe standing on a grass covered field. |
| | 3 | A giraffe is walking in some tall grass |
| | 4 | A single giraffe looks over the green brush. |
| | 5 | there is a very tall giraffe standing in the wild |
| Original | – | A single giraffe looks over the green brush. |

Table 22: The query image 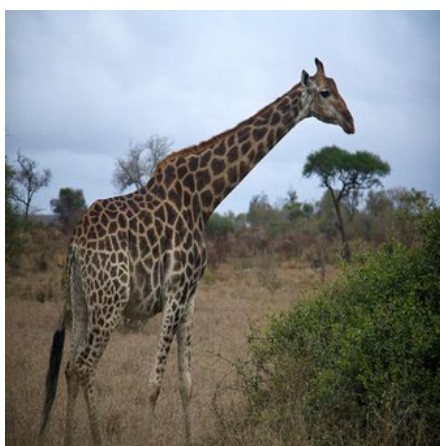 is used to retrieve the captions. Green color is for the original caption from the dataset.

| Model | Rank | Caption |
|-------|------|---------|
| Global DINO | 1 | A man riding a surfboard on a wave in the ocean. |
| | 2 | A surfer in the ocean trying not to wipeout. |
| | 3 | A man on a surfboard riding a wave in the ocean. |
| | 4 | a person riding a skate board on a wave |
| | 5 | A man is surfing on a small wave. |
| Local DINO | 1 | A group of people swimming in the ocean with a surfboard. |
| | 2 | there are many surfers that are in the water |
| | 3 | Pair of surfers paddling out to open ocean. |
| | 4 | A man riding on the back of a surfboard next to kids. |
| | 5 | a man on a blue surfboard on top of some rough water |
| Global CLIP | 1 | A man riding a surfboard on a wave in the ocean. |
| | 2 | A man on a surfboard riding a wave in the ocean. |
| | 3 | a person riding a skate board on a wave |
| | 4 | a person riding a surf board on a wave |
| | 5 | A man on a surfboard, who is riding a wave. |
| Local CLIP | 1 | A person on a surfboard in the water. |
| | 2 | A para sailor with his board with sail in the surf. |
| | 3 | A group of people swimming in the ocean with a surfboard. |
| | 4 | A man riding on the back of a surfboard next to kids. |
| | 5 | Pair of surfers paddling out to open ocean. |
| Original | – | a person riding a surf board on a wave |

Table 23: The query image 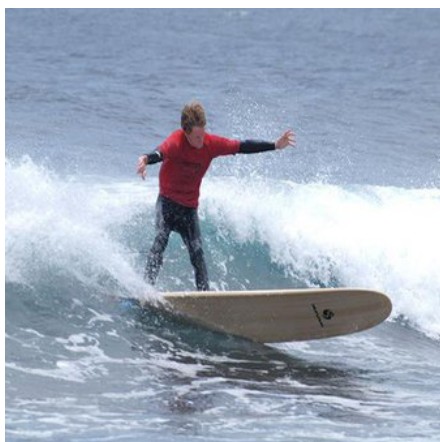 is used to retrieve the captions. Green color is for the original caption from the dataset.

| Model | Rank | Caption |
|---|---|---|
| Global DINO | 1 | A brown bear walking with rocks in the background. |
| | 2 | A large brown bear standing next to a pile of rocks. |
| | 3 | A big burly grizzly bear is show with grass in the background. |
| | 4 | A majestic bear looks out across a grass plain. |
| | 5 | a brown bear is walking away from a river |
| Local DINO | 1 | A baby brown bear standing on top of a rock. |
| | 2 | A majestic bear looks out across a grass plain. |
| | 3 | A brown bear walking with rocks in the background. |
| | 4 | A statue of a large brown bear tearing off a cars door. |
| | 5 | A brown bear lays down in the woods. |
| Global CLIP | 1 | A brown bear walking with rocks in the background. |
| | 2 | A large brown bear standing next to a pile of rocks. |
| | 3 | A big burly grizzly bear is show with grass in the background. |
| | 4 | A baby brown bear standing on top of a rock. |
| | 5 | A majestic bear looks out across a grass plain. |
| Local CLIP | 1 | A baby brown bear standing on top of a rock. |
| | 2 | A majestic bear looks out across a grass plain. |
| | 3 | A brown bear walking with rocks in the background. |
| | 4 | A big burly grizzly bear is show with grass in the background. |
| | 5 | A large brown bear standing next to a pile of rocks. |
| Original | – | A brown bear walking with rocks in the background. |

Table 24: The query image 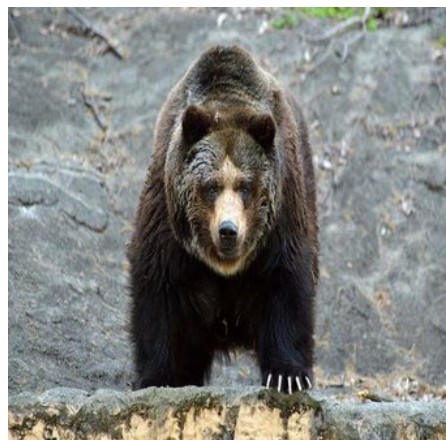 is used to retrieve the captions. Green color is for the original caption from the dataset.

| Model | Rank | Caption |
|---|---|---|
| Global DINO | 1 | Person cooking an eggs on a black pot on a stove. |
| | 2 | A man pokes his head in front of an oven open to baking cookies. |
| | 3 | Belgium waffle loaded with bananas topped with powdered sugar with syrup and more fruit as a garnish. |
| | 4 | Several breakfast foods are on top of a refrigerator. |
| | 5 | A plate has a waffle, some fruit and ice cream on it. |
| Local DINO | 1 | A person is holding a spatula near slices of bread on a stove. |
| | 2 | Twp cake pans sitting and cooling on the stove |
| | 3 | an image of a man slicing a small pizza |
| | 4 | A woman observing something on a kitchen stove. |
| | 5 | A man pokes his head in front of an oven open to baking cookies. |
| Global CLIP | 1 | A pastry station, with an assortment of fillings and sauces |
| | 2 | A young man is working behind a counter. |
| | 3 | A group of three chefs preparing food in a kitchen. |
| | 4 | A man preparing food in a restaurant kitchen. |
| | 5 | The donut robot machine is mechanically making donuts. |
| Local CLIP | 1 | A table with many different objects, including a plate of sandwiches. |
| | 2 | A pastry station, with an assortment of fillings and sauces |
| | 3 | A table topped with plates, bowls and containers of food. |
| | 4 | A buffet of casserole dishes on a kitchen counter. |
| | 5 | A bunch of items that are on a counter. |
| Original | – | A pastry station, with an assortment of fillings and sauces |

Table 25: The query image 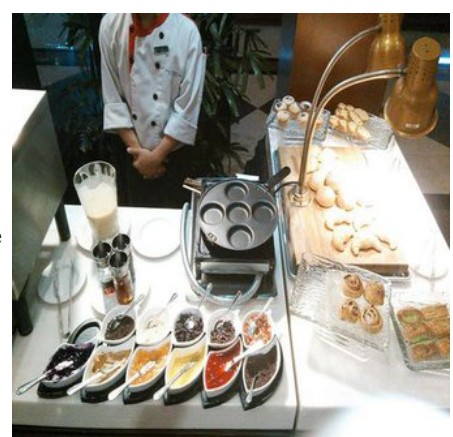 is used to retrieve the captions. Green color is for the original caption from the dataset.

## H.2 Caption → Image Retrieval (In-Distribution)

We evaluate cross-modal alignment through caption-to-image retrieval using SPARC latent representations. Figures 31, 32, 33, and 34 show retrieval results comparing Global vs Local TopK training configurations.

The four model configurations represent:

- **Global CLIP**: Query caption's CLIP-text features → Reference database of CLIP-image features (both from Global SPARC)

- **Local CLIP**: Query caption's CLIP-text features → Reference database of CLIP-image features (both from Local SPARC)

- **Global DINO**: Query caption's CLIP-text features → Reference database of DINO features (both from Global SPARC)

- **Local DINO**: Query caption's CLIP-text features → Reference database of DINO features (both from Local SPARC)

Each model shows top-10 retrieved images ranked by cosine similarity in the SPARC latent space.

### Open Images dataset's Image -> Caption Retrieval

Figures 31 and 32 show results on the Open Images test set using diverse query captions.

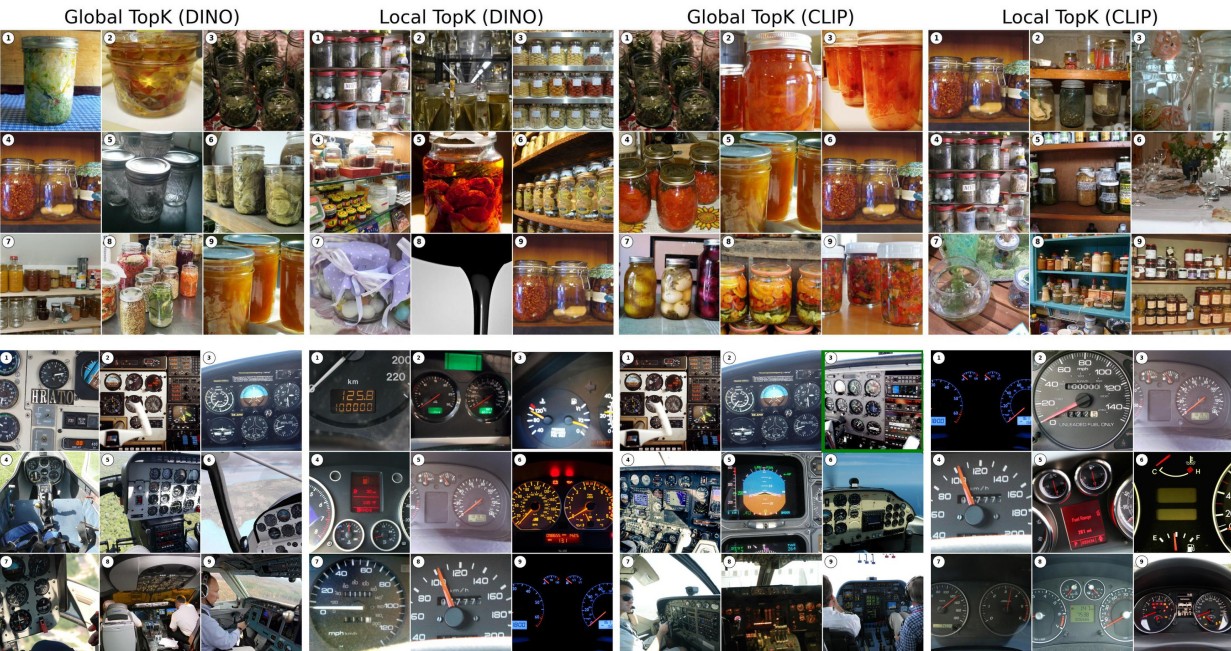

Figure 31: Images retrieved for captions are (1) "In this picture we can see some food products in the glass jars.", (2) "In this image might be taken in the airplane. In this image we can see the speedometers, knobs and some digital screens." Captions are from Open Images test dataset and images are retrieved from the same split. Green boxes indicate when the corresponding image for a caption is successfully retrieved. The second caption shows such a match (Global TopK with CLIP, 3rd rank).

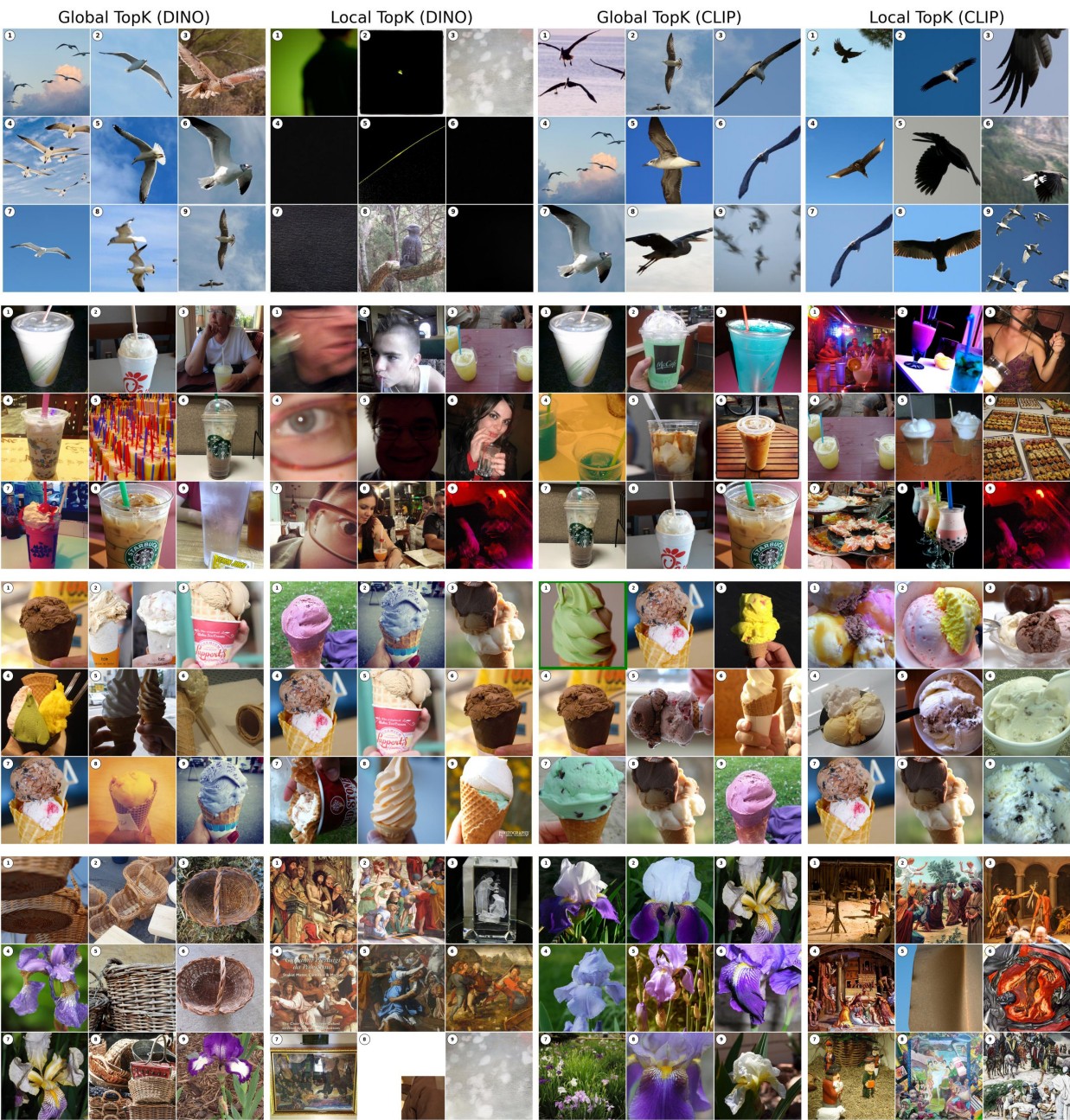

Figure 32: Images retrieved for captions are (1) "In this image in the center there is one bird flying, and in the background there is sky.", (2) "In this image we can see a table on which some glasses are there in which some food items and straws are there and we can see a pot like structure. On the left side we can see a person hand. In the background we can see some posters and bottles.", (3) "In the picture we can see an ice cream with a green and brown color cream.", and (4) "In this image we can see a wooden basket placed on the ground, we can also see the photo frame on the wall." Captions are from Open Images test dataset and images are retrieved from the same split. Green boxes indicate when the corresponding image for a caption is successfully retrieved. The third caption shows such a match (Global TopK with CLIP, 1st rank).

**MS COCO dataset's Image -> Caption Retrieval**

Figures 33 and 34 show results on the MS COCO validation set. We show 1 of the 5 captions available per image in MS COCO in our results.

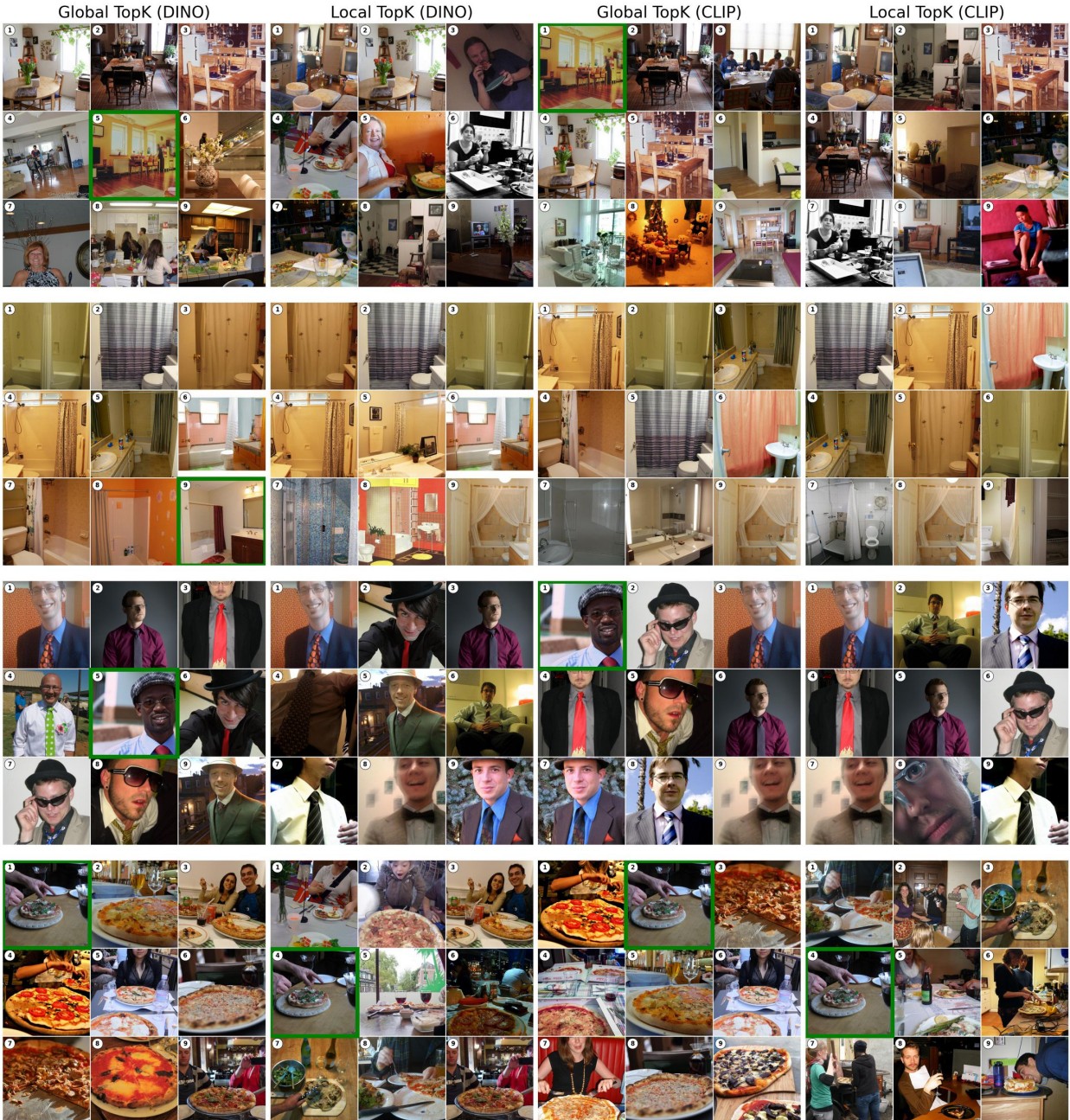

Figure 33: Images retrieved for captions are (1) "A woman stands in the dining area at the table.", (2) "A shower curtain sits open in an empty and clean bathroom.", (3) "a man in a blue shirt and red tie.", and (4) "These people are going to have pizza and wine." Captions are from Open Images test dataset and images are retrieved from the same split. Green boxes indicate when the corresponding image for a caption is successfully retrieved.

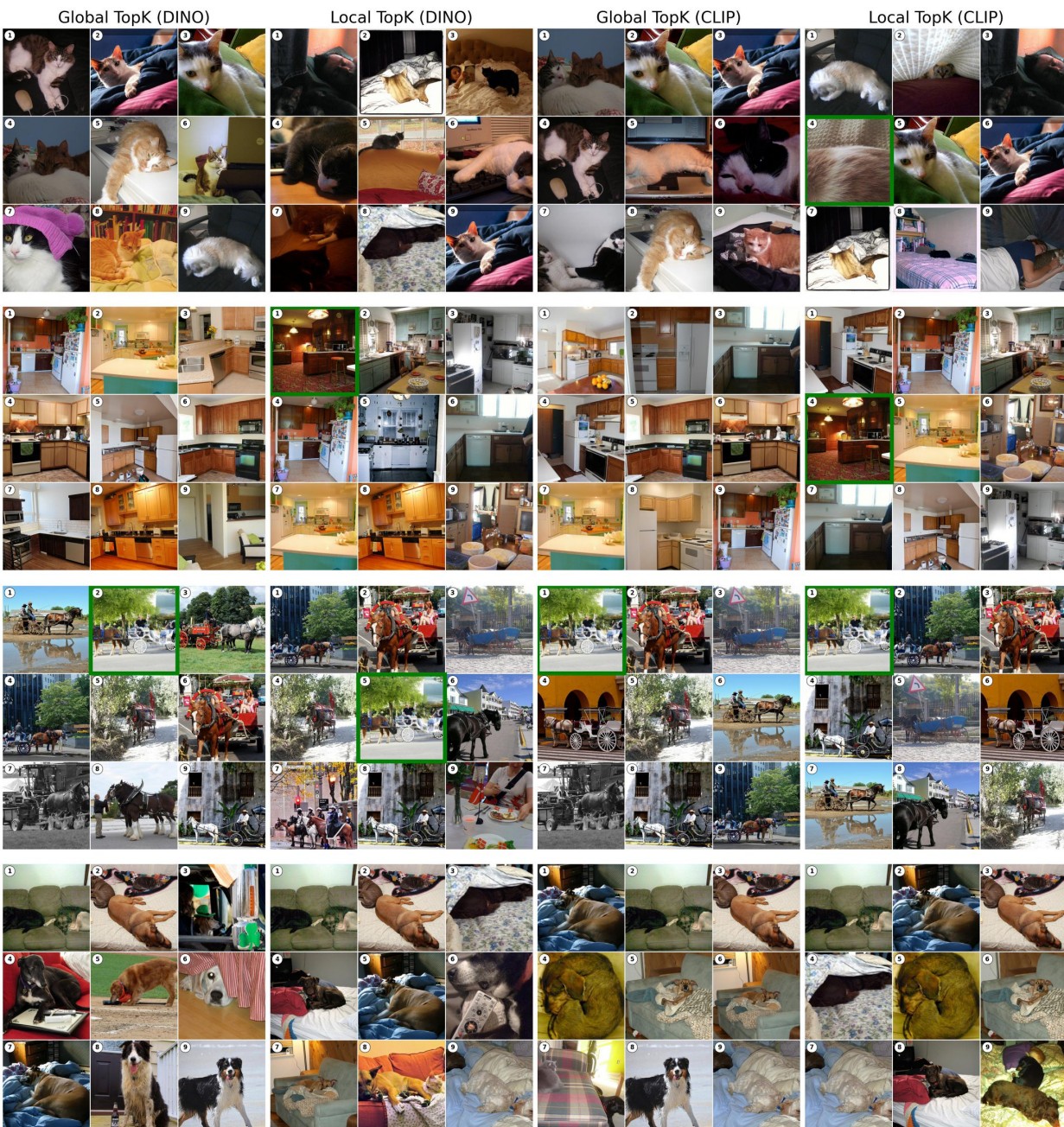

Figure 34: Images retrieved for captions are (1) "White and orange fur lays on a white blanket.", (2) "A kitchen that has carpeted floors and wooden cabinets.", (3) "A street scene with a horse pulling a white carriage.", and (4) "A large dogs comfortably sleeping on someones bed". Captions are from Open Images test dataset and images are retrieved from the same split. Green boxes indicate when the corresponding image for a caption is successfully retrieved.

### H.3   Image -> Image Retrieval (In-Distribution)

We evaluate cross-model alignment by retrieving images across DINO and CLIP-image encoders using SPARC latent representations. Figures 35, 36, 37, and 38 show results in a 4-row layout comparing Global vs Local TopK training configurations.

The four rows represent:

- **DINO Global**: Query image's DINO features → Reference database of CLIP features (both from Global SPARC)

- **DINO Local**: Query image's DINO features → Reference database of CLIP features (both from Local SPARC)

- **CLIP Global**: Query image's CLIP features → Reference database of DINO features (both from Global SPARC)

- **CLIP Local**: Query image's CLIP features → Reference database of DINO features (both from Local SPARC)

Each row shows the query image (left) followed by top-10 retrieved images.

**Open Images dataset's Image -> Image Retrieval**

Figures 35, 36, and 37 show cross-model retrieval results on the Open Images test set.

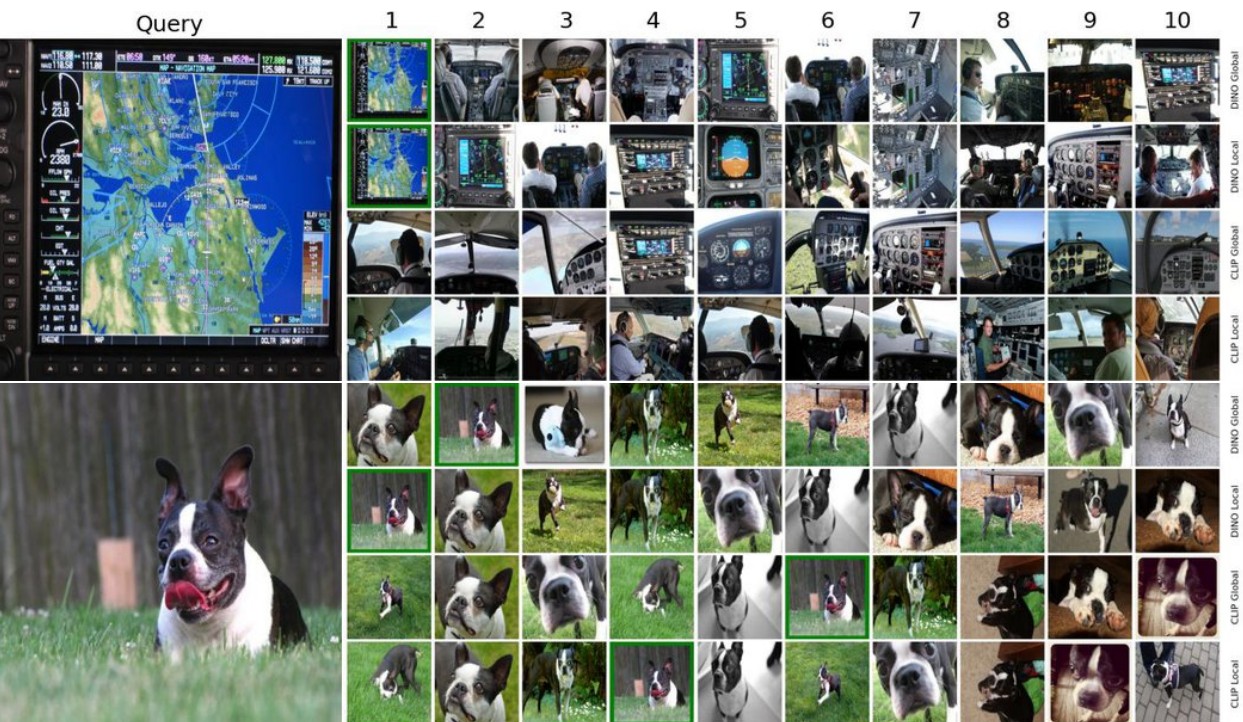

Figure 35: Cross-model image retrieval results. Each image shows a 4-row layout comparing query stream and reference database combinations. Query image (left) with top-10 retrieved images from reference database (right). All models trained with $\lambda = 1$ on Open Images training set. The retrieval is done on the test set of Open Images. Green border is used to show the exact match of the query image was found.

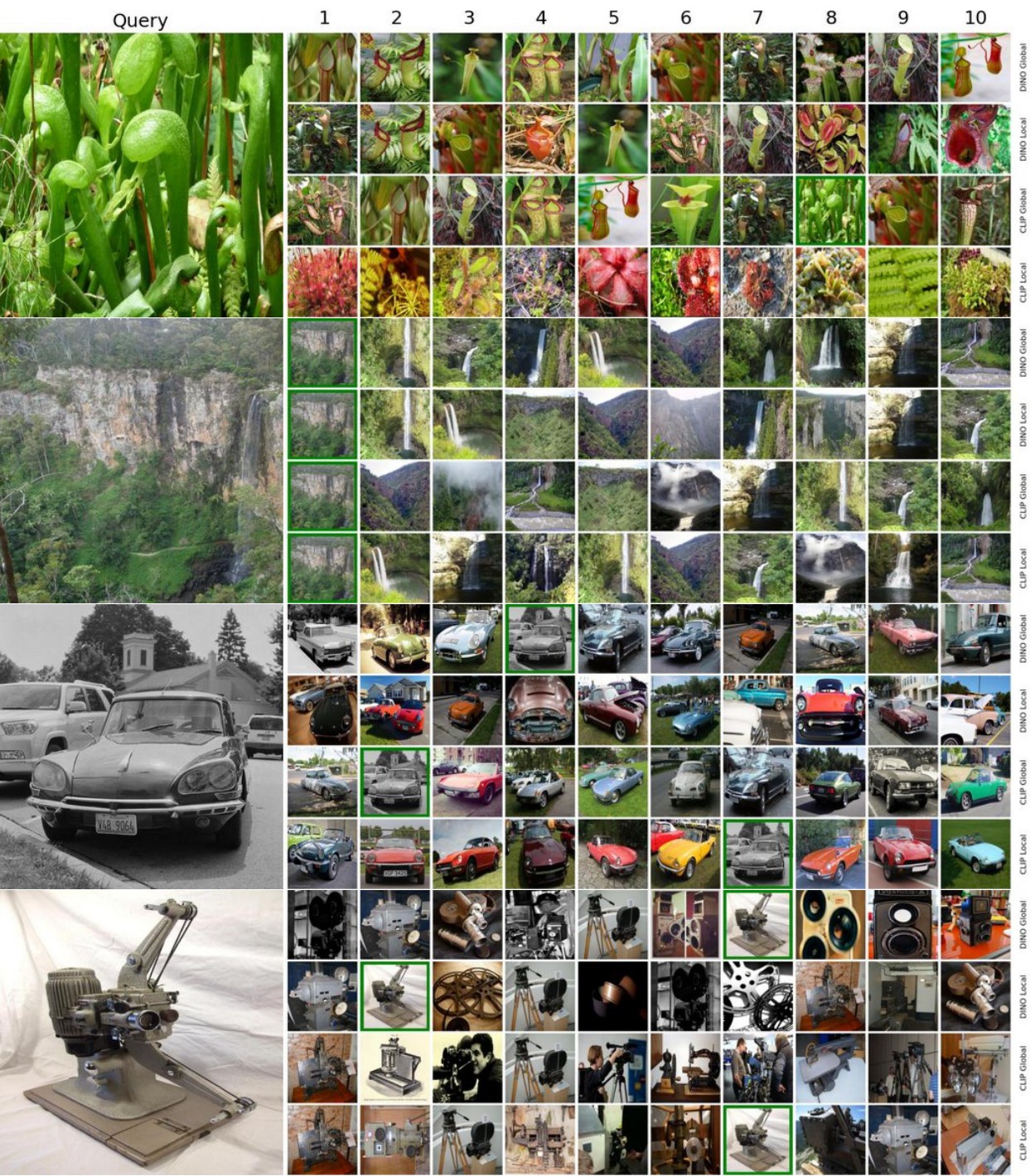

Figure 36: Cross-model image retrieval results. Each image shows a 4-row layout comparing query stream and reference database combinations. Query image (left) with top-10 retrieved images from reference database (right). All models trained with $\lambda = 1$ on Open Images training set. The retrieval is done on the test set of Open Images. Green border is used to show the exact match of the query image was found.

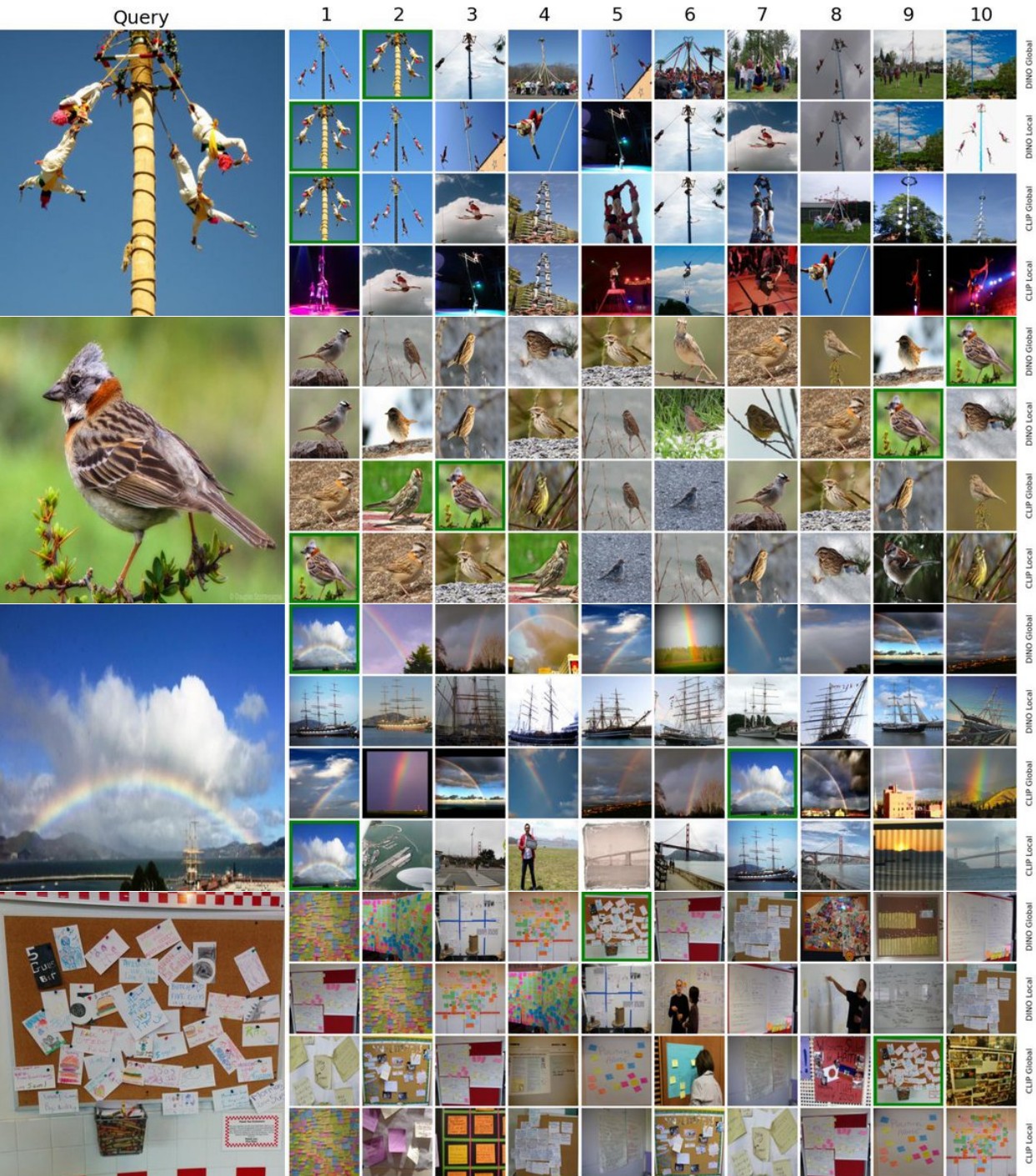

Figure 37: Cross-model image retrieval results. Each image shows a 4-row layout comparing query stream and reference database combinations. Query image (left) with top-10 retrieved images from reference database (right). All models trained with $\lambda = 1$ on Open Images training set. The retrieval is done on the test set of Open Images. Green border is used to show the exact match of the query image was found.

**MS COCO dataset's Image -> Image Retrieval** Figure 38 shows cross-model image-to-image retrieval results on the MS COCO validation set.

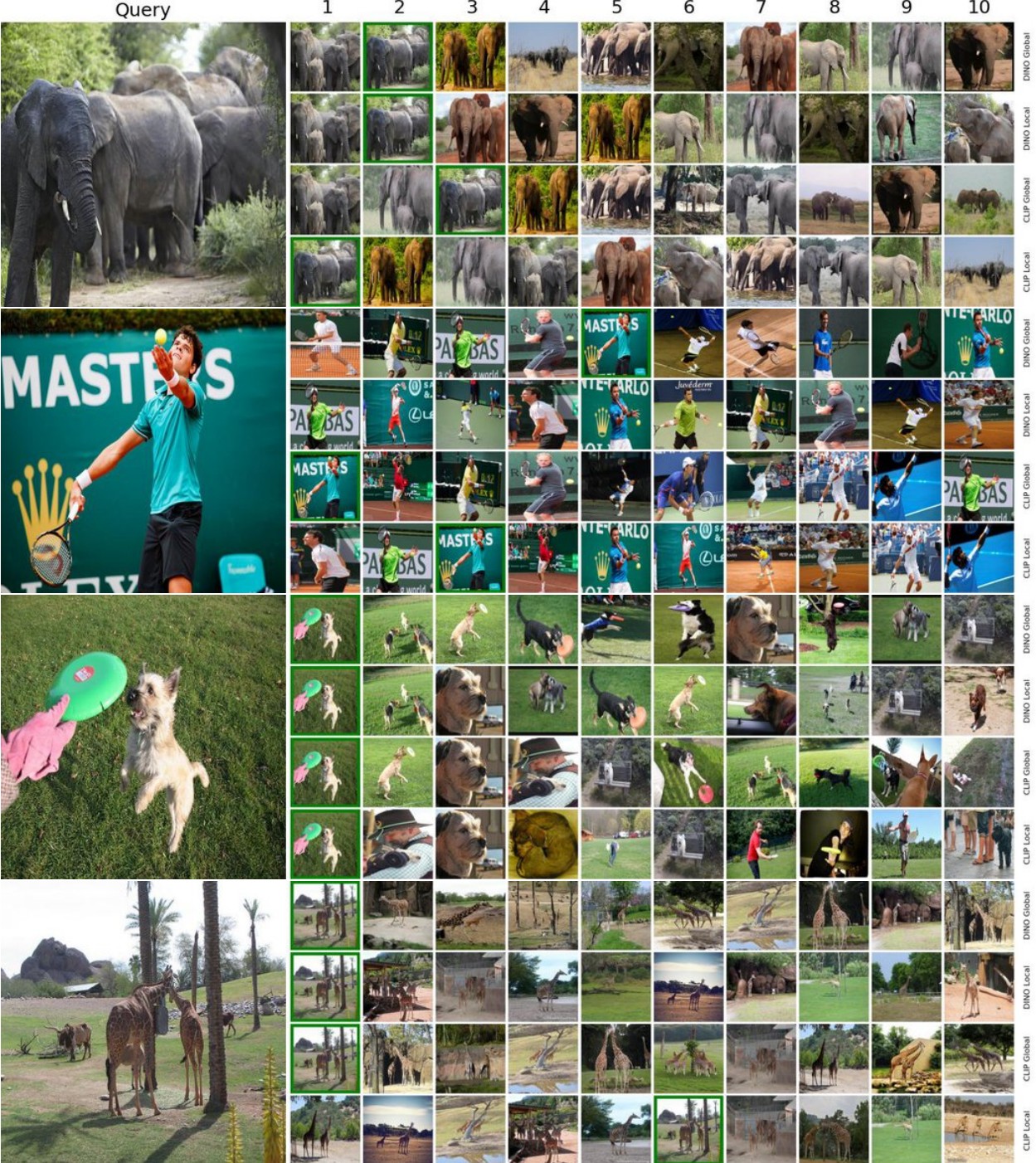

Figure 38: Cross-model image retrieval results. Each image shows a 4-row layout comparing query stream and reference database combinations. Query image (left) with top-10 retrieved images from reference database (right). All models trained with $\lambda = 1$ on MS COCO training set. The retrieval is done on the validation set of MS COCO. Green border is used to show the exact match of the query image was found.

### H.4 External Image → Caption (OOD)

We evaluate SPARC's out-of-distribution generalization using external images not present in the Open Images dataset, following the same methodology as Section H.1. Tables 26, 27, 28, 29, 30, 31, and 32 show top-5 retrieved captions for each model configuration using external query images with the Open Images test set as the reference database.

| Model | Rank | Retrieved Caption |
|---|---|---|
| Global DINO | 1 | In this image we can see a laptop with a screen and keys. At the bottom of the image there is a surface. On the surface we can see reflections. On the image there is a watermark. |
| | 2 | In this image we can see black and white picture of a laptop, we can also see some pictures on the screen. |
| | 3 | In this image there is a poster. There is a screen, sound speakers, laptop on the devices. Bottom of the image there is some text. Background is in grey color. Top of the image there is some text. |
| | 4 | In this image we can see a laptop containing some text on its screen. |
| | 5 | In this picture we can see keypad of laptop. |
| Local DINO | 1 | In this image we can see a laptop. |
| | 2 | In this picture I can observe black color joystick on the laptop. There is an apple logo on the laptop. The background is in white color. |
| | 3 | In this image we can see a laptop containing some text on its screen. |
| | 4 | There is a white color laptop present on the cloth as we can see in the middle of this image. There is one device connected to the laptop. It is dark in the background. |
| | 5 | In the center of the image we can see the text. In the background of the image we can see a laptop. On the laptop we can see the objects. |
| Global CLIP | 1 | This image is taken indoors. In the background there is a wall. In the middle of the image there are two laptops on the table. |
| | 2 | In this picture we can see laptop, keys and screen. In the background of the image it is blurry. |
| | 3 | In this image there is a laptop on the table. There are letters, numbers, symbols on the keyboard. There is some text at the top of the image. |
| | 4 | In this picture we can see a laptop with keys and on this laptop screen we can see text, symbols and buttons. |
| | 5 | In this picture we can see a laptop. We can see a person, numbers, a symbol and a few things on the screen of this laptop. |
| Local CLIP | 1 | In this picture it looks like a laptop. We see the laptop keyboard which has the alphabet and the number keys. We see the text written on the laptop and we see the buttons. At the bottom, we see the touchpad on the laptop. |
| | 2 | In this picture we can see a laptop with keys and on this laptop screen we can see text, symbols and buttons. |
| | 3 | In the image there is a laptop on a cloth and in front of the laptop there is a teddy bear and it seems like both the things are kept on a sofa and in the background there is a wall. |
| | 4 | In this picture we can see a laptop. We can see a person, numbers, a symbol and a few things on the screen of this laptop. |
| | 5 | In this image, I can see a laptop. On which I can see keys having letters, numbers, symbols and some text. |

Table 26: The query image 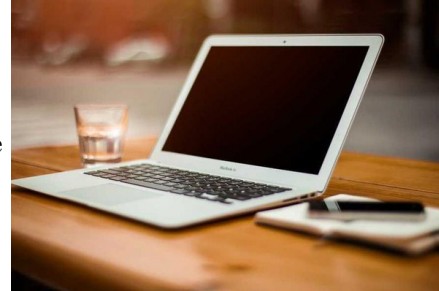 is used to retrieve the captions.

| Model | Rank | Retrieved Caption |
|---|---|---|
| Global DINO | 1 | In this image we can see a laptop with a screen and keys. At the bottom of the image there is a surface. On the surface we can see reflections. On the image there is a watermark. |
| | 2 | In this image we can see black and white picture of a laptop, we can also see some pictures on the screen. |
| | 3 | In this image there is a poster. There is a screen, sound speakers, laptop on the devices. Bottom of the image there is some text. Background is in grey color. Top of the image there is some text. |
| | 4 | In this image we can see a laptop containing some text on its screen. |
| | 5 | In this picture we can see keypad of laptop. |
| Local DINO | 1 | In this image we can see a laptop. |
| | 2 | In this picture I can observe black color joystick on the laptop. There is an apple logo on the laptop. The background is in white color. |
| | 3 | In this image we can see a laptop containing some text on its screen. |
| | 4 | There is a white color laptop present on the cloth as we can see in the middle of this image. There is one device connected to the laptop. It is dark in the background. |
| | 5 | In the center of the image we can see the text. In the background of the image we can see a laptop. On the laptop we can see the objects. |
| Global CLIP | 1 | This image is taken indoors. In the background there is a wall. In the middle of the image there are two laptops on the table. |
| | 2 | In this picture we can see laptop, keys and screen. In the background of the image it is blurry. |
| | 3 | In this image there is a laptop on the table. There are letters, numbers, symbols on the keyboard. There is some text at the top of the image. |
| | 4 | In this picture we can see a laptop with keys and on this laptop screen we can see text, symbols and buttons. |
| | 5 | In this picture we can see a laptop. We can see a person, numbers, a symbol and a few things on the screen of this laptop. |
| Local CLIP | 1 | In this picture it looks like a laptop. We see the laptop keyboard which has the alphabet and the number keys. We see the text written on the laptop and we see the buttons. At the bottom, we see the touchpad on the laptop. |
| | 2 | In this picture we can see a laptop with keys and on this laptop screen we can see text, symbols and buttons. |
| | 3 | In the image there is a laptop on a cloth and in front of the laptop there is a teddy bear and it seems like both the things are kept on a sofa and in the background there is a wall. |
| | 4 | In this picture we can see a laptop. We can see a person, numbers, a symbol and a few things on the screen of this laptop. |
| | 5 | In this image, I can see a laptop. On which I can see keys having letters, numbers, symbols and some text. |

Table 27: The query image 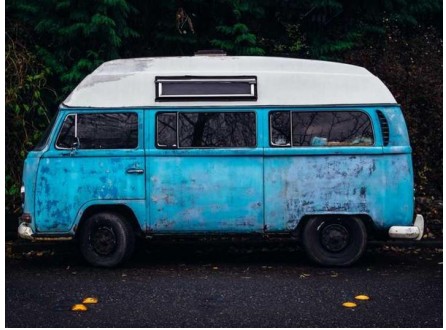 is used to retrieve the captions.

| Model | Rank | Retrieved Caption |
|---|---|---|
| Global DINO | 1 | In this picture we can see a few people, snowy mountains and hills. We can see other things and the cloudy sky in the background. |
| | 2 | In the picture we can see the mountains covered with the snow and behind it, we can see the sky with clouds. |
| | 3 | In this picture we can see the man wearing a black jacket and standing. In the front we can see some stones. Behind we can see mountains and some snow. On the top we can see the sky and clouds. |
| | 4 | In this picture I can see few people. There are rocks. I can see snowy mountains, and in the background there is the sky. |
| | 5 | In this image we can see mountains with snow. In the background there is sky. |
| Local DINO | 1 | In this picture I can see mountains and I can see snow and a blue cloudy sky. I can see text at the bottom left corner of the picture and looks like a cross symbol on the mountain. |
| | 2 | In the image we can see the person standing, wearing clothes, gloves, shoes and spectacles, here we can see the stones, snow, mountains and the sky. |
| | 3 | In this picture, we see the man in the jacket is standing. He is wearing a red cap, spectacles and the gloves. On the left side, we see the snow. In the background, we see the hills. These hills are covered with the snow. |
| | 4 | In this image we can see a man in blue color jacket is siting on a snow covered ground and he is also wearing a white color cap and a gaggle as well. We can also see a bag and a stick on the snow covered ground. In background we can see a mountain and a clear blue sky. |
| | 5 | In this picture I can see there is a person and he is wearing a coat and a bag pack, there are a few mountains in the background and they are covered with rocks and snow. |
| Global CLIP | 1 | In this picture I can see mountains and I can see snow and a blue cloudy sky. I can see text at the bottom left corner of the picture and looks like a cross symbol on the mountain. |
| | 2 | In the foreground we can see rocks, mountain, soil and people. In the middle of the image there are mountains. In the background it is the sky. |
| | 3 | In this image I can see a person standing on the mountain, he is wearing a bag. There are few rocks on the mountain, I can see there are few mountains in the background and the sky is clear. |
| | 4 | In this image we can see so many rocks and snow on the mountain. In background we can see some more mountains and a clear blue sky. |
| | 5 | In this picture I can see few people. There are rocks. I can see snowy mountains, and in the background there is the sky. |
| Local CLIP | 1 | In this image I can see some people on the mountain, and there is snow on the mountain and the sky is blue. |
| | 2 | In this picture we can see a few people, snowy mountains and hills. We can see other things and the cloudy sky in the background. |
| | 3 | In this picture I can see a person standing in a foreground of the image and there are a few rocks visible, in the background there is a mountain visible and there are few trees and there is snow covered on the mountain and I can see the sky. |
| | 4 | In this image in the center there are a group of people who are standing, and they are wearing bags. And at the bottom there is snow and one stick is there, and in the background there are mountains and at the top of the image there is sky. |
| | 5 | In the image there are few people sitting on the rocks. And there is a person standing on the rocks. In the background there is a hill covered with snow. And also there are few hills in the background. At the top of the image there is sky. |

Table 28: The query image 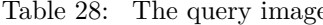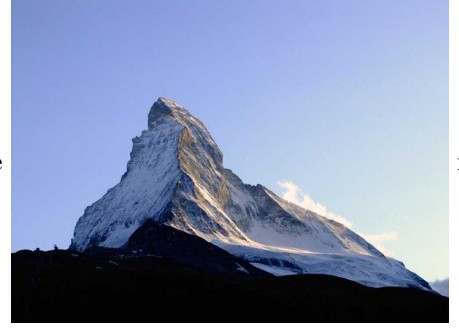 is used to retrieve the captions.

| Model | Rank | Retrieved Caption |
|---|---|---|
| Global DINO | 1 | In this picture I can see a sea otter in the water and looks like a rock at the top left corner. |
| | 2 | In the middle of this image I can see a sea otter in the water. At the bottom, I can see the sticks and leaves in the water. |
| | 3 | In this picture, we see a sea otter is swimming in the water. In the background, we see the water and this water might be in the swimming pool. |
| | 4 | At the bottom of the picture, we see the rocks. In front of the picture, we see a walrus is sleeping on the rock. Behind that, we see water and this water might be in the pond. There are rocks and a walrus in the background. |
| | 5 | In this image we can see a sea lion in the water. |
| Local DINO | 1 | In this image we can see a paper. On the paper we can see painting of birds on branch. Also we can see leaves. And there is text on the paper. |
| | 2 | In this image, we can see painting of parrots on the branch and some text at the bottom. |
| | 3 | In this image, we can see a water animal picture on a cream surface. At the top and bottom of the image, we can see text. |
| | 4 | In this image we can see a red color crab with two sticks on a paper on which something is written. |
| | 5 | In this picture I can see the bactrian camel in the foreground. It is looking like the green grass, plants in the background. It is looking like the fence on the right side. |
| Global CLIP | 1 | In this image we can see a sea lion in the water. |
| | 2 | In the image we can see water, on the water we can see some rafts. On the rafts we can see some seals. |
| | 3 | In this image I can see two sea lions in the water. |
| | 4 | In this image we can see sea lion in the water. |
| | 5 | In this image we can see sea lion in the water. |
| Local CLIP | 1 | In this image I can see the hippopotamus and the rock in the water. |
| | 2 | In this image we can see hippopotamus. Also we can see water. In the back we can see stones. |
| | 3 | In this picture, we see a hippopotamus is in the water. It might be a pool. At the bottom, we see a wall. In the right top, we see the railing and a wall. We see the rods or the stands in the pool. |
| | 4 | In this image we can see hippopotamus in water and ground. In the background we can see trees. |
| | 5 | In this image I can see an animal in the water, looks like hippopotamus. |

Table 29: The query image 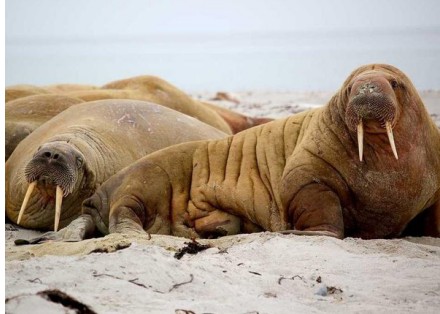 is used to retrieve the captions.

| Model | Rank | Retrieved Caption |
|---|---|---|
| Global DINO | 1 | In this image there is ground at the bottom. There are rocks in the foreground. And there is greenery in the background. And there is sky at the top. |
| | 2 | In this image there are few rocks, grass, a river, tree and in the background there is the sky. |
| | 3 | In this picture we can see rocks, trees and in the background we can see the sky with clouds. |
| | 4 | In this image we can see rocks. On the ground there is grass. In the background there is sky. |
| | 5 | In this image we can see rocks. Also there are people. In the background there is sky. |
| Local DINO | 1 | This image consists of stones, rocks, grass, a group of people, trees and the sky. |
| | 2 | In this image we can see stones. Also we can see grass on the ground. There are trees. In the background there is sky with clouds. |
| | 3 | In this picture we can see stones and here we can see the grass and trees on the ground. In the background we can see the sky. |
| | 4 | In this image we can see rocks. On the ground there is grass. In the background there is sky. |
| | 5 | In this picture, we see the stones, rocks and the grass. In the background, we see the stones and the rocks. |
| Global CLIP | 1 | In this image we can see stones. Also we can see grass on the ground. There are trees. In the background there is sky with clouds. |
| | 2 | In this image we can see rocks. Also there is water. On the left side we can see grass on the ground. In the background there is sky with clouds. |
| | 3 | In this picture we can see stones and here we can see the grass and trees on the ground. In the background we can see the sky. |
| | 4 | In this image we can see in front there are many rocks, trees, at the back there are hills, mountains, the sky is at the top. |
| | 5 | There are stones and grassland in the foreground area of the image, there are people, trees, grassland and the sky in the background. |
| Local CLIP | 1 | In this image we can see rocks. On the ground there is grass. In the background there is sky. |
| | 2 | In this picture, we see the stones, rocks and the grass. In the background, we see the stones and the rocks. |
| | 3 | This image consists of stones, rocks, grass, a group of people, trees and the sky. |
| | 4 | There are stones and grassland in the foreground area of the image, there are people, trees, grassland and the sky in the background. |
| | 5 | In this picture we can see stones and here we can see the grass and trees on the ground. In the background we can see the sky. |

Table 30: The query image 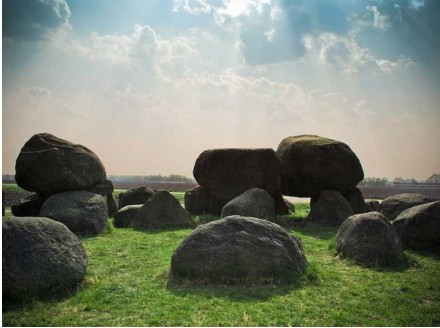 is used to retrieve the captions.

| Model | Rank | Retrieved Caption |
|---|---|---|
| Global DINO | 1 | In this image in the foreground there is a tree, and at the bottom there is a river and in the background there are hills and trees. And at the top there is sky. |
| | 2 | In this image, there are mountains and clouds. |
| | 3 | In this picture we can see hills in the background, it looks like snow at the bottom, we can see the sky at the top of the picture, there are clouds in the middle. |
| | 4 | In this image, there is a sewing machine. Under this sewing machine, there is a sheet. On the right side of this image, there is a gray color sheet. And the background of this image is dark in color. |
| | 5 | In this image, there are few electronic devices on a desk and also there are drawers at the bottom. Beside, there is a woman sitting on the chair on the floor and operating a computer. In the background, there are glass walls, few lights to the ceiling, a chair, dustbin, few electronic devices on the desk and also there are few drawers on the right. |
| Local DINO | 1 | In this image there is a dog on the rock. Behind the dog there are plants and trees. In the background of the image there are mountains. At the top of the image there are clouds in the sky. |
| | 2 | In this image I can see a person standing on the mountain, he is wearing a bag. There are few rocks on the mountain, I can see there are few mountains in the background and the sky is clear. |
| | 3 | In this image we can see a person sitting on a rock, in front of the person there is a water flow and there is a dog. In the background there is a snow on the mountains and the sky. |
| | 4 | In the image there are cats sitting on the stones. At the top of the image there is water. |
| | 5 | In this picture there are dogs and we can see rocks, trees and water. In the background of the image we can see hills and sky with clouds. |
| Global CLIP | 1 | In this image there is a dog on a couch which is on the floor. Background is blurry. |
| | 2 | In the image I can see the picture of a dog which is on the seat. |
| | 3 | In the image there is a dog on a couch. And also there is a pillow and a towel. |
| | 4 | In this image there are two dogs on a sofa, on that dogs there is a blanket, in the background there is a wall. |
| | 5 | In this picture, we see a dog is on a sofa or on a bed. The dog is in black and white color. The leash of the dog is in red color. In the background, we see a bed sheet or a cloth in brown color. |
| Local CLIP | 1 | In the image there is a dog walking on the rock surface, behind the dog there are huge rocks. |
| | 2 | In this image we can see a person sitting on a rock, in front of the person there is a water flow and there is a dog. In the background there is a snow on the mountains and the sky. |
| | 3 | In this image, I see a group of dogs standing on a stone field with belts around their necks and I see couple of persons standing beside them. |
| | 4 | At the bottom of the image there is a dog in the water. In the background there are rocks. And also there are few stones in the water. |
| | 5 | In this picture there are dogs and we can see rocks, trees and water. In the background of the image we can see hills and sky with clouds. |

Table 31: The query image 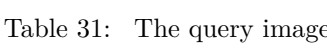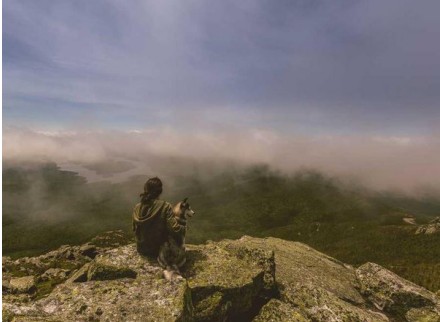 is used to retrieve the captions.

| Model | Rank | Retrieved Caption |
|---|---|---|
| Global DINO | 1 | In this image I can see two umbrellas with lot of colors like pink, blue, yellow, orange, red, green, and the sky is cloudy. |
| | 2 | In this image I can see different color of umbrellas. In the background I can see clouds in the sky. |
| | 3 | This image consist an umbrella. In the middle, we can see a pole. The background is blue in color. |
| | 4 | This image consists of an umbrella along with a metal rod. At the top, there is sky. |
| | 5 | In this picture we can see a partial part of a colorful umbrella. We can see the top view of an umbrella. |
| Local DINO | 1 | In this image we can see a blue color umbrella and on the umbrella we can see some different paintings in different colors and in the background we can see a white color iron and we can see the tip of the umbrella is in red color. |
| | 2 | In the middle of the image we can see an umbrella on the dried grass. |
| | 3 | In this image, there is a black and red umbrella on the white surface. Behind it, there is a white wall. |
| | 4 | In this picture there is a woman holding the umbrella. At the back there is a wall. At the bottom there is a floor. |
| | 5 | In this image we can see a man holding an umbrella. |
| Global CLIP | 1 | In this image we can see a blue color umbrella and on the umbrella we can see some different paintings in different colors and in the background we can see a white color iron and we can see the tip of the umbrella is in red color. |
| | 2 | This image consist an umbrella. In the middle, we can see a pole. The background is blue in color. |
| | 3 | In this image, I can see an umbrella with colorful design and there is a thread. In the background, I can see glass windows. In the bottom left corner of the image, there is an object. |
| | 4 | In this image I can see two umbrellas with lot of colors like pink, blue, yellow, orange, red, green, and the sky is cloudy. |
| | 5 | In this image, we can see an umbrella. |
| Local CLIP | 1 | In this image we can see a man holding an umbrella. |
| | 2 | In this picture there is a woman holding the umbrella. At the back there is a wall. At the bottom there is a floor. |
| | 3 | In this image, we can see an umbrella. |
| | 4 | In the middle of the image we can see an umbrella on the dried grass. |
| | 5 | In this image, in the foreground we can see a person wearing a costume and hold an umbrella. On the right side, we can see a person. Background of the image is blurred. |

Table 32: The query image 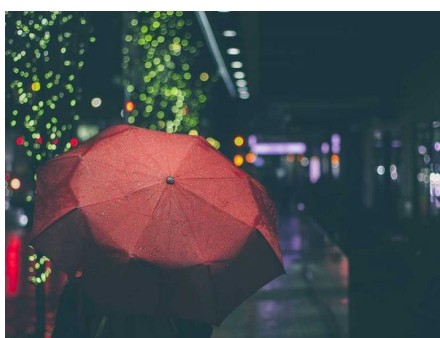 is used to retrieve the captions.

## H.5 Free-Form Caption → Image (OOD)

We evaluate SPARC's capability with free-form captions not present in the Open Images dataset, following the same methodology as Section H.2. Figures 39, 40, and 41 show retrieval results using simple descriptive captions to query the Open Images test database.

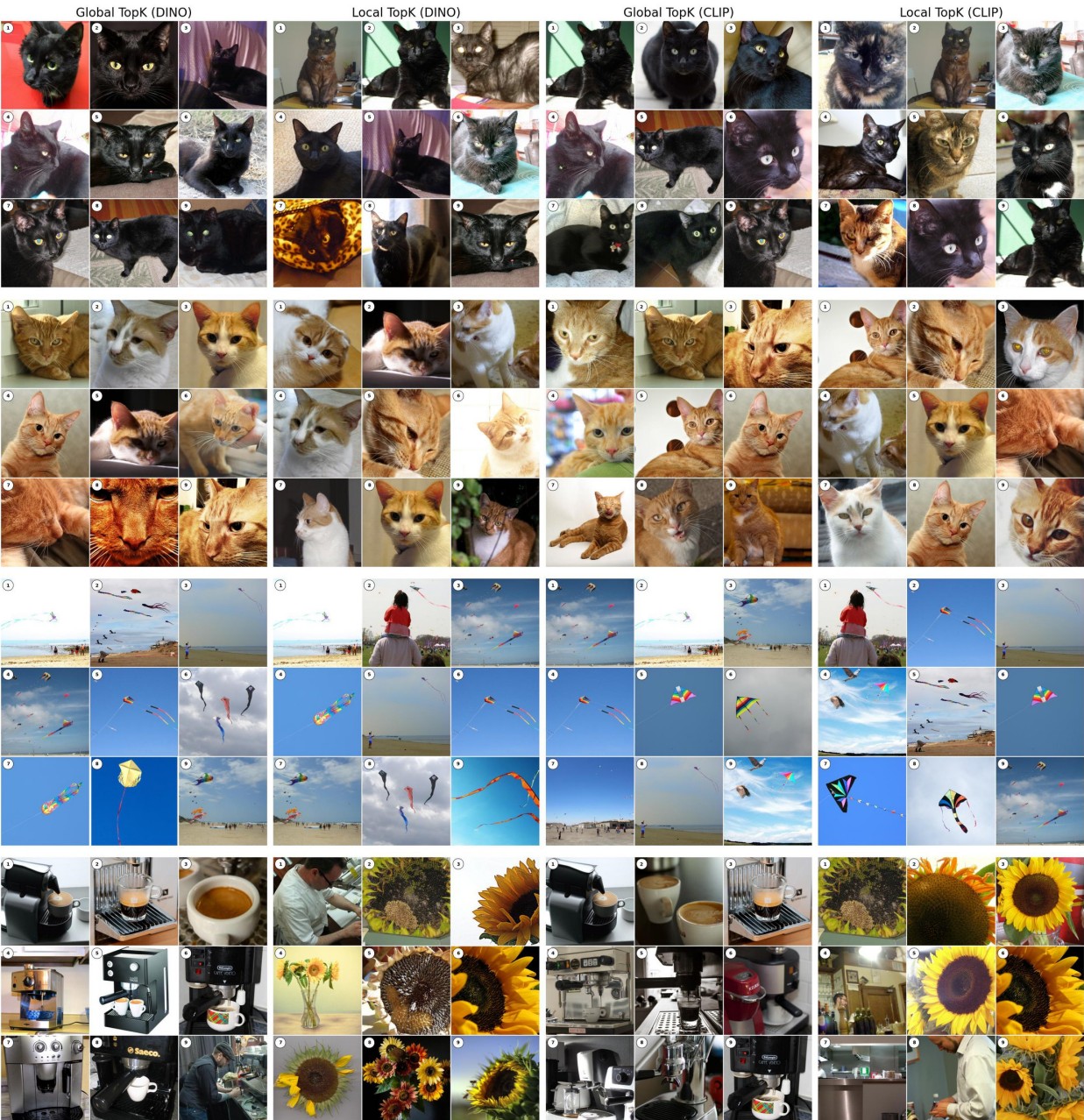

Figure 39: Images retrieved for captions: (1) "Black cat", (2) "Orange cat", (3) "A child flying a red kite on a sunny beach", (4) "A barista making coffee behind a counter." The images are retrieved from the test set of the Open Images dataset, but the captions are not part of the dataset.

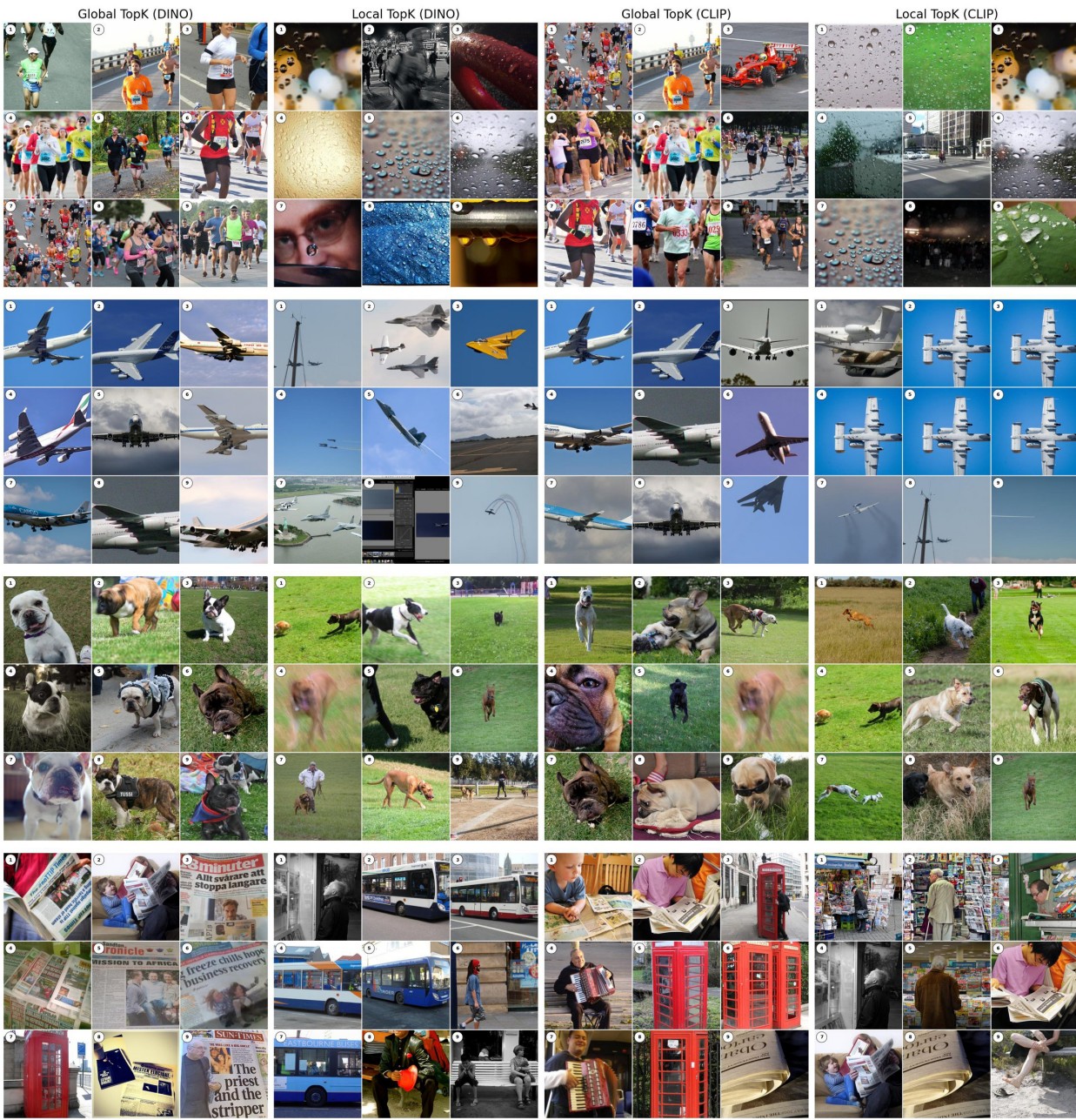

Figure 40: Images retrieved for captions: (1) "People crossing a busy city street in the rain", (2) "A passenger airplane flying in a clear blue sky", (3) "A dog running through a grassy field", (4) "A man reading a newspaper at a bus stop." The images are retrieved from the test set of the Open Images dataset, but the captions are not part of the dataset.

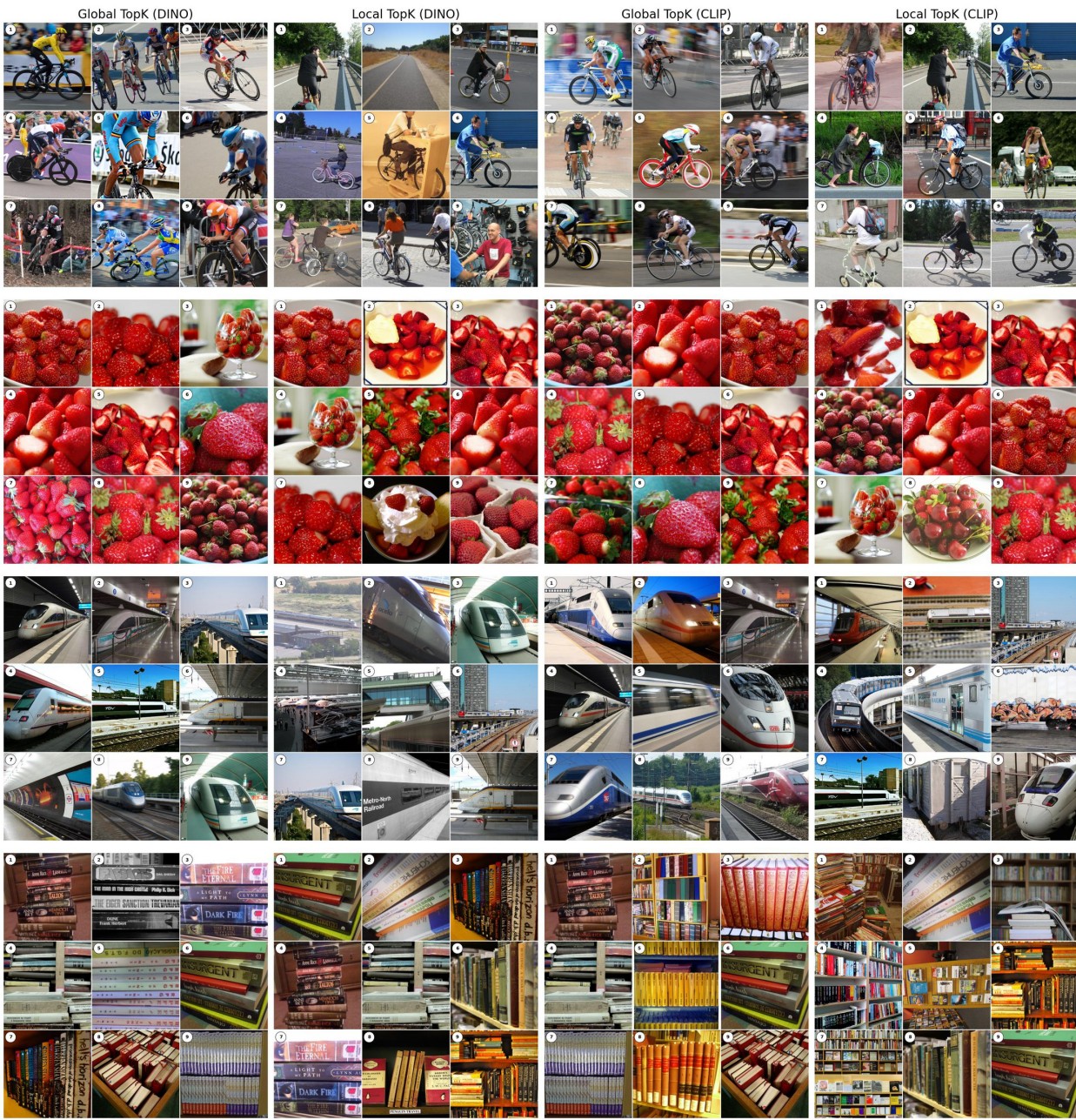

Figure 41: Images retrieved for captions: (1) "A person riding a bicycle on a country road", (2) "A bowl of fresh strawberries on a kitchen counter", (3) "A train arriving at an underground metro station", (4) "Books stacked on a wooden desk." The images are retrieved from the test set of the Open Images dataset, but the captions are not part of the dataset.

### H.6 External Image→Image (OOD)

We evaluate SPARC's out-of-distribution generalization using external images not present in the Open Images dataset for cross-model image retrieval. Figures 42, 43, and 44 show cross-model retrieval results using external query images with the Open Images test set as the reference database.

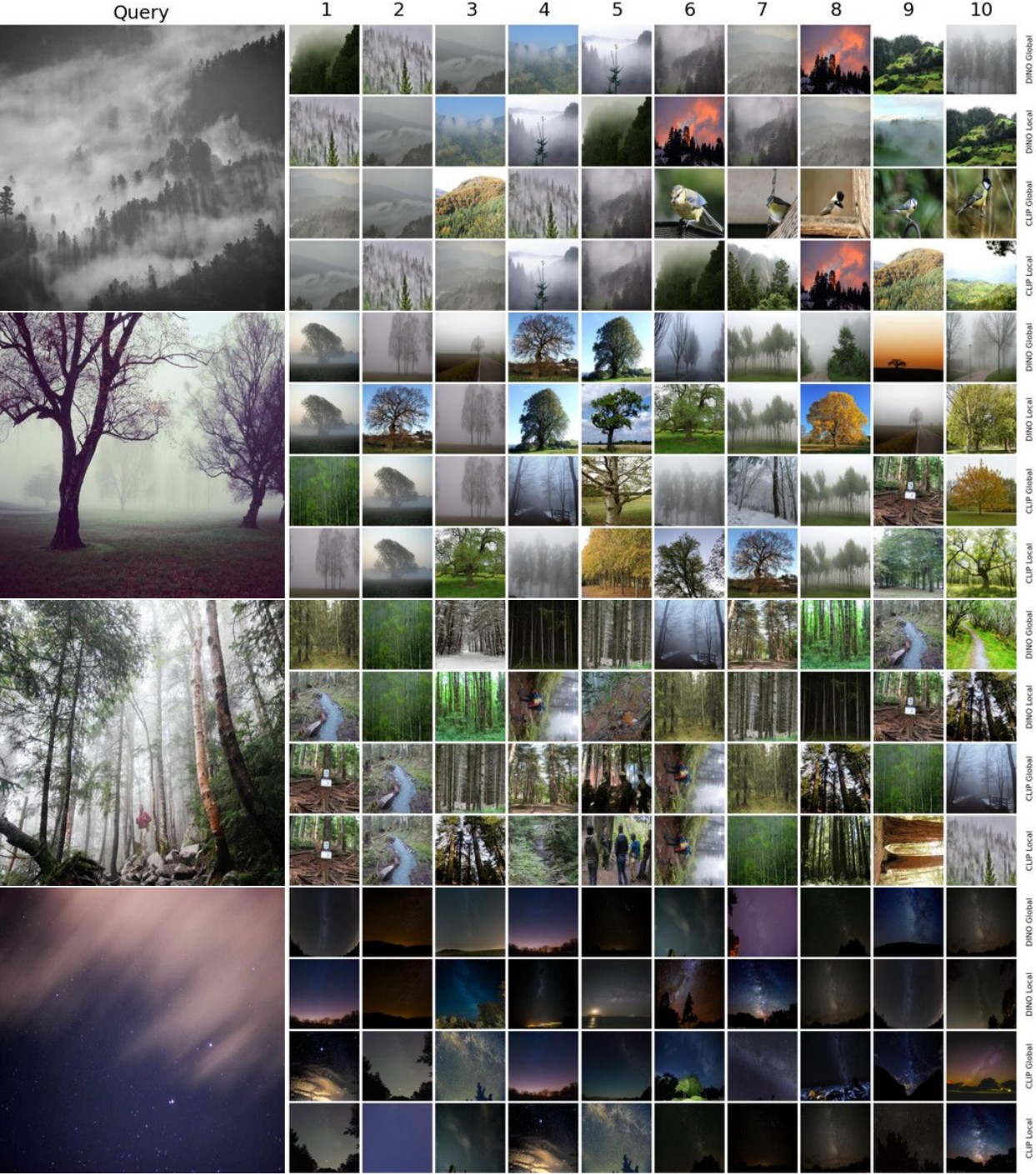

Figure 42: Cross-model image retrieval results using OOD query images. The references are from the Open Images test set.

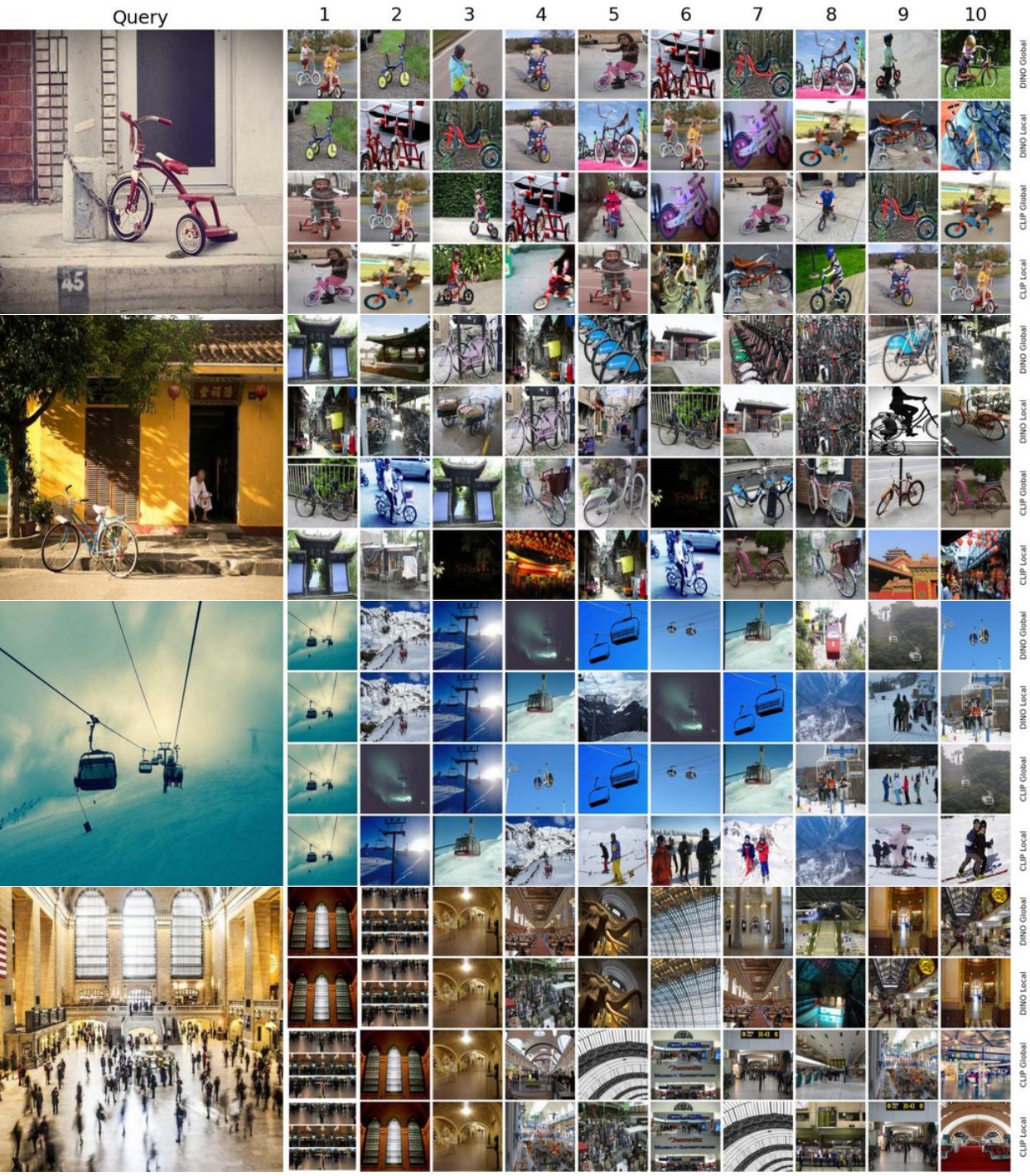

Figure 43: Cross-model image retrieval results using OOD query images. The references are from the Open Images test set.

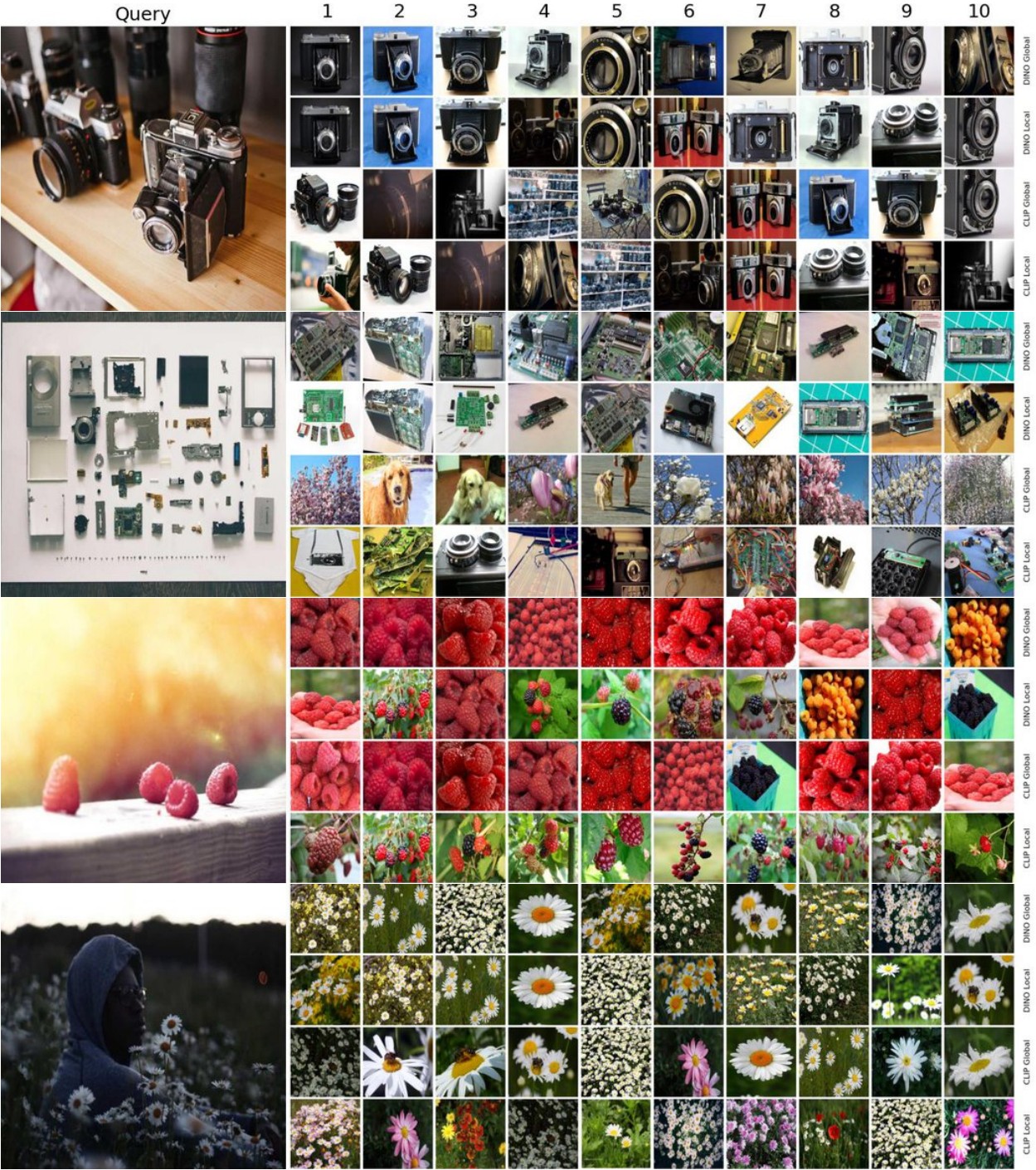

Figure 44: Cross-model image retrieval results using OOD query images. The references are from the Open Images test set.

