# OpenReview forum: "SPARC: Concept-Aligned Sparse Autoencoders for Cross-Model and Cross-Modal Interpretability"
_TMLR — Accepted by TMLR_

### Review · Reviewer_7VUR · 2025-11-26

**Summary Of Contributions:**

The authors aim to align the representations learned between different ML models and modalities. Previous approaches of (universal) sparse autoencoders tried to achieve this by randomly selecting one encoder to produce a latent vector and decode in all modalities. With the goal of improving latent-, and therefore, concept alignment, the authors propose to simultaneously produce a joint activation vector from all encoder outputs and select the same top-k indices from every latent stream. To further promote latent alignment the authors employ a cross-reconstruction loss to reconstruct expected latent outputs of one stream from those of the others.

The authors inspect latent/concept alignment by measuring the simultaneous activation/vanishing of activations across streams and Jaccard similarity to concepts. Results indicate that the global top-k loss strongly promotes joint activations. Furthermore, the authors inspect the ability to reconstruct latent across streams and monosemanticity showing moderate reconstruction and class prediction abilities, as well as improved alignments for activation maps and relevance scores. Finally, several variation experiments, e.g. on the number of latents and the top-k sparsity, are presented that seem to indicate a robust training and reinforce the requirement of the global top-k regularization.

**Additional Comments:**

I generally find the paper well written and the method to be clearly described. The proposed architecture and loss terms are well visualized and formalized. A general benefit of the method seems to be its straight forward implementation and general applicability, which might make it easy to extend to models beyond the ones tested. Furthermore, the learning of a joint 'drop-in' latent space leaves the weights of the original models unchanged, therefore avoiding costly retrainings or finetunings of model weights.

My main concern with the method lies with the point of 'stream separate latents' as described above. The authors seemingly propose a more complicated setup than necessary only to then counteract the induced complexity with additional loss terms. While I can imagine how the proposed multi-latent setup can be beneficial, in terms of performance and disentanglement, in contrast to an aggregated latent representation, I feel like the authors need to justify and compare benefits over the more simple approach.


**Further Comments:**
* Following Eq. (3), "$\mathcal{I}$global" is not properly set in subscript.
* Local TopK is first mentioned in Sec. 4.1 without any prior definition. While I assume it to simply be selecting the top-k activations from each latent stream individually, the authors might want to add a brief clarification.

**Audience:**

Yes

**Audience Explanation:**

While the presented results show moderate reconstruction abilities, class prediction abilities, and partly noisy heatmaps, the method seem to generally be a step in the right direction on post-training representation alignment. More generally, the proposed method seems easy to plug into different models without costly retraining steps. The experiments are well conducted, highlighting strengths and possible weaknesses of the proposed approach.

**Claims And Evidence:**

Yes

**Claims Explanation:**

The authors propose a well formulated method on improving cross-model and cross-modal representation alignment. While the obtained results show moderate improvements in representation alignment, the authors do not overstate contribution and accurately represent the abilities of their method. Claims in are backed up with data and the interpretation of results in throughout the paper seems to align with the obtained results.

**Requested Changes:**

**Linearity of Latents.** In Sec. 3.2 the authors detail an linear affine transformation from model features into/from the shared latent space. Aligning latent spaces between models via affine transformations requires that features are already linear within the models. While there has been some evidence that larger baseline models tend to learn linearized representations, this assumption should be made explicit and might be tested in the paper. (Alternatively, nonlinear transforms could be considered instead, e.g., two-layer NN, although I'm aware that such linear transforms are commonly accepted in literature and would require a whole new training and evaluation run.)

**Stream separate latents.** In Eq. (4) the authors create separate 'intermediate' latent vectors $z^s$ for every stream.  Given that $\hat{x}^s$ is expected to be produced once by the first term via $D\_s(z^s)$ and equally by all other streams via $D\_s(z^t)$ in the second term, wouldn't this imply $z^s \approx z^t$ anyway? More generally, I'm wondering why the authors choose this seemingly more complicated setup of creating individual latents per stream, which are only then implicitly enforced to obtain the same value via the second loss term in Eq. (6). Wouldn't using $\mathbf{h}\_{agg}$ (or its average version) in Eq. (4) suffice?



**Standard Errors.** Across all tables the authors only report means. The paper might be improved by reporting standard error where applicable. (E.g., per latent dimension or stream variances in table 2, ...)

---

> ### Author Response · Authors · 2026-01-06
> **Response to Reviewer 7VUR**
>
> We thank the reviewer for their assessment and for highlighting both strengths and key conceptual concerns.
>
> 1. Linearity of Latents
>
> Comment: In Sec. 3.2 the authors detail a linear affine transformation from model features into/from the shared latent space. Aligning latent spaces between models via affine transformations requires that features are already linear within the models. While there has been some evidence that larger baseline models tend to learn linearized representations, this assumption should be made explicit and might be tested in the paper. (Alternatively, nonlinear transforms could be considered instead, e.g., two-layer NN, although I'm aware that such linear transforms are commonly accepted in literature and would require a whole new training and evaluation run.)
>
>
> Response:
> Thank you for pointing this out. We agree that the linearity assumption is currently implicit. In the revised version, we now state explicitly that each stream’s encoder and decoder are affine (linear + bias) maps from the frozen features into and out of the shared latent space. This follows common practice in sparse autoencoders and linear probes, where late-layer representations of large models (such as CLIP and DINO) are routinely shown to support strong linear readouts. Our reconstruction $R^2$ scores and 1D probe results are consistent with this, suggesting that a linear latent layer already captures substantial structure. Exploring shallow nonlinear heads (e.g., two-layer MLPs) is a natural extension, but it would require retraining all configurations; we therefore keep the linear design in this submission and explicitly flag this as an assumption and direction for future work. As the reviewer notes, linear transforms are widely accepted in the literature for this purpose, and moving beyond them would entail a full new training and evaluation run.
>
>
> 2. Stream separate latents.
>
> Comment: In Eq. (4) the authors create separate `intermediate' latent vectors $z^s$ for every stream. Given that $\hat{x}^s$ is expected to be produced once by the first term via $D_s(z^s)$ and equally by all other streams via $D_s(z^t)$ in the second term, wouldn't this imply $z^s \approx z^t$ anyway? More generally, I'm wondering why the authors choose this seemingly more complicated setup of creating individual latents per stream, which are only then implicitly enforced to obtain the same value via the second loss term in Eq. (6). Wouldn't using $\mathbf{h}_{agg}$ (or its average version) in Eq. (4) suffice?
>
> Response:
> The design is intentional: Global TopK is used to enforce a \emph{shared support} (which indices are active) across streams, but we do not want to force $\mathbf{z}^s = \mathbf{z}^t$. In practice, for a shared concept, we expect $\mathbf{z}^s$ and $\mathbf{z}^t$ to be similar on the active coordinates, but exact equality would be unnecessarily restrictive.
>
> There are two main reasons. First, the feature spaces of DINO, CLIP-image, and CLIP-text are different. Even when they represent the same concept, the magnitudes needed on a given latent dimension for good reconstruction need not match. A single shared latent (e.g., derived directly from $\mathbf{h}_{\text{agg}}$ or its average) would enforce one compromise set of coefficients and can hurt self- and cross-reconstruction. Second, one of our goals is to compare how strongly different models use the same latent dimension on the same input. Separating $\mathbf{z}^s$ in a shared index basis makes these differences visible; collapsing to one $\mathbf{z}$ would hide them. The cross-reconstruction term encourages $\mathbf{z}^s$ and $\mathbf{z}^t$ to be mutually useful, not identical. In the revised version, we now state this explicitly in Sec.~3.2.
>
>
> 3. Standard errors.
>
> Comment: Across all tables the authors only report means. The paper might be improved by reporting standard error where applicable. (E.g., per latent dimension or stream variances in Table 2, \dots)
>
> Response:
> We added standard errors where appropriate to the main quantitative tables (e.g., averaged concept-alignment) so that the variability across latents or samples is visible alongside the mean values.
>
>
>
> 4. Minor clarity issues.
>
> Comment: Following Eq.~(3), ``global'' is not properly set in subscript. Local TopK is first mentioned in Sec.~4.1 without any prior definition. While I assume it to simply be selecting the top-$k$ activations from each latent stream individually, the authors might want to add a brief clarification.
>
> Response:
> Thank you for pointing out these formatting and clarity issues. We fixed the subscript formatting for ``global'' following Eq.~(3). We also added a brief definition of Local TopK at its first occurrence, stating explicitly that it applies the TopK operation independently to each stream’s logits $\mathbf{h}^s$ (i.e., selecting the top-$k$ activations per stream).

---

### Review · Reviewer_w77c · 2025-11-28

**Summary Of Contributions:**

The paper proposes SPARC, a sparse autoencoder framework that learns a shared, interpretable latent space across heterogeneous model architectures and modalities, overcoming the isolation of concept spaces produced by standard SAEs. SPARC introduces two innovations to achieve aligned representation: a Global Top-K sparsity mechanism that enforces consistent latent activations across streams, and a cross-reconstruction loss that encourages semantic consistency between models. Experiments demonstrate large gains in concept alignment, with Jaccard similarity improving to 0.80 compared to 0.26 for Local Top-K, along with a substantial reduction in dead or partially aligned neurons and improved cross-stream reconstruction fidelity. These aligned sparse latents also enable practical applications, including cross-modal attribution and weakly-supervised segmentation in vision models.

**Audience:**

Yes

**Audience Explanation:**

The work directly advances cross-model interpretability, a topic of broad interest in both vision and multimodal communities. The method is applicable to many model families and aligns well with current interest in feature-level interpretability, sparse dictionary learning, and unified concept spaces across architectures.

**Broader Impact Concerns:**

The work unlocks cross-model interpretability, which is largely positive. However:
- Shared concept spaces could enable model-to-model transfer of harmful or private representations if misused.
- Text-guided spatial localization can be misapplied for surveillance or privacy-sensitive content analysis.

I recommend adding a Broader Impact section noting dual-use risks and proposed mitigations. No critical ethical flaws identified.

**Claims And Evidence:**

Yes

**Claims Explanation:**

Overall, the empirical evaluation supports the central claims regarding cross-model concept alignment. The paper includes comprehensive ablations demonstrating that both core components, namely Global Top-K sparsity and cross-reconstruction loss, are necessary to achieve high alignment performance. This is supported by the substantial increase in Jaccard similarity to 0.80 and the elimination of mixed-activation neurons across streams, as well as improvements in cross-stream reconstruction quality. The experiments also present practical applications such as cross-modal attribution and weakly supervised segmentation, providing additional confirmation that the learned latents retain semantically meaningful information across modalities.

That said, some interpretability claims would benefit from deeper validation. While the work argues for improved monosemanticity, the evaluation does not include automated interpretability scores or human annotations of latent semantics. In addition, comparisons to related approaches such as USAE remain primarily qualitative, and a more direct quantitative comparison would strengthen the argument for superiority. Despite these limitations, the evidence is largely convincing with respect to the paper’s primary objectives of achieving structurally and semantically aligned latent representations.

**Requested Changes:**

1. Add quantitative comparisons against USAE: Provide numerical alignment metrics to strengthen claims of improvement over prior cross-model SAE approaches.
2. Include additional interpretability evaluation: Incorporate ablation experiment metrics or human-verified labeling to better validate interpretability claims beyond alignment.
3. Expand the diversity of evaluated streams: Add more modalities or distinct model types in the core experiments to support generalization claims.
4. Provide computational cost and runtime analysis: Clarify the efficiency trade-offs introduced by Global Top-K selection and cross-reconstruction loss relative to baseline approaches.
5. Discuss segmentation and attribution limitations: Offer a more nuanced evaluation of failure modes or conditions where SPARC’s downstream performance remains modest.

---

> ### Author Response · Authors · 2026-01-06
> **Response to Reviewer w77c**
>
> We thank the reviewer for the detailed and constructive feedback, especially the requests for stronger quantitative comparison to USAE and more explicit interpretability evaluation. Below, we address each point in turn.
>
> Part 1/2
>
> 1. Quantitative Comparison with USAE
>
> Comment: Add quantitative comparisons against USAE: provide numerical alignment metrics to strengthen claims of improvement over prior cross-model SAE approaches.
>
> Response:
> We agree that a direct quantitative comparison with USAE is important to substantiate our claims. To avoid duplication across reviewer threads, we present the full set of USAE results in our "General Response on USAE Comparison" comment and summarize the key findings here. Using the authors' official implementation of USAE (adapted only to our Open Images data pipeline), we compute the same metrics as for SPARC: Jaccard-based concept alignment, self- and cross-stream $R^2$, and activation-consistency statistics. We incorporated these results into the revised manuscript. Briefly, USAE's concept alignment lies close to our Local TopK regime and well below SPARC with Global TopK + cross-reconstruction, and USAE fails to reconstruct DINO reliably, with near-zero or negative cross-stream $R^2$, whereas SPARC maintains strong, balanced transfer.
>
>
>
> 2. Additional Interpretability Evaluation
>
> Comment:
> Include additional interpretability evaluation: Incorporate ablation experiment metrics or human-verified labeling to better validate interpretability claims beyond alignment.
>
>
> Response:
> To assess whether individual latent dimensions correspond to coherent semantic concepts, we measure "label purity" on Open Images. For each latent and stream, we order examples by activation magnitude and consider a subset of examples with the largest activations (we vary the subset size as shown in the table below). For a given latent, purity is defined as the fraction of these examples that contain the most frequent Open Images label within that subset. Because Open Images is multi-label, we normalize by the number of examples rather than the total number of label instances, so that additional annotations on the same image do not artificially reduce purity. We then report the mean purity across active latents.
>
> The table below compares SPARC (Global TopK, $\lambda=1$) against USAE. Across all subset sizes and all three streams, SPARC exhibits consistently higher label purity. For example, when using 25 strongly activating examples per latent, SPARC attains mean purities of $0.7416/0.6825/0.7403$ for CLIP-Image/CLIP-Text/DINO, while USAE achieves $0.5525/0.4941/0.5862$. Even when all activations are considered, SPARC maintains higher purity in every stream. These results indicate that SPARC’s latents not only align across models but also concentrate their strongest responses on semantically coherent sets of images.
>
> | #Examples | SPARC CLIP-Image | SPARC CLIP-Text | SPARC DINO | USAE CLIP-Image | USAE CLIP-Text | USAE DINO |
> |:---|:---:|:---:|:---:|:---:|:---:|:---:|
> | 10 | 0.7950 | 0.7256 | 0.7960 | 0.6014 | 0.5479 | 0.6426 |
> | 25 | 0.7416 | 0.6825 | 0.7403 | 0.5525 | 0.4941 | 0.5862 |
> | 50 | 0.6950 | 0.6495 | 0.6892 | 0.5235 | 0.4681 | 0.5427 |
> | 100 | 0.6395 | 0.6121 | 0.6298 | 0.4947 | 0.4468 | 0.4960 |
> | All | 0.5300 | 0.5256 | 0.5275 | 0.4203 | 0.4018 | 0.4201 |
>
> Label purity is an imperfect and noisy proxy for interpretability, especially under Open Images' multi-label annotations, but it is fully automatic and cheap to compute. Here, we use it primarily as a relative measure to contrast SPARC and USAE under identical conditions, rather than as an absolute notion of semantic quality. We added this label purity comparison as a new Table 3 in Section 4.3, along with the methodology description for the metric.
>
>
> 3. More streams
>
> Comment: Expand the diversity of evaluated streams: Add more modalities or distinct model types in the core experiments to support generalization claims.
>
> Response: While the main text focuses on the three-stream CLIP-image/CLIP-text/DINO setting for concept alignment and interpretability, Appendix C already evaluates cross-stream reconstruction across up to ten encoders (CLIP-Image, CLIP-Text, SigLIP-Image, SigLIP-Text, DINOv2, ViT, Swin, E5, GTE, Qwen) using full $R^2_{s \to t}$ matrices for CLIP-only, SigLIP-only, vision-only, text-only, image-only, and the full ten-stream configuration. These results show that Global TopK yields stable, non-pathological $R^2$ across a wide range of architectures, whereas Local TopK and USAE can produce large negative $R^2$ values (especially for DINO and Swin). In the revision, we added explicit pointers from the main experiments to Appendix C and clarified that the three-stream setup is used for detailed alignment and interpretability analyses, while the ten-stream setup probes broader cross-model transfer.

---

> > ### Author Response · Authors · 2026-01-06
> > **Response to Reviewer w77c (Part 2/2)**
> >
> > Part 2/2
> > 4. Computational Cost
> >
> > Comment: Provide computational cost and runtime analysis: Clarify the efficiency trade-offs introduced by Global Top-K selection and cross-reconstruction loss relative to baseline approaches.
> >
> > Response: We have added a new appendix section, A.4 Computational Cost, that reports both complexity and wall-clock runtimes. In all experimental configurations on Open Images, we first run each frozen encoder once over the Open Images training set, pre-extract the features, and store them on disk; SPARC then trains only the linear encoder/decoder layers on these stored features. On a single H100 GPU, this results in less than 30 seconds per epoch for a full pass over the Open Images training set in the three-stream CLIP-image/CLIP-text/DINO setup (i.e., under 25 minutes for 50 epochs). For the ten-stream configuration in Appendix C, it takes around 2.5 minutes per epoch for a full pass over the same training set (about 2 hours for 50 epochs) under the same feature pre-extraction strategy. All reported experiments use standard dense torch.nn.Linear layers; in principle, the sparse activations produced by Global TopK would allow further speedups with sparse or Triton kernels, but we did not rely on such optimizations. For context, the USAE paper reports training times on the order of three days (Appendix A.1 of their paper), without pre-extracting encoder features; we cited this to give readers a rough sense of SPARC’s relative cost.
> >
> >
> >
> > 5. Segmentation
> >
> > Comment:
> > Discuss segmentation and attribution limitations: Offer a more nuanced evaluation of failure modes or conditions where SPARC’s downstream performance remains modest.
> >
> > Response:
> > We expanded Section F.3 of the Appendix to discuss failure modes of the segmentation and highlighted it clearly in the main text.
> >
> >
> > 6. Broader Impact
> >
> > Thank you for the suggestion. We added a Broader Impact section explicitly addressing these dual-use risks. The revised text notes that SPARC can support positive use cases such as cross-model auditing, debugging, and safety analysis, but could also be misused to transfer harmful concepts or localize sensitive content. We briefly discuss mitigation strategies, including avoiding sensitive targets, restricting use in high-risk domains, and coupling SPARC with appropriate governance and auditing practices.

---

### Review · Reviewer_8EwW · 2025-12-15

**Summary Of Contributions:**

The authors introduced a method (SPARC) that learns a unified latent space across modalities (CLIP image, CLLP text and DINO). The main innovations over vanilla SAE are: 1. a cross-modality Top K mechanism (as related to Top K in SAEs); 2. a cross-modality re-construction loss. Both innovations were shown to improve cross-modality re-construction loss over standard SAEs and CLIP similarity.

**Strengths**:

1.	The method is well-motivated and intuitive.

2.	The evaluation covered multiple aspects of desired outcomes.

3.	Abalation studies clearly demonstrate the merits of the innovations.

**Weaknesses**:

1.	A direct comparison with one existing method (USAE) seems to be missing.
2.	The improvement in cross-modality alignment comes at the cost of self-reconstruction.

**Audience:**

Yes

**Audience Explanation:**

Learning a unified latent space across modalities is interesting and valuable for composite tasks.

**Claims And Evidence:**

Yes

**Claims Explanation:**

Overall, the experiments are nuanced and well explained, supporting the claim of cross modality performance. Please refer to the requested change for one statement where the evidence seems insufficiently presented.

**Requested Changes:**

In the last paragraph of the introduction, it was briefly mentioned that USAE suffers from “inconsistent concept activation across models” and this weakness is addressed by this paper (SPARC). This point seems insufficiently supported by evidence: the experiments showed that SPARC produced consistent concept activation but not how USAE underperformed in this aspect. It would be helpful to add citations or experiments that supports the claim.

Typo: Figure 3 “with \lambda = 1 (top right)”: should this be “with \lambda = 0 (top left)”?

---

> ### Author Response · Authors · 2026-01-06
> **Response to Reviewer 8EwW**
>
> We thank the reviewer for their feedback, and we are glad the motivation and ablations were found clear and convincing. Below, we address the two requested changes.
>
>
> 1. Evidence for the USAE statement in the Introduction
>
> Comment: The introduction states that USAE suffers from “inconsistent concept activation across models,” but the submission did not show how USAE underperforms on this aspect. Please add citations or experiments to support the claim.
>
> Response: We agree that this statement should be supported directly in our evaluation. To avoid redundancy across reviewers, we address this request in another comment "General Response on USAE Comparison", where we add a quantitative USAE baseline using the USAE's official implementation under our Open Images evaluation protocol (concept-alignment/Jaccard, $R^2$ reconstruction, and activation consistency). In the revision, we also updated the last paragraph of the Introduction to explicitly point to this new evidence (and phrase the comparison precisely in terms of the lack of an explicit shared-support activation constraint in USAE versus SPARC's Global TopK, which enforces identical active indices across streams).
>
>
>
> 2. Figure 3 caption (“$\lambda=1$ top right”)
>
> Comment: Typo: Figure~3 “with $\lambda = 1$ (top right)”: should this be “with $\lambda = 0$ (top left)”?
>
> Response: The figure panels correspond to the intended configurations, and this is not a typo. The top right panel showcases that cross-reconstruction alone could lead to mixed activation patterns, which is undesirable, as quantified in Table 1. We clarified this in the updated caption.

---

### Author Response · Authors · 2026-01-06
**General Response on USAE Comparison**

Two reviewers (8EwW and w77c) requested a more direct and quantitative comparison to Universal Sparse Autoencoders (USAE). To address this, we report USAE results obtained using the authors' official implementation (i.e., we did not re-implement USAE). We only adapted the data pipeline to run on Open Images, ensuring it matches our evaluation setup and stream configurations. For reproducibility, we document the complete USAE baseline configuration in Appendix A.5.



Below, we summarize USAE performance using the same evaluation criteria as SPARC: (1) concept alignment via the Jaccard score, (2) within- and cross-stream reconstruction via $R^2$, and (3) activation pattern consistency (dead/mixed latents across streams). This provides a controlled, quantitative baseline for assessing how SPARC differs from USAE.


1. Concept Alignment (Jaccard Score)
We compare concept alignment using the same Jaccard score defined in Section 4.2 and reported for SPARC in Table 2. Under this evaluation, SPARC’s alignment is high only when Global TopK is paired with cross-reconstruction (Global+Cross: $\mu=0.8018$), whereas Local TopK variants are substantially lower (e.g., Local+Cross: $\mu=0.2599$). Using the official USAE implementation (adapted only to Open Images), we compute the same score and obtain USAE: $\mu=0.2166$, which is close to the Local TopK regime and far below SPARC Global+Cross. Taken together, these results indicate that a reconstruction-driven shared dictionary alone does not yield an index-consistent concept space under the Section 4.2 alignment evaluation. We updated Table 2 in Section 4.2 to include these results.

2. Cross-Stream Reconstruction $R^2$ Comparison
We evaluate cross-model transfer using the $R^2_{s\to t}$ protocol from Section 4.4 (rows = targets $t$, columns = sources $s$, bold diagonals = self-reconstruction). The table below compares SPARC and USAE on Open Images. SPARC demonstrates strong, balanced cross-stream transfer with all $R^2$ values in the $0.407$–$0.559$ range, notably achieving high reconstruction into DINO ($0.559$ from clip_img, $0.407$ from clip_txt) alongside a robust DINO self-term ($0.690$). In contrast, USAE fails to reconstruct DINO: cross $R^2$ values are near-zero ($0.017$) or negative ($-0.005$), with a weak self-term ($0.111$).

| Target | SPARC clip_img | SPARC clip_txt | SPARC dino | USAE clip_img | USAE clip_txt | USAE dino |
|:---|:---:|:---:|:---:|:---:|:---:|:---:|
| clip_img | 0.663 | 0.513 | 0.556 | 0.506 | 0.389 | 0.421 |
| clip_txt | 0.526 | 0.725 | 0.519 | 0.474 | 0.616 | 0.454 |
| dino | 0.559 | 0.407 | 0.690 | 0.017 | -0.005 | 0.111 |


We updated the tables in the paper and in Appendix C to include USAE reconstruction results, showing a similar trend across different stream configurations.

3. Activation Consistency and Active Latents

Similar to Table 1 of the paper, we compare activation consistency across streams. SPARC with Global TopK exhibits perfectly balanced activation (84.4\% in each stream), ensuring consistent cross-stream utilization. In contrast, USAE shows severe imbalance: 69.0\% activation in CLIP-Image, 61.0\% in CLIP-Text, and 91.0\% in DINO. This indicates that many latents are inactive in some streams while remaining active in others, preventing effective cross-stream reconstruction. We updated the table to include these results.

| Model | CLIP-Image | CLIP-Text | DINO |
|:---|:---:|:---:|:---:|
| SPARC (Global TopK, $\lambda=1$) | 84.4% | 84.4% | 84.4% |
| USAE | 69.0% | 61.0% | 91.0% |




In addition to the above results, we also extended our existing 1D probe (Sec. 4.5) and MS-COCO weakly supervised segmentation experiments (Sec.~4.6) to include USAE as an additional baseline; the corresponding tables now report USAE alongside SPARC and its ablations, and in both settings USAE underperforms SPARC by a significant margin, with consistently higher probe losses and noticeably lower AP and mIoU scores.

---

### Decision · Action_Editor_vGhU · 2026-03-03

**Recommendation:** Accept as is

**Audience:**

Yes

**Audience Explanation:**

All reviewers are in agreement that the paper would be of interest to the TMLR community, particularly due to its advancements in cross-model interpretability, making progress in post-training representation alignment. Reviewers highlight that the method's implementation is simple and accessible to be plugged into many different models. Overall, the experiments are well done, showing a pragmatic view on the method, highlighting both the strengths and weaknesses of the approach.

**Claims And Evidence:**

Yes

**Claims Explanation:**

The paper introduced a method termed SPARC that learns a shared, interpretable latent space across different model architectures and modalities (CLIP image, CLLP text and DINO), overcoming the isolation of concept spaces produced by standard sparse autoencoders. All three reviewers agree that the claims are supported by accurate and convincing evidence. To get aligned representations, the authors introduce two mechanisms: Global TopK sparsity that enforces consistent latent activations across streams and a cross-reconstruction loss that encourages semantic consistency between models. Experiments demonstrate large improvements in concept alignment.

Although there were some concerns with respect to missing quantitative comparisons against USAE and including additional interpretability evaluation, they were addressed during the rebuttal period. Overall, the reviewers are convinced that the proposed claims are supported with sufficient evidence.